# Design of a water-soluble transmembrane receptor kinase with intact molecular function by QTY code

Mengke Li [1,2], Hongzhi Tang [1], Rui Qing [1], Yanze Wang[3], Jiongqin Liu[1], Rui Wang[1], Shan Lyu[1], Lina Ma[1], Ping Xu [1] ✉, Shuguang Zhang [2] ✉ & Fei Tao [1] ✉

Membrane proteins are critical to biological processes and central to life sciences and modern medicine. However, membrane proteins are notoriously challenging to study, mainly owing to difficulties dictated by their highly hydrophobic nature. Previously, we reported QTY code, which is a simple method for designing water-soluble membrane proteins. Here, we apply QTY code to a transmembrane receptor, histidine kinase CpxA, to render it completely water-soluble. The designed CpxA^QTY exhibits expected biophysical properties and highly preserved native molecular function, including the activities of (i) autokinase, (ii) phosphotransferase, (iii) phosphatase, and (iv) signaling receptor, involving a water-solubilized transmembrane domain. We probe the principles underlying the balance of structural stability and activity in the water-solubilized transmembrane domain. Computational approaches suggest that an extensive and dynamic hydrogen-bond network introduced by QTY code and its flexibility may play an important role. Our successful functional preservation further substantiates the robustness and comprehensiveness of QTY code.

The membrane-bound region, featuring strong hydrophobicity with large numbers of hydrophobic amino acids such as leucine, isoleucine, valine, and phenylalanine, marks a clear boundary between membrane proteins and water-soluble proteins and greatly hampers the study and application of membrane proteins[1]. Challenges associated with membrane proteins include low expression levels, high cost and laboriousness for detergent screening, low stability in detergent-solubilized forms, and greater difficulties in obtaining high-quality crystals[2]. Although constituting approximately 20% to 30% of cellular proteins and having great importance on all kinds of life activities[3], membrane proteins make up less than 1% of the proteins whose structures have been resolved up to November 2021[4]. An alternative approach to the problem is to redesign the membrane-bound region to render it water-

soluble. Since the early 2000s, successive efforts have proved its feasibility, with designed water-soluble variants showing various degree of solubility and preservation of structural characteristics[2,5–11]. Particularly, the structural study of the water-soluble analog of the potassium channel KcsA provided concrete evidence that water-solubilization design was capable of preserving the three-dimensional structure well[8]. A recent study accurately designed the water-soluble analogs of integral membrane protein folds, further demonstrating the feasibility of converting membrane protein folds into water-soluble protein folds[11]. However, all of these studies only showed limited functional preservation (such as ligand-binding) and no work showed any product with intact intrinsic molecular function, including membrane-bound regions (such as signal transduction). It is unclear

[1]State Key Laboratory of Microbial Metabolism, Joint International Research Laboratory of Metabolic and Developmental Sciences, School of Life Sciences and Biotechnology, Shanghai Jiao Tong University, Shanghai 200240, China. [2]Laboratory of Molecular Architecture, Media Lab, Massachusetts Institute of Technology, Cambridge, MA 02139, USA. [3]Department of Chemistry, Massachusetts Institute of Technology, Cambridge, MA 02139, USA. ✉e-mail: pingxu@sjtu.edu.cn; shuguang@mit.edu; taofei@sjtu.edu.cn

whether we can simultaneously render membrane proteins completely water-soluble and retain their functional integrity. The challenge is obvious, as we need to balance at least three factors: (1) solubility, to enable the protein to stay stable in the lipid-free milieu; (2) stability, to prevent properly folded structures from collapsing while systematically imposing a significant number of mutations; (3) activity, to avoid the disruption to the specific interaction network essential for the intrinsic biological activities. And it becomes much more challenging when there is no available experimental structure.

Previously, we conceived a simple QTY code (Q, Gln; T, Thr; Y, Tyr), a straightforward method for solubilizing water-insoluble proteins, especially transmembrane α-helices, without depending on any available structure input[12]. QTY code defines a clear correspondence between the major constituent hydrophobic residues in transmembrane domains and hydrophilic residues (Fig. 1a) mainly based on the similarity of their electron density maps and chemical structures. The fact is that Nature has evolved three chemically distinct α-helices, (1) Type I: hydrophilic helix, (2) Type II: hydrophobic helix, and (3) Type III: amphiphilic helix, with identical molecular structures[13]. The independence of available structure inputs and complex computational programs makes QTY code promising to be a simple method of water-soluble protein design. The initial application of QTY code was on chemokine receptors (a class of G-protein coupling receptors, GPCRs), exhibiting favorable water-solubility, stability, and ligand-binding affinity[12,14], and showing excellent sensitivity and robustness in a biomimetic sensing system[15]. Nonetheless, to what extent QTY code can preserve the biological activities needs to be further investigated.

Transmembrane signaling, taking place in all cells, is of great importance on communication with their environment, regulating numerous physiological activities. Two-component systems (TCSs), widespread in prokaryotes and some eukaryotes, are known as one of the best-studied models of signal transduction schemes[16,17]. In typical TCSs, sensor histidine kinases (SHKs), one of the most abundant receptors, function with cognate response regulators to manipulate the expression of genes associated with cell growth, metabolism, antibiotic resistance, etc[18,19]. Canonically, signals are sensed by the extra-cytoplasmic sensor domain and transduced by transmembrane helices and other signal-relay modules into the activity changes of the cytoplasmic catalytic domain. Due to the intrinsic difficulty in studying the structural biology, shared by most membrane proteins, the mechanism of transmembrane signaling by SHKs remained elusive, although the studies in recent years on transmembrane signaling mechanism provide more and more details[20–22]. We chose the SHKs as the design subject for both their biological significance and technical advantages. Histidine kinases are the most intensively studied and multitudinous membrane receptors in microorganisms, which control the greatest variety of cellular responses[23] and are promising to become new antimicrobial drug targets[19,24,25]. Importantly, their highly modular structure and signaling pattern allows easy and explicit examination of the functional effects introduced by design. Designed interhelical polar interaction networks have been proved to be capable of regulating the protein activities, with favorable plasticity and tunable specificity[26–28]. Theoretically, it is possible that designed interhelical polar interaction networks by QTY code may replace the interaction networks in transmembrane hydrophobic helices responsible for SHK signaling and realize signal transduction through conformational changes that mimic the original ones.

Here, we designed a water-soluble SHK, namely CpxA^QTY using QTY code (simplified below as "QTY design"). We showed that CpxA^QTY exhibited expected biophysical properties and intact molecular function, including three catalytic activities and signaling receptor activity. Moreover, by combining classical molecular dynamics and parallel bias metadynamics simulations, we probed the principles underlying the balance of solubility, stability, and activity of the water-solubilized transmembrane domain in the case of CpxA^QTY.

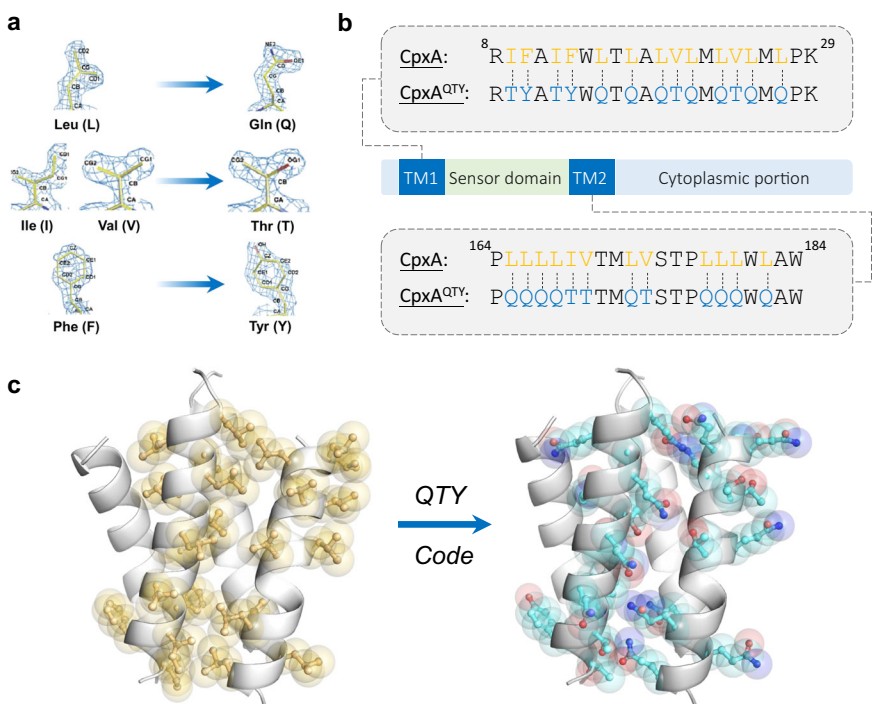

**Fig. 1 | QTY design of CpxA. a** Schematic diagram of QTY code. Crystallographic 1.5 Å electron density maps of Leucine(L), Asparagine (N), Glutamine (Q), Isoleucine (I), Valine (V), Threonine (T), Phenylalanine (F), and Tyrosine (Y) and their corresponding relationship defined by QTY code are shown. **b** Overall architecture of CpxA and comparison between the transmembrane sequences of CpxA and CpxA^QTY. Residue numbering is shown on the top of the sequences. The target hydrophobic residues are in orange and the polar residues introduced by QTY design are in blue. TM1/2, the first/second transmembrane helix. **c** Structural view of comparison between the transmembrane regions before (left) and after (right) QTY design. The structure models are from AlphaFold2 prediction. Only residues targeted by QTY design of one protomer are shown in ball-and-stick and transparent spheres, in yellow for CpxA and in cyan for CpxA^QTY.

## Results

### Design and biophysical characterization of CpxA^QTY

CpxA was selected as the subject of QTY design, as it was one of the most studied SHKs[29–32] and explicitly confirmed to respond to the signals in vitro[33]. We applied the simple mode of QTY design to CpxA, i.e., substituting all transmembrane Leu (L), Ile (I), Val (V), and Phe (F) (Fig. 1), automatically by our QTY enabling server[4]. The overall variation rate is 5.47% and the variation rate in the transmembrane domain is 58.1%. The basic characteristics, including isoelectric point (pI) and molecular weight (MW), of the design CpxA^QTY are preserved well (Supplementary Fig. 1a).

The synthetic gene encoding CpxA^QTY was obtained and the his-tagged protein was expressed in *Escherichia coli* (*E. coli*) and was purified by immobilized metal affinity chromatography (IMAC) and gel filtration. CpxA^QTY was expressed well as the soluble form (Fig. 2a). It is worth noting that as with the typical water-soluble proteins, we purified CpxA^QTY under the native and detergent-free condition in the entire process. Size exclusion chromatography (SEC) results showed that CpxA^QTY predominantly folded as dimers, the functional oligomeric state of a canonical SHK (Supplementary Fig. 1b). Previous biochemical studies also indicated that CpxA functions as dimers[30].

We then characterized the biophysical properties of CpxA^QTY. By analytical ultracentrifugation (AUC), we re-confirmed the oligomeric state of CpxA^QTY (Fig. 2b and Supplementary Fig. 1c) and the calculated friction ratio (1.560) indicated that its shape tended to be elongated, distinct from typical globular proteins[34]. This elongated shape was as expected and implied the correct fold as a canonical SHK. For thermostability, nano differential scanning fluorimetry (nanoDSF) revealed a melting temperature ($T_m$) of $53.3 \pm 1.8\,°C$ (Fig. 2c), which falls within the typical range of $T_m$ values of proteins in *E. coli*[35]. This value is also close to those of QTY variants of GPCRs[12]. To further characterize the folding state of CpxA^QTY, we conducted circular dichroism (CD) measurements (Fig. 2d). The analysis of the secondary structure composition revealed α-helix and β-sheet content of 60.9% and 15.0%, respectively, which closely matched the content in the predicted structure by AlphaFold2[36], 61.9% for α-helix and 17.5% for β-sheet. The high content of ordered secondary structures and the high consistency in the experimental and predicted secondary structure content support that CpxA^QTY is well-folded.

These results showed that CpxA^QTY exhibited favorable solubility and the expected biophysical characteristics.

### QTY design of CpxA retains the three catalytic activities

We then systematically investigated the molecular function of the QTY variant of CpxA, the common SHK with multiple activities (Fig. 3a). We first examined the basic autokinase activity of CpxA^QTY. Utilizing anti-phosphohistidine (pHis) western blotting, we found that CpxA^QTY showed a high autokinase activity (Fig. 3b). By mass spectrometry, we identified the phosphorylation of the expected phosphorylation site of CpxA^QTY, His248[29] (Supplementary Fig. 1d). We noticed that, after 10 min of autophosphorylation, CpxA in lipid environment (proteoliposomes, detergent or nanodiscs) approaches the saturation[29,30,33,37], whereas CpxA^QTY remains unsaturated, albeit with a significantly reduced rate. This may suggest a slight decrease in the autokinase activity of CpxA^QTY compared to CpxA.

In addition to the autophosphorylation, phosphorylated CpxA can transfer its phosphate to CpxR (phosphotransferase activity), the cognate response regulator of CpxA, and conversely, unphosphorylated CpxA can also dephosphorylate phosphorylated CpxR (phosphatase activity), to dynamically regulate the downstream gene expression[33]. Regarding phosphotransferase activity, employing [γ −32P] ATP and autoradiography, we observed that phosphorylated CpxA^QTY could transfer its phosphate to CpxR very rapidly (Fig. 3c). Phosphorylated CpxA^QTY became almost undetectable within 0.5 min, with the saturation of CpxR phosphorylation reached within 1 min; in

contrast, after 15 min of phosphotransfer, phosphorylated CpxA became barely detectable, with CpxR still acquiring phosphate groups[30,33]. This tends to suggest that the phosphotransferase activity of CpxA^QTY is higher than that of CpxA. We also confirmed the rapid phosphotransfer using anti-pHis western blotting (Supplementary Fig. 1e). For phosphatase activity, we showed that the phosphorylation level of CpxR significantly decreased after the addition of unphosphorylated CpxA^QTY (Fig. 3d), assayed by Phos-tag SDS-PAGE. In the case of CpxA, phosphorylated CpxR became barely detectable within 30 min after the addition of CpxA[33,37]; whereas after 120 min of dephosphorylation by CpxA^QTY, a small amount of phosphorylated CpxR could still be detected. Thus, we speculate that the phosphatase activity of CpxA^QTY is lower than that of CpxA.

These data suggest that CpxA^QTY has the native three catalytic activities, including autokinase, phosphotransferase, and phosphatase activity, although some differences in the activity levels were observed between CpxA^QTY and CpxA.

### CpxA^QTY remains sensitive to multiple signals

For QTY design for CpxA, the most important functional benchmark is the *transmembrane* signaling receptor activity (we use italic "*transmembrane*" to differentiate the signaling through the water-soluble transmembrane domain from the bona fide transmembrane signaling). CpxA is known as a sensor for diverse signals[38]. Here, we chose three relatively well-characterized signals, namely pH, K+, and periplasmic protein CpxP to assess the signaling receptor activity of CpxA^QTY in vitro. To rule out the possibility that these signals regulate the activity of CpxA^QTY by interacting with its cytoplasmic portion rather than its periplasmic sensor domain, we prepared the cytoplasmic portion of CpxA (CpxAc) as the "signal-blind" control group. Here we did not include the water-soluble transmembrane domain, considering the rarity of signal-sensing by SHK transmembrane domains and the difficulty of restoring the original conformational constraints by the sensor domain on the transmembrane domain using a designed linker.

Mild alkaline pH stimulates the Cpx pathway[39], directly via the periplasmic sensor domain of CpxA, leading to an up-regulation of its autokinase activity[33]. We measured the autokinase activity of CpxA^QTY under different pH conditions, ranging from 7.0 to 9.0. The results showed that the phosphorylation level of CpxA^QTY significantly increased as pH value rose up to 8.5, with no further increase at pH 9.0 (Fig. 4a). The response of CpxA^QTY to mild alkaline pH was in good agreement with that of CpxA in previous study[33,39]. In the case of CpxA[33], compared to pH 7.0, the phosphorylation rate was increased by approximately 2.5-fold at pH 7.5 and by about 4-fold at pH 8.0; as for CpxA^QTY, compared to pH 7.0, the phosphorylation rate was increased by approximately 2.7-fold at pH 7.5 and by about 7.5-fold at pH 8.0. In contrast, the "signal-blind" group CpxAc showed no notable difference of the phosphorylation level across the tested pH range (Fig. 4a). Therefore, we verified that CpxA^QTY responded to pH variations in a highly similar manner to CpxA by signaling through the designed transmembrane domain, though with great variation in sequence (58%).

In vitro data have previously demonstrated that CpxP inhibits the autokinase activity of CpxA[33]. Consistent with the effect on CpxA, CpxP was observed to inhibit the autokinase activity of CpxA^QTY (Fig. 4b), in a concentration-dependent manner. The autophosphorylation rate of CpxA[33] was reported to be inhibited by approximately half when equimolar concentration of CpxP was added, while CpxA^QTY requires approximately 10-fold CpxP concentrations. This indicates that the sensitivity of CpxA^QTY to CpxP tends to be lower than that of CpxA. Interestingly, CpxAc exhibits a sensitivity to CpxP similar to that of CpxA^QTY.

The other signal, K+, reported as the stimulus on the autokinase activity of CpxA[33], strongly activated the autokinase activity of CpxA^QTY, but also that of CpxAc (Fig. 4c). The approximate response

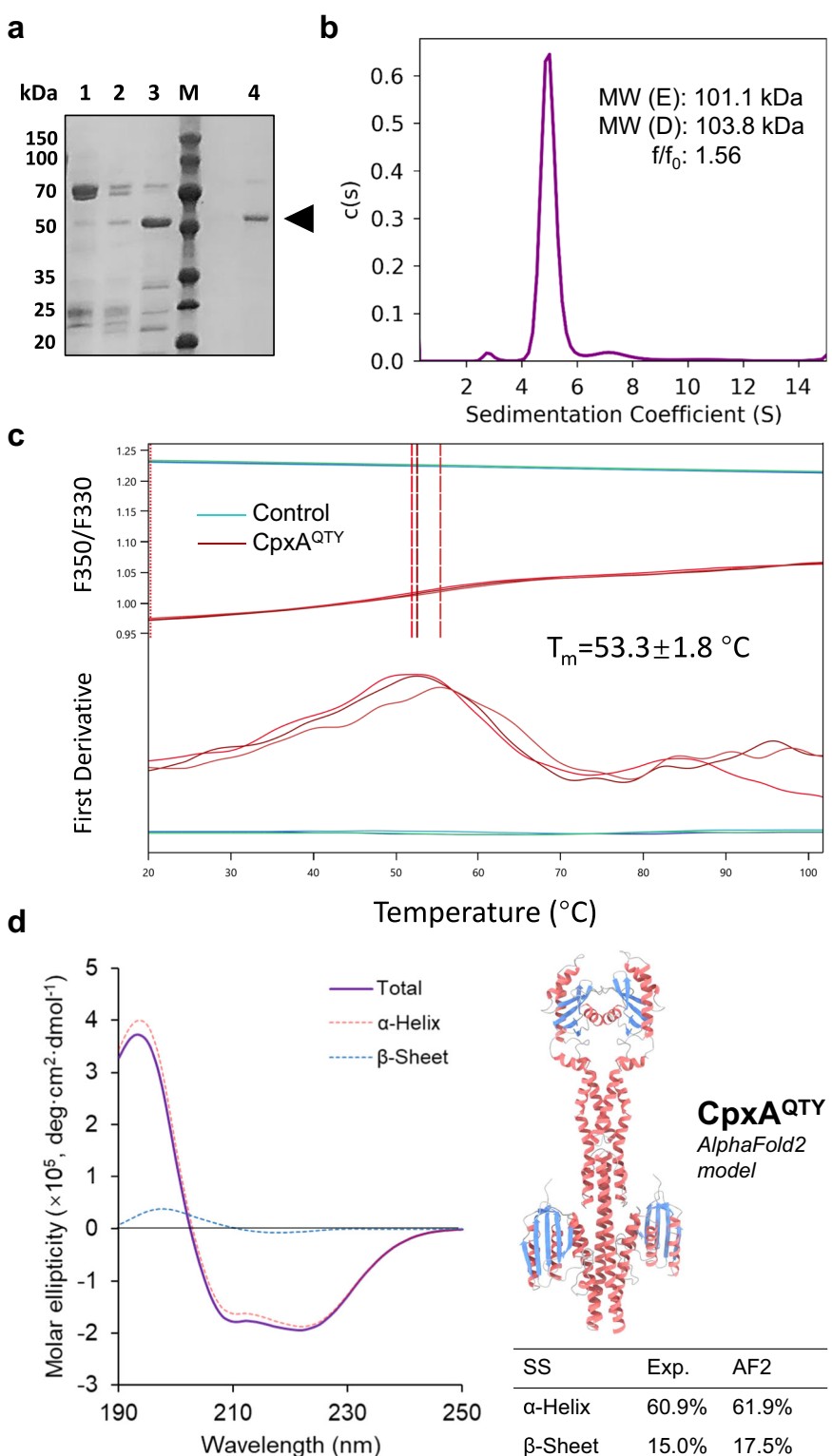

**Fig. 2 | Purification and biophysical characterizations of CpxA^QTY. a** Sodium dodecyl-sulfate polyacrylamide gel electrophoresis (SDS–PAGE) image of purification of CpxA^QTY. Lane M, molecular weight markers; lane 1/2/3, the fractions of IMAC washed/eluted off by 50/80/300 mM imidazole, 25 mM Tris·HCl, 300 mM NaCl, 5% glycerol, pH = 8.0; lane 4, the collected fraction corresponding to the major peak of gel filtration, in 25 mM Tris·HCl, 150 mM NaCl, 50 mM arginine, 5% glycerol, pH = 7.5. The experiment was repeated three times independently with similar results and the representative result was shown. **b** Sedimentation velocity AUC data of CpxA^QTY. The fitted continuous c(s) distribution curve was shown.

E, experimental; D, designed. **c** Thermostability of CpxA^QTY measured by nanoDSF. Technical triplicates of CpxA^QTY (red lines) were assayed and the fitted $T_m$ values were indicated by red dashed lines. The control group (green lines) was set as only buffer, no proteins added. **d** Circular dichroism (CD) results of CpxA^QTY. CD spectra (Total, purple solid line) and its α-helix (red) and β-sheet (blue) component spectra. The secondary structure analysis was performed using JASCO multivariate secondary structure (SS) estimation program. The content of α-helix and β-sheet in the AlphaFold2 (AF2) model were determined using PyMOL. Source data are provided as a Source Data file.

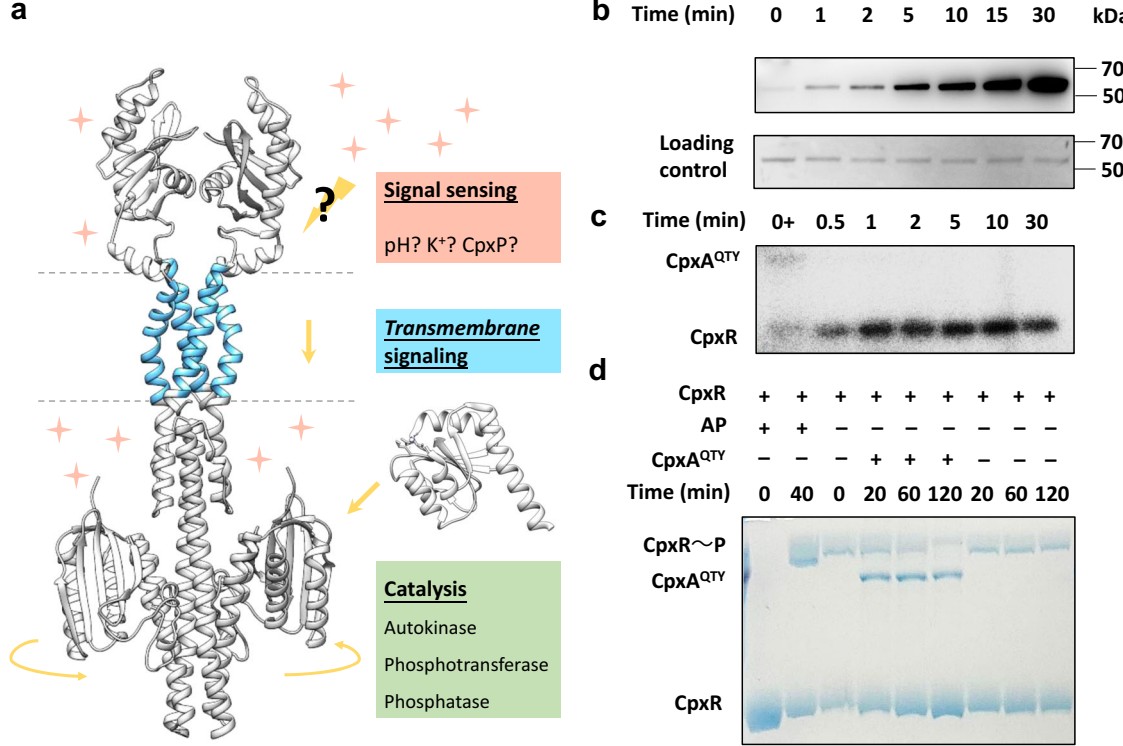

**Fig. 3 | Functional characterization of CpxA$^{QTY}$. a** Schematic diagram showing the activities of CpxA$^{QTY}$ (left). Phosphotransferase activity and phosphatase activity involve CpxR (right). The predicted structure of CpxA$^{QTY}$ by AlphaFold2 was shown. The transmembrane region was shown in cyan. Only the receiver domain of CpxR was shown. **b** Time-course of CpxA$^{QTY}$ autophosphorylation. CpxA$^{QTY}$ (1 μM) were phosphorylated with 1 mM ATP at room temperature (RT). **c** CpxA$^{QTY}$ has phosphotransferase activity. After the autophosphorylation of 1 μM CpxA$^{QTY}$ with 40 μM [γ−$^{32}$P]-ATP (1.2 Ci/mmol) for 30 min, 4 μM CpxR was added to initiate the phosphotransfer. "0 + " means that the sample was taken after the addition of CpxR at $t$ = 0. **d** CpxA$^{QTY}$ has phosphatase activity, shown by Phos-tag SDS-PAGE. CpxR was phosphorylated by acetyl phosphate (AP) for 40 min, and then AP was removed by desalting. The dephosphorylation was initiated by adding CpxA$^{QTY}$ (1:10 molar ratio)/equivalent buffer and 1 mM ADP. Phosphorylated CpxR (CpxR-P) moved slower than unphosphorylated one. For (**b**–**d**), each experiment was repeated three times independently with similar results, and the representative result was shown. Source data are provided as a Source Data file.

range for CpxA$^{QTY}$ was 10–100 mM, and for CpxAc, it was 5–50 mM. CpxA in liposomes showed a minor response to 0.5 mM KCl and a major response to 500 mM[33]. It appears that CpxA$^{QTY}$ is less sensitive to K$^+$ compared to CpxA.

Therefore, we speculated that these two signals might regulate the autokinase activity of CpxA by interacting with both the extracytoplasmic sensor domain and the cytoplasmic portion, or only by interacting with cytoplasmic portion. Nevertheless, all these results indicated that CpxA$^{QTY}$ remained sensitive to the signals, to which CpxA responds, although the sensitivity to the signals seems to be altered to varying degrees.

Our results showed that, in spite of the great sequence variation of the transmembrane domain and the lipid-free milieu, the functional integrity of CpxA$^{QTY}$ was still preserved, including *transmembrane signaling* receptor activity.

## An extensive and dynamic H-bond network in the transmembrane domain

We ask how the successful balance of solubility, stability and activity is achieved in the transmembrane domain of CpxA$^{QTY}$. Despite the absence of an experimental structure of the transmembrane domain, the powerful structure prediction tool AlphaFold2[36] offers us an alternative approach to visualize the overall interior interaction landscape. We carried out all-atom Molecular Dynamics (MD) simulations in explicit water, with the AlphaFold2 model as the input structure. Although we focused on the transmembrane domain, the extracellular sensor domain and intracellular HAMP domain were preserved to maintain appropriate constraints and mimic the native structural

context. Interestingly, the sensor domain in this model showed an untypical dimerization (Supplementary Discussion). The predicted local distance difference test (pLDDT) and predicted alignment error (PAE) test showed a favorable prediction quality of the transmembrane domain (Supplementary Fig. 2a). By comparing the transmembrane domain with that in the wildtype CpxA AlphaFold2 model, which showed a very high prediction quality (Supplementary Fig. 2b), we found a good agreement of the overall structure (RMSD = 1.333 Å) and residue positions (Supplementary Fig. 2d, e).

During the 100-ns simulation, the transmembrane domain showed favorable structural stability, with backbone RMSD versus medoid below 2 Å for the majority of the simulation time (1.88 Å on average) (Supplementary Fig. 3a, b); the sensor domain and HAMP domain showed similar stability (Supplementary Fig. 3c–f). In the transmembrane domain, the root-mean-square fluctuation (RMSF) of most residues was around 1 Å, but that of the periplasm-facing region reached 2 Å, even 3 Å (Supplementary Fig. 3g), indicating the high local flexibility. Multiple previous structural analyses of SHKs and homologous receptors have indicated the presence of a flexible region on one side of the transmembrane domain, which was inferred or demonstrated to be important for transmembrane signaling[20,21,40–42].

To scrutinize the interactions stabilizing the transmembrane helices, we analyzed the hydrogen bond (H-bond) formation during the simulation. We observed an extensive H-bond network inside the helical bundle across the simulation time (Fig. 5a), similar as the cases in the QTY variants of chemokine receptors[14]. These H-bonds were formed by the polar side chains (mainly introduced by QTY design)

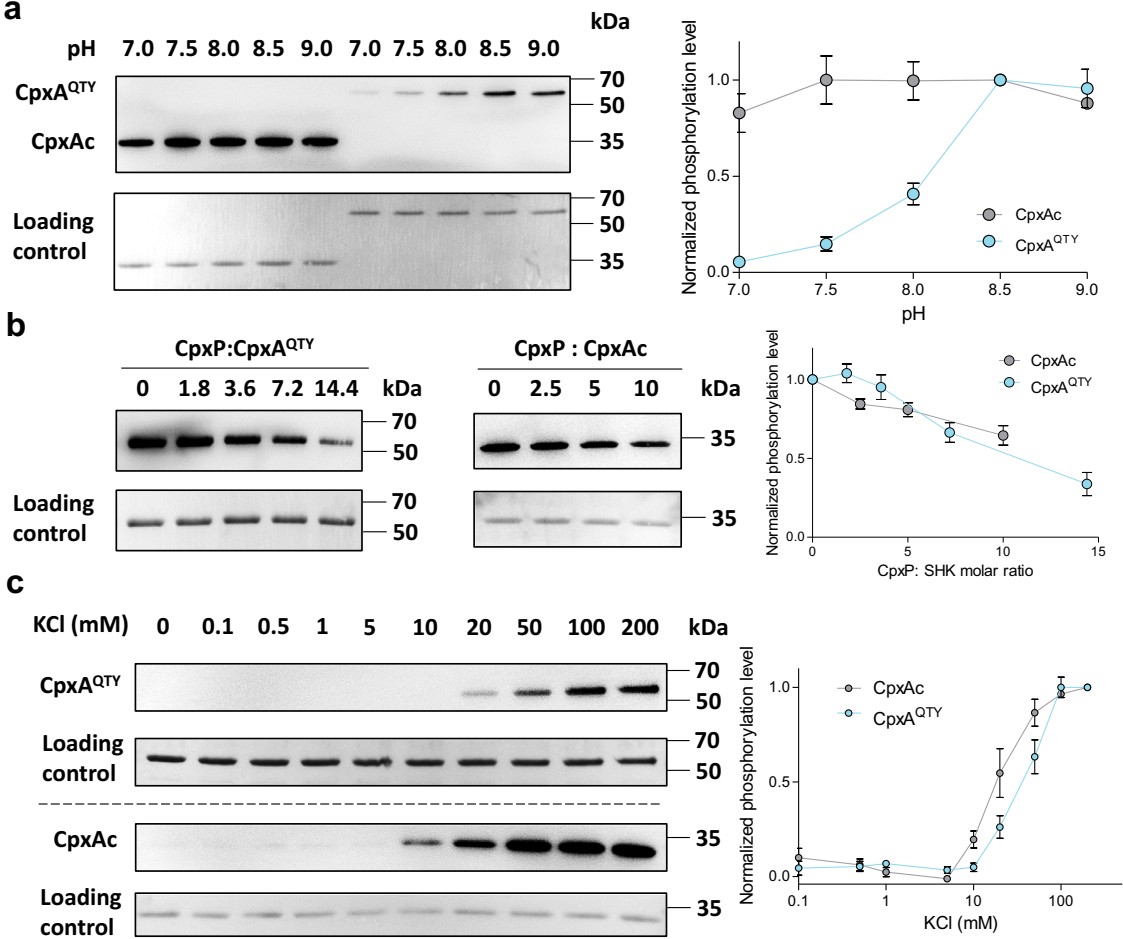

**Fig. 4 | Signal-sensing activities of CpxA$^{QTY}$. a** CpxA$^{QTY}$ can sense pH variation. CpxA$^{QTY}$ (1 μM) or CpxAc (1 μM) were phosphorylated with 1 mM ATP at RT for 2 min, with indicated pH. The phosphorylation level at pH 8.5 was set as 100% for normalization. **b** CpxA$^{QTY}$ can sense CpxP. CpxA$^{QTY}$ (1 μM) or CpxAc (1 μM) were phosphorylated with 1 mM ATP at RT for 2 min, with different concentrations of CpxP. The phosphorylation level with no CpxP addition was set as 100% for normalization. **c** CpxA$^{QTY}$ can sense K$^+$. CpxA$^{QTY}$ (1 μM) or CpxAc (1 μM) were phosphorylated with 1 mM ATP at RT for 2 min, with different concentration of K$^+$. The phosphorylation level with 200 mM KCl was set as 100% for normalization. For all three experiments, representative anti-pHis western blot results were shown (left). The quantified phosphorylation level of the three independent experiments was shown (right). Data are shown as mean ± SEM. Source data are provided as a Source Data file.

with side chains, main chains and water molecules (Fig. 5b–d), spread uniformly and throughout the transmembrane domain.

To show how well the side chains introduced by QTY design engage in H-bond formation, we calculated the number of the H-bonds formed by each QTY side chain (i.e., side chain introduced by QTY design) in each frame and defined "average H-bond saturation", the average formed H-bond number during the simulation divided by the theoretical maximum H-bond number ($N_{max}$). For Gln (Q), $N_{max}$ equals four, because Q side chain can form four H-bonds, namely, two donors from the −NH2 group, and two acceptors from the =O group; and for Thr (T) or Tyr (Y), $N_{max}$ equals three[1]. We found that most of QTY side chains engaged in H-bond formation well (Fig. 5e). Even some highly-buried residues (e.g., T9, Q17 and Q182) engaged in H-bond formation as excellently as exposed ones (Fig. 5e, f). It is worth noting that Q outperformed T and Y significantly on H-bond formation, most with average H-bond saturation value over 80%, in spite of a higher $N_{max}$.

Furthermore, we observed significant dynamics of this H-bond network. By categorizing the H-bonds into different stability levels according to their occupancy, we found some buried QTY side chains also form mobile H-bonds as actively as the exposed ones (Fig. 5e, f), especially Q in the abovementioned periplasm-facing region (e.g., Q27, Q167, and Q168), in line with the observed flexibility (Supplementary

Fig. 3g). Moreover, we found that there were always a significant number of water molecules in the interior of the helical bundle during the simulation (Supplementary Fig. 4), mediating the interactions of polar residues. For example, there was a water molecule forming H-bonds with Q17 of both chains for most time of the simulation, though the water molecule was not always the same one and the side chains of the two residues fluctuated and adopted different conformations during the simulation (Supplementary Fig. 5). Although these H-bonds were formed by mobile water molecules, these water-mediated interactions were highly stable, contributing remarkably to the interhelical dynamic stabilization. Interestingly, we did find a highly stable interaction mediated by the same water molecule (Fig. 5d): it formed stable H-bonds with Q182 of both chains and Y13 of Chain A, with occupancy 94.2%, 77.1%, and 94.8% respectively, indicating that this water molecule might provide robust support for interhelical stabilization.

From these results of MD simulations of CpxA$^{QTY}$ AlphaFold2 structure, we reasoned that extensive H-bond formation might account for the stabilization of the transmembrane domain of CpxA$^{QTY}$. Moreover, the dynamics of the H-bond network provided appropriate structural flexibility, largely contributed by QTY side chains and water molecules.

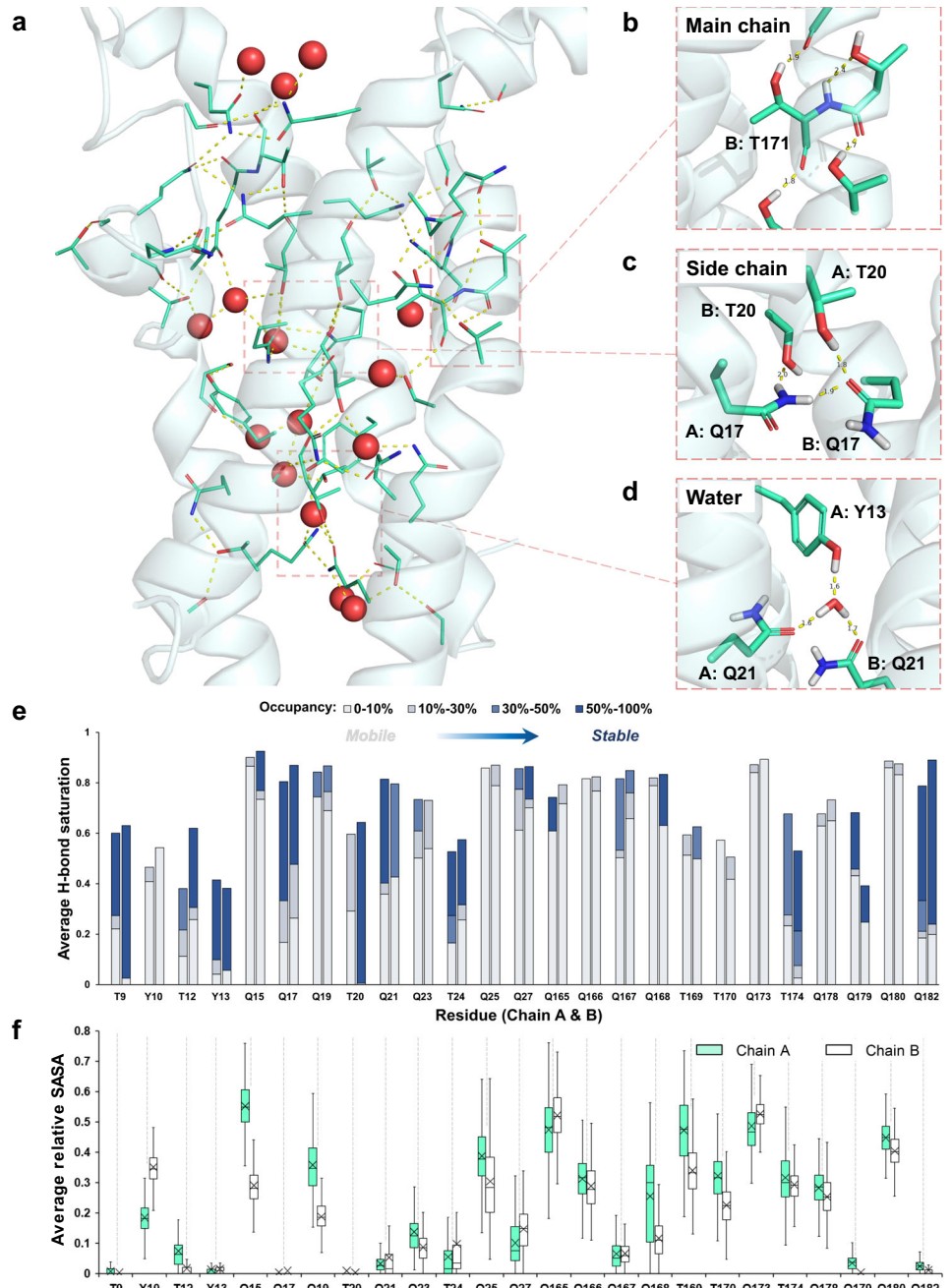

**Fig. 5 | MD simulations of CpxA$^{QTY}$ identified an extensive and dynamic hydrogen-bond network stabilizing the transmembrane domain. a** The H-bond network inside the transmembrane helical bundle. The backbone is shown as transparent cartoon, the polar side chains and heavy atoms that are engaged in H-bond formation are shown as sticks and water molecules are shown as spheres. Hydrogen atoms are hidden for simplicity. Some representative H-bonds formed with main chains (**b**), side chains (**c**) and water (**d**). Yellow dashed lines indicate H-bonds. "B: T171" indicates T171 in Chain B. The unit of the labeled distance is Å. For the coloring of the elements, carbon atoms are in green, oxygen in red, nitrogen in blue and hydrogen in white. The snapshots were from the 90-ns simulation frame, classified into the largest structure cluster. **e** Average H-bond formation of QTY side chains in the simulation. "Average H-bond saturation" was defined as the average H-bond number divided by the theoretical maximum H-bond number (N$_{max}$). For Q, N$_{max}$ = 4; T/Y, N$_{max}$ = 3. The stacking columns

indicate the proportions of the H-bonds with different levels of stability. The stability of H-bonds is classified into four levels according to the occupancy: 0–10% (in light gray), 10%–30% (in gray), 30–50% (in blue) and 50–100% (in deep blue). For each residue, the H-bond number is the sum of the occupancy of the H-bonds. The left columns denote the residues in Chain A, and right denote Chain B. **f** Average relative solvent-accessible surface area (SASA) of QTY side chains in the simulation, indicating how exposed or buried the individual residue is. Gray dashed lines are used to link the relative SASA values to the Average H-bond saturation values of the same residues. Lower and upper box boundaries 25th and 75th percentiles, respectively; line inside box indicates median, lower and upper error lines 10th and 90th percentiles, respectively; "X" indicates the mean. The outliers are not shown for simplicity, considering the large sample size (*n* = 1001). The detailed data underlying (**e**, **f**) and a version of (**f**) with outliers shown are available in Source data. Source data are provided as a Source Data file.

## Identification of scissoring motion in the transmembrane domain

To investigate if and how this extensive and dynamic H-bond network plays a role in the *transmembrane* signaling receptor activity of CpxA$^{QTY}$,

we performed Parallel Bias MetaDynamics (PBMetaD) simulation[43] to enhance sampling of the conformations of the transmembrane helices. Diagonal scissoring, i.e., two opposing helices move inward and concomitantly the other two opposing helices move outward, is one of the

most conserved conformational changes in SHK signaling[22], as observed in multiple structural studies of SHK signaling[20–22]. Therefore, we defined two collective variables (CVs) to represent the distance between TM1 helices (D1), and TM2 helices (D2) (Fig. 6a).

Convergence of the simulation was reached within around 1.5 µs (Supplementary Fig. 6a, b). Through clustering analysis of the conformations, we identified three major conformational clusters during the simulation (Supplementary Fig. 6c). Further analyses of the CV trajectories (Fig. 6b) and free energy profiles (Supplementary Fig. 6d, e) revealed that these three clusters corresponded to two stable states and one intermediate state. In State-1, TM1 helices were close, while TM2 helices were distant from each other; in State-2, the situation is reversed. During the transition from State-1 to State-2, TM2 helices converges with each other while TM1 helices moves outward, exhibiting a highly consistent diagonal scissoring motion (Fig. 6c). An intermediate state (State-M), where D1 and D2 were comparable, was observed, which might facilitate the transition between the conformationally distinct State-1 and State-2. We noted that the conformational changes in the periplasm-facing region were more pronounced than those in the cytoplasm-facing region (Supplementary Fig. 7a), consistent with previous structural studies of SHK signaling[20–22]. Furthermore, as shown in the free energy profiles, State-2 tended to be more stable. Previous studies[20,22] have indicated that the conformations similar to State-2 corresponded to an activated state. We inferred that since apo-CpxA was in the activated state[33], State-2 might correspond to the activated state.

Then we scrutinized the transformation of the interaction network during the transition between the two stable states (Fig. 6d and Supplementary Fig. 7b–e). We found that the original interaction network was virtually completely disrupted and a new interaction network formed. However, we still observed the presence of extensive H-bond networks in both states, as observed in the classical MD simulations. Similar to the classical MD simulations, numerous water molecules actively engaged in the formation of H-bond networks and mediated interactions between residues. We also observed that QTY side chains, particularly Q, flexibly formed H-bonds during the conformational change. We take the example of the top layer of the interhelical H-bond network (Fig. 6d), which showed the most significant conformational changes: some QTY side chains contributed to interhelical stabilization in both states (e.g., A: 168); some QTY side chains were fully exposed to the solvent in one state, contributing to water-solubility, while actively engaged in the interior H-bond network in the other state (e.g., A: Q21, A: Q23), resulted from the helical rotation, also common in SHK signaling[18,20,22], accompanying the scissoring motion. This demonstrates that under the coordination of water molecules, QTY residues flexibly adapt to conformational changes and aid the formation of a new interaction network to achieve a new stable state.

These results provide support for the signaling receptor activity of CpxA$^{QTY}$ and suggest the important roles of water molecules and QTY residues.

## CpxA shows similar structural dynamics and conformational changes

We ask if and how the transmembrane domains of wild-type CpxA and CpxA$^{QTY}$ are similar in terms of structural dynamics and signaling mechanism. We conducted classical MD simulations and PBmetaD simulations of CpxA. Similar to CpxA$^{QTY}$ simulations, the AlphaFold2 model was input without the catalytic domain, yet in a phospholipid bilayer.

In the 100-ns classical MD simulations, the transmembrane domain of CpxA showed similar overall structural stability (RMSD = 1.63 Å on average) (Supplementary Fig. 8a) and similar high local flexibility (Supplementary Fig. 8b) in the periplasm-facing region, compared to those of CpxA$^{QTY}$. The periplasmic-facing region exhibited significantly higher RMSF, consistent with the

trend seen in CpxA$^{QTY}$, suggesting their similar dynamics. Since the predominant content of hydrophobic residues in the transmembrane domain, especially the periplasm-facing region, we infer that the flexibility might be attributed to the loose packing. The role of the flexibility contributed by loose packing in transmembrane signaling has been discussed in multiple studies[20,40,44,45]. By comparing interactions in CpxA and CpxA$^{QTY}$, we found many H-bonds formed by QTY side chains mimicked the packing of the original hydrophobic residues in CpxA (Fig. 7), indicating the structural preservation by QTY design.

In the PBmetaD simulations, convergence of the simulation was reached within around 2.0 µs (Supplementary Fig. 9a, b). Similar to CpxA$^{QTY}$, we identified three major conformational clusters, which correspond to two stable states and one intermediate state (Supplementary Fig. 9c–f). The conformational transition between the two stable states also corresponds to diagonal scissoring motion (Supplementary Fig. 9g). By comparing the interhelical interactions with those in CpxA$^{QTY}$, we found that many hydrophobic residues went through similar transition of roles in the conformation compared to those corresponding QTY residues. For example (Supplementary Fig. 9h), both A: L21 of CpxA and A: Q21 of CpxA$^{QTY}$ are proximal to the helical axis and make pivotal contributions to interhelical stabilization in State-1, yet they are both distant from the helical axis in State-2; both A: L168 of CpxA and A: Q168 of CpxA$^{QTY}$ are distant from the helical axis and mainly interact with adjacent helices in State-1, whereas they both actively stabilize the helical core and closely interact with the opposite helices in State-2. Additionally, by comparing the overall conformations, we have also identified some differences. For example, the scissoring motion of CpxA$^{QTY}$ is more pronounced than that of CpxA; in State-1/M, the crossing angle between the TM1 helices of CpxA$^{QTY}$ is larger than that of CpxA (Supplementary Fig. 10). These differences might be relevant to the functional performance discrepancies observed previously.

The consistent performance of CpxA$^{QTY}$ and CpxA in both MD simulations and PBmetaD simulations further supports that CpxA$^{QTY}$ shares similar structural characteristics and conformational changes with wild-type CpxA.

## Reversing QTY design abolishes the signaling activity

To further study the role of the H-bond network in *transmembrane* signaling, we designed a variant, CpxA$^{QTY}$-v.2 (Supplementary Fig. 11a), with weakened QTY design. Among the residues introduced by QTY code, we selected the residues that were basically buried and replaced them back with the original hydrophobic residues. Through the same protocol of preparation, we obtained CpxA$^{QTY}$-v.2, also predominantly as dimers shown by SEC (Supplementary Fig. 11b). However, CpxA$^{QTY}$-v.2 lost the *transmembrane* signaling receptor activity, showing no response to pH variation (Supplementary Fig. 11c).

We generated the structure model of CpxA$^{QTY}$-v.2 by AlphaFold2, which showed high prediction quality in the transmembrane domain and a similar structure to CpxA$^{QTY}$ (Supplementary Fig. 2c, f, g). We performed classical MD and PBMetaD simulations, with the same protocol and nearly identical settings as CpxA$^{QTY}$. The MD simulation showed that the flexibility observed in the periplasm-facing region of CpxA$^{QTY}$ was absent in that of CpxA$^{QTY}$-v.2 (Supplementary Fig. 12a). We also observed the absence of an interhelical H-bond network and few water molecules inside the helical bundle (Supplementary Fig. 12b–e). The PBMetaD simulation showed no significant two-state pattern (Supplementary Fig. 13).

The simple restoration of the original hydrophobic core from the polar core of the transmembrane helical bundle resulted in the loss of the *transmembrane* signaling receptor activity, underscoring the crucial role of the H-bond network in *transmembrane* signaling of CpxA$^{QTY}$.

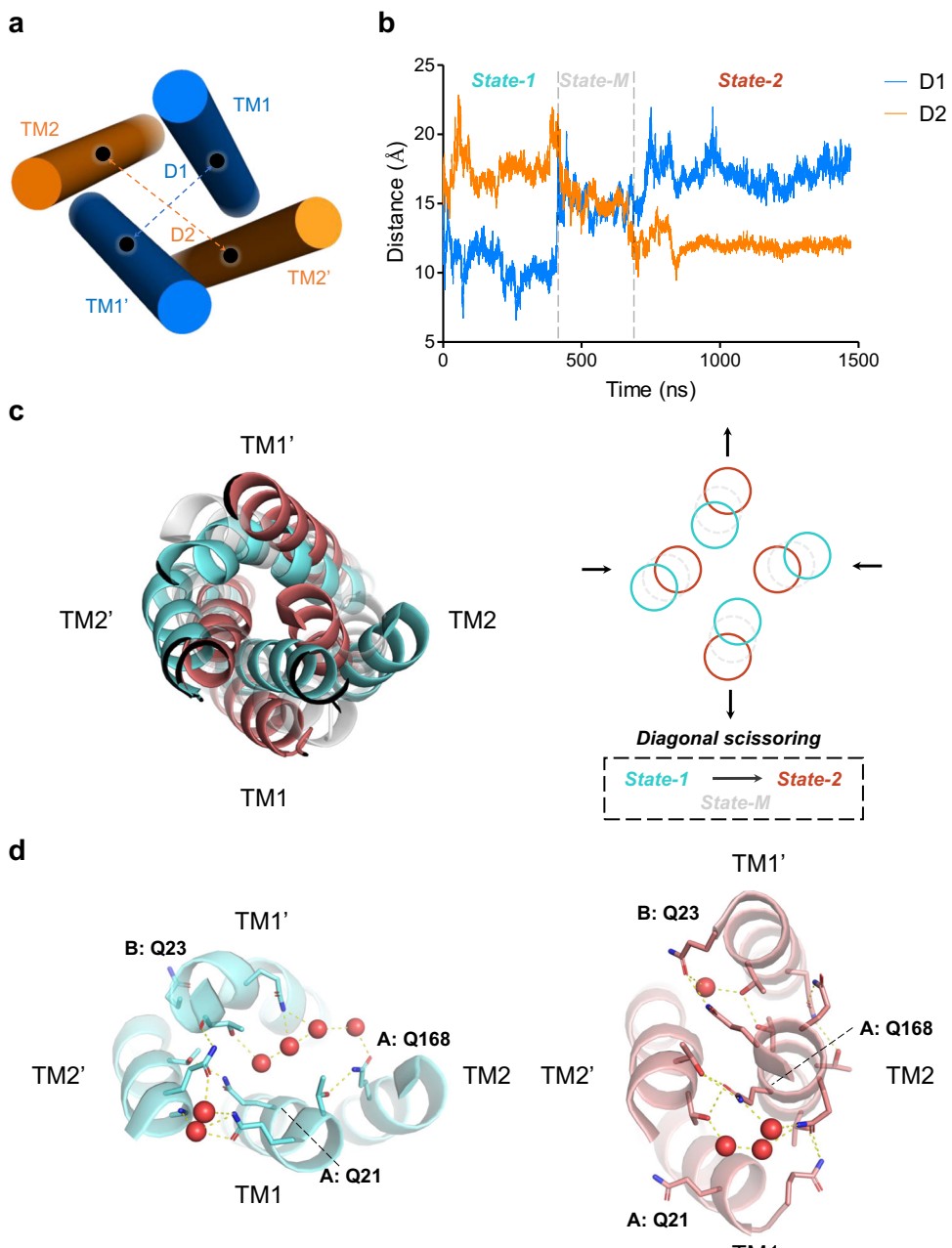

**Fig. 6 | PBMetaD simulation of CpxA$^{QTY}$ identified a common pattern of conformational changes in SHK transmembrane signaling. a** Schematic diagram of collective variable (CV) definition. The TM1 (blue) and TM2 (orange) helices were shown as simple cylinders. The center of mass of the helices were shown as black dots. The CVs, i.e., D1 and D2 were indicated by the dashed lines. **b** The curves of the CVs with simulation time. The patterns of the CV variation were classified into three states, separated by gray dashed lines. **c** The conformational transition of the different states of the transmembrane domain. (Left) Top view of the conformations of State-1 (cyan), State-M (transparent gray), and State-2 (red), as viewed from the periplasm looking into the cytoplasm. (Right) Schematic diagram of the conformational transition from State-1 to State-2. The movement is typical diagonal scissoring, which means two opposing helices move inward and concomitantly the other two opposing helices move outward. The helices were simplified as the circles. "TM1" indicates TM1 helix in Chain A and "TM1'" indicates TM1 helix in Chain B. **d** Snapshots of the top layer of the interhelical H-bond network of State-1 (left) and State-2 (right). The display style is the same as Fig. 5a. The structural snapshots of **c** and **d** were from the medoid frames of the clusters, corresponding to Supplementary Fig. 6c. Source data are provided as a Source Data file.

## Point mutations diminish the signaling activity

We attempted to "turn off" the signaling activity by mutagenesis, based on modeling and simulation results, to further experimentally validate the consistency of the general signaling mechanism between CpxA$^{QTY}$ and CpxA, i.e., sensing pH by the sensor domain and transducing the signal through the transmembrane domain. Four residues in the transmembrane domain were individually mutated back to their original hydrophobic counterparts, namely Q17L, T20V, Q21L, and Q182L. These residues played critical roles in the H-bond network of CpxA$^{QTY}$

in the simulations (Fig. 5e). We found that all these mutants exhibited diminished (Q17L, T20V, and Q182L) or abolished sensitivity (Q21L) to pH variation (Supplementary Fig. 14). Furthermore, two basic residues in the sensor domain were mutated to alanine individually. We inferred that these two residues might be involved in the perception of alkaline pH, as they are both located at the domain center and exposed to the solvent according to the AlphaFold2 model (Supplementary Fig. 15a). It was observed that R99A no longer responds to pH variation, while R106A exhibits reduced pH sensitivity (Supplementary Fig. 15b, c).

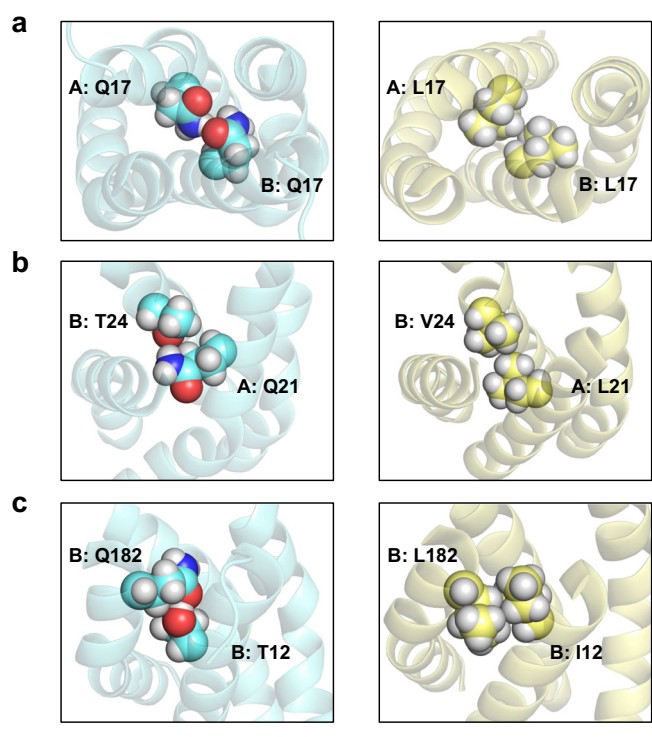

**Fig. 7 | Interactions of polar residues in CpxA^QTY mimicked the packing of the original hydrophobic residues in CpxA, revealed by MD simulations. a** (Left) Q17 in Chain A and Q17 in Chain B of CpxA^QTY. (Right) L17 in Chain A and L17 in Chain B of CpxA. **b** (Left) Q21 in Chain A and T24 in Chain B of CpxA^QTY. (Right) L21 in Chain A and V24 in Chain B of CpxA. **c** (Left) T12 in Chain B and Q182 in Chain B of CpxA^QTY. (Right) I12 in Chain B and L182 in Chain B of CpxA. Transmembrane regions are shown as cyan cartoon in CpxA^QTY simulation (left) and yellow cartoon in CpxA (right). Highlighted residues are shown as spheres. The structural snapshots were from the simulation medoid frames.

The point mutations in the transmembrane domain and sensor domain diminished the signaling receptor activity of CpxA^QTY, further supporting that the general transmembrane signaling mechanism of CpxA is well-conserved by QTY design.

## Discussion

Membrane proteins have long been of great interest but remain challenging to study[46–48]. Water-solubilization designs circumvent the difficulties faced by the traditional system and provide insights into the structural and functional principles of membrane proteins[1]. Although previous water-solubilization designs managed to show their preservation of structural characteristics and some specific activities[2,6,9], the inability of retaining the intrinsic function of the transmembrane domain and the reliance of structure inputs greatly limited the application of the water-solubilization design.

Our work provides the first evidence that water-solubilization design is capable of retaining the integrity of molecular function, including the transmembrane domain. By combining the advantages of QTY code and SHKs, we designed CpxA^QTY, the completely water-soluble full-length transmembrane receptor capable of *transmembrane* signaling, which holds promise for unveiling the molecular mechanism of transmembrane signaling by SHKs at the full-length level. Notably, the surprising preservation of *transmembrane* signaling receptor activity with 58% transmembrane domain variation rate underlines the robustness of the signaling mechanism. In addition, we investigated the principles of the balance of solubility, stability and activity, providing insights into the water-solubilization design.

Stability-activity trade-off of proteins is common[49,50]. Most of the previous water-solubilization designs were based on the energic

parameters and almost only focus on exposed residues[2,9,10], aiming to render membrane proteins close to typical water-soluble proteins. However, when the overall environment changes from lipids to water, the force condition of the hydrophobic core could be altered dramatically[51,52], which might be fatal for the appropriate conformational changes essential to the activity of the transmembrane domain. Such strategy appears to be only capable of balancing solubility and stability, thus showing very limited activity. By contrast, aiming to render them close to "themselves", QTY design replaces both the exposed and buried hydrophobic residues in transmembrane domain with polar residues. This strategy tends to sacrifice some stability but allows the dynamic stabilization and appropriate flexibility of the transmembrane domain. CpxA^QTY showed remarkably preserved *transmembrane* signaling receptor activity, while CpxA^QTY-v.2 lost this activity, after we merely mutated the buried residues back to the original hydrophobic ones. Although this design structurally resembles typical water-soluble proteins, the loss of the function of the transmembrane domain indicates the stability-activity trade-off. Another advantage of this strategy, i.e., mimicking themselves and substituting both interior and exterior residues, is its compatibility to conformational changes essential to function. Some QTY residues only contribute to water-solubility in one state but directly engage in interhelical stabilization in the other state (Fig. 6d), aiding the formation of the new conformation, which is challenging for other water-solubilization designs. In addition, one of the core elements of QTY code is the striking similarity of amino acid residue "shapes" despite their distinct chemical differences, hydrophilic vs hydrophobic side chains[13] (Fig. 1a). We observed that at many sites hydrophobic packing were replaced by inter-side-chain H-bonds, two interaction patterns closely resembling each other (Fig. 7). This resemblance provided the basic support for the preservation of structural stability and activity. These results underscore the balancing of solubility, stability and activity by QTY water-solubilization design.

The importance of structural flexibility in transmembrane signaling has been intensively studied in different classes of signaling receptors[53,54]. We have seen a helical bundle dynamically stabilized by an extensive H-bond network in simulations of CpxA^QTY, showing significant flexibility (Fig. 5e and Supplementary Fig. 3). By contrast, CpxA^QTY-v.2, incapable of *transmembrane* signaling, showed significantly reduced flexibility and absence of scissoring motion in simulations. The flexibility of the transmembrane helices has been reported to have functional significance in different SHKs[19]. The flexibility was observed in the transmembrane helices of NarQ, the only SHK by which transmembrane signaling mechanism has been structurally unveiled, and the transmembrane domain tends to be simply accommodating and transmitting the conformational changes[20]. QTY design seems to fit this mechanism well, as the flexibility determined by the nature of QTY design allows the interaction rearrangement, mediated by QTY residues and water molecules in a concerted manner, as we observed in the transition between the two conformational states of CpxA^QTY. Gln (single letter = Q) features its strong polarity and elongated side chain, dictating its excellent solubility and flexibility[55,56]—exploring different conformations to maximize H-bond formation. Therefore, the high content of Gln, determined by the fact that Leu contributes the most to the composition of transmembrane α-helices[57], in QTY-designed transmembrane domain might contribute greatly to H-bond formation. In fact, it was indeed observed that Gln performed very well in H-bond formation, in both stable and mobile manners. In addition, water molecules inside transmembrane helix bundles could further increase the diversity of chemical groups available to carry out function[52]. Designed H-bonds between side chains being displaced by ordered water molecules were observed in the crystal structure of a designed helical bundle, without affecting the accurate design of the backbone[26]. We observed that the mobile water molecules, flexibly forming H-bonds, considerably facilitated the

interactions of the buried polar side chains, smoothening the contacts of residues like "lubricant". During the conformational changes, the water molecules actively engaged in the rearrangement of the H-bond network. These findings underscore the important role of water molecules in the stabilization and dynamics of CpxA$^{QTY}$. In summary, the structural flexibility introduced by QTY design could play an important role in *transmembrane* signaling of CpxA$^{QTY}$.

Despite the remarkable preservation of the molecular function by QTY design, we still noticed some discrepancies in functional performance between CpxA$^{QTY}$ in water and CpxA in lipid environment, by comparing our results with data from previous publications (Supplementary Table 1). Considering the significant sequence variation, the structure of the transmembrane domain may be still slightly affected, which is indeed observed in the simulations (Supplementary Fig. 10). Moreover, interactions with the lipid bilayer are one of the key factors influencing the activity of membrane proteins[1]. Therefore, the discrepancies could be attributed to some alterations in activity caused by QTY design-induced structural perturbations, the removal of the lipid environment, or potential subtle differences in experimental setups, which requires further investigation.

On the road of removing the boundary between membrane proteins and water-soluble proteins, our work made an encouraging progress. It is foreseeable that water-solubilized membrane proteins with intact function hold promise for a broad spectrum of applications, such as biosensing, structural biology, synthetic biology, drug discovery, and more. This work could provide guidance for future water-solubilization design and functional design of membrane proteins.

## Methods

### Design of CpxA$^{QTY}$
The input sequences of CpxA were from Uniprot database (Entry: P0AE82). QTY design and the corresponding property prediction were done automatically by QTY enabling server (http://pss.sjtu.edu.cn)[4]. The default setting was applied: simple design mode was used and TM data from Uniprot was used.

### Molecular cloning
The sequence of CpxA were codon-optimized for *E. coli* expression and synthesized (Azenta Life Sciences). The sequences of CpxR and CpxP were obtained by cloning from *E. coli* K-12 strain. The sequences of CpxAc were obtained by cloning from the plasmid expressing CpxA$^{QTY}$. All genes were cloned into pET28a (+) expression vector and the plasmids were transformed into *E. coli* BL21(DE3) strain. Strep-tag II were constructed by site-directed mutagenesis using a modified QuickChange protocol. The sequences of the primers used in this study are summarized in Supplementary Table S2.

### Protein purification
Proteins used for function-related characterization were purified by immobilized metal affinity chromatography (IMAC) followed by gel filtration. *E. coli* cultures were grown at 37 °C until the OD600 reached 0.4–0.8, after which isopropyl-β-D-thiogalactoside (Sangon Biotech) was added to a final concentration of 0.2 mM followed by 16 h expression at 16 °C. Harvested cells were disrupted by pressure homogenization (ATS Scientific) at 800 bar in the binding buffer (25 mM Tris-HCl, pH 8.0, 300 mM NaCl, 20 mM imidazole, and 5% glycerol) with addition of 0.1% DNase I (Smart-Lifesciences), 1 mM PMSF and 5 mM 2-mercaptoethanol. After centrifugation, the supernatant was loaded onto an Ni-NTA gravity column (Qiagen) (2 mL) equilibrated with the binding buffer. After washes with the same buffer, the protein was then eluted with a 20-50-80-300 mM imidazole gradient. The contents of the samples from IMAC were visualized by SDS-PAGE. The fractions containing the target protein were pooled and concentrated to 1 mL and further purified and desalted on Superdex 200 (Cytiva Life Sciences) size-exclusion chromatography using an ÄKTA Purifier system (Cytiva Life Sciences) with the running/storage buffer (25 mM Tris-HCl, 150 mM NaCl, 50 mM arginine and 5% glycerol, pH = 7.5). The contents of all fractions were visualized by SDS-PAGE. The peak fractions corresponding to the protein of interest and showing high purity were collected, concentrated, and flash-frozen for long-term storage at −80 °C.

Proteins used for biophysical characterization were purified by a three-step protocol, with insertion of a Strep-tag II purification step prior to gel filtration. After IMAC, the fractions containing the target protein were loaded onto a column, loaded with 2 mL STarm Strep-tactin Beads 4FF (Smart-Lifesciences), equilibrated with the eluting buffer (25 mM Tris-HCl, pH 8.0, 300 mM NaCl, 300 mM imidazole and 5% glycerol). After repeating the loading 5 times, the protein was washed with the storage buffer and then eluted by the Strep-tag eluting buffer (the storage buffer with addition of 5 mM D-biotin).

### Analytical ultracentrifugation
The sedimentation-velocity experiment was conducted at 20 °C in a Beckman-Optima AUC analytical ultracentrifuge using an An-60 Ti rotor (Beckman-Coulter). The sample was freshly prepared CpxA$^{QTY}$ (with Strep-tag II) in PBS, 50 mM arginine, pH = 7.5. The A280 of the sample was 0.8 before being loaded to AUC. Concentration profiles were measured at 280 nm at a speed of 40,000 rpm, followed by evaluation using the SEDFIT software[58] and GUSSI[59] software to fit the single-ideal species model and obtain the diffusion-corrected sedimentation coefficient distributions (c(s)-distributions).

### Thermostability measurement by nanoDSF
The nanoDSF experiment was carried out using a Prometheus NT.48 instrument (NanoTemper Technologies). The sample was freshly prepared CpxA$^{QTY}$ (with Strep-tag II) in 25 mM HEPES, 150 mM NaCl, 5% glycerol, pH = 7.5, with a concentration of 10 µg mL$^{-1}$. The control group was set as only buffer, no proteins added. For each loading, 10 µL of the sample per capillary was prepared. The temperature gradient was set to increase 1 °C per min in a range from 20 °C to 100 °C. The data analysis was performed in the software PR.ThermoControl (NanoTemper Technologies).

### Circular dichroism
The circular dichroism (CD) experiment was carried out using a JASCO J-1500 spectropolarimeter (JASCO) in a 0.1 cm path-length quartz cuvette (Starna Scientific). The sample was freshly prepared CpxA$^{QTY}$ (with Strep-tag II) in 25 mM NaH$_2$PO$_4$/Na$_2$HPO$_4$, 150 mM NaF, 5% glycerol, pH = 7.4, with a concentration of 0.1 mg mL$^{-1}$. The baseline was assayed using only buffer. Spectra of the sample and baseline were recorded at 25 °C by averaging three scans from 260 to 176 nm (1 nm bandwidth) in 0.2-nm steps at a scanning rate of 50 nm min$^{-1}$. Spectra from 190 to 250 nm were subjected to secondary structure analysis using JASCO multivariate secondary structure estimation program (JASCO), with the molar ellipticity calibration model applied.

### Phosphorylation and dephosphorylation assays
All reactions were conducted at RT. For autophosphorylation, the kinase (at indicated concentration) was incubated in phosphorylation buffer (50 mM Tris-HCl, 5% glycerol, 50 mM KCl, 20 mM MgCl$_2$, pH = 7.5, if no specific indication) for 10 min before initiating the reaction by addition of ATP (Promega) with indicated concentration. At different time points, aliquots were removed and mixed with SDS-PAGE loading buffer. Then, anti-pHis western blot was performed as reported[60] with some modifications. Briefly, after SDS-PAGE, the proteins were electroblotted onto a PVDF membrane (Bio-Rad Laboratories) at 24 V for 10 min by a semi-dry method (Bio-Rad Laboratories). The membrane was blocked with 3% BSA in TBST for 1 h at RT and then after washing with TBST, incubated with anti-pHis antibody

(Anti-N3-Phosphohistidine Antibody, clone SC39-6, Sigma-Aldrich) diluted 1:5000 in TBST with 3% BSA overnight at 4 °C. After washing with TBST, the membrane was then incubated with goat anti-rabbit IgG-HRP conjugate (Absin, abs20040) diluted 1:5000 in TBST with 3% BSA for 1 h at RT, followed by washing with TBST. The membrane was incubated with ECL chemiluminescence solution (Bio-Rad Laboratories) for 1 min at RT and chemiluminescence from the membrane was imaged using the imager Tanon 4600 (Tanon Science & Technology). The membrane was washed with water and stained with Coomassie blue for the visualization of loading control. For phosphotransfer, two methods were used. For anti-pHis western blot, after the autophosphorylation of 1 μM CpxA$^{QTY}$ with 1 mM ATP for 30 min, 4 μM CpxR was added to initiate the phosphotransfer, followed by the same protocol of abovementioned anti-pHis western blot. For autoradiography, after the autophosphorylation of 1 μM CpxA$^{QTY}$ with 40 μM [γ-$^{32}$P]-ATP (PerkinElmer, 1.2 Ci/mmol) for 30 min, 4 μM CpxR was added to initiate the phosphotransfer. At different time points, aliquots were removed and mixed with SDS-PAGE loading buffer. After SDS-PAGE, the gel was then dried, and incubated in the exposure cassette for 6 h at RT. After development, the autoradiogram was scanned by Typhoon IP phosphor Imaging FLA Scanner (GE Healthcare). For dephosphorylation, 10 μM CpxR was phosphorylated by 20 mM acetyl phosphate[61] (AP) in phosphorylation buffer for 40 min, and then AP was removed by quick desalting (GE Healthcare). The dephosphorylation was initiated by adding 1 μM CpxA$^{QTY}$ or equivalent buffer and 1 mM ADP. At different time points, aliquots were removed and mixed with SDS-PAGE loading buffer. Phosphorylated proteins were separated by Phos-tag acrylamide gel electrophoresis. Polyacrylamide gels were polymerized with 50 mM Phos-tag acrylamide (Wako) and 50 μM ZnCl$_2$. Following electrophoresis, gels were stained with Coomassie blue.

## Phosphorylation site identification by mass spectrometry

After the autophosphorylation reaction, SDS-PAGE was run and the gel was stained by Coomassie brilliant blue. The expected band was excised and dehydrated for 10 min in acetonitrile. The gel fragment was digested with trypsin/chymotrypsin for 16 h at 37 °C. Digested peptides were extracted with 50% and 80% acetonitrile respectively. The extractions were combined and dried in a vacuum centrifuge at 40 °C. The dried peptides were dissolved in 0.1% formic acid and then were analyzed with EasynanoLC1000 coupled with an Orbitrap Q-Exactive Plus Mass spectrometer (Thermo Fisher Scientific). A 15 cm × 50 μm reverse-phase column (2 μm, C18, 120 Å, Thermo Fisher Scientific) was utilized for the separation of peptides. The mobile phases were Buffer A (0.1% formic acid in water) and Buffer B (80% acetonitrile with 0.1% formic acid). The gradient was: 2–20% Buffer B in 98 min, 20–30% in 10 min, 30–95% in 2 min, and then kept at 95% for 8 min, 95–2% in 1 min, and sustained for 1 min at 2%. The flow rate was 300 nL per min. The MS scan was ranged from 350 to 1500 m/z with 70,000 resolution. For MS/MS, the scan was ranged from 200 to 2000 m/z with 17,500 resolution. The LC/MS software Data Analysis 4.0 (Bruker Compass software) was used to process the raw files. To enhance the identification of phosphopeptides, all MS/MS samples were analyzed using MASCOT 2.3. The search was configured to search a database of annotated protein sequences from the *E. coli* database on SwissProt. The following search criteria were used: trypsin digestion; fixed modification of carbamidomethylation (Cys); variable modifications of oxidation (M), phospho (ST), phosphor (H), and phospho (Y); and allowance of two missed cleavages. The precursor ion and fragment maximum mass deviation were 0.4 Da (monoisotopic) and 0.6 Da (monoisotopic), respectively. The peptide-spectrum matches (PSMs) were filtered based on the score threshold of a 1% false discovery rate (FDR) using the formula: FDR = 2[nDecoy/(nDecoy + nTarget)]. To confirm the identification of phosphorylated peptides, all fragmentation spectra were manually evaluated and identified using the neutral loss of ion MH + −80. Additionally, the localization probabilities for phosphorylation sites were calculated using Protein Discovery software (Thermo Fisher Scientific) and the posttranslational modification score algorithm with PhosphoRS analysis. Phosphorylation sites with a probability of >0.95 were considered as identified phosphorylation sites.

## Structure prediction by AlphaFold2

The structure prediction was performed using AlphaFold2 via the ColabFold[62] pipeline applying mostly default parameters. Structure relaxation was applied; no template was applied; MMSeq2 was used for MSA; the maximal number of recycles was set as 12. The predicted structures with the highest model rank were analyzed and used as the input structure of classical MD and PBMetaD simulations. The pLDDT (for assessing local structure confidence) and PAE (for assessing complex structure confidence) plots in Supplementary Fig. 2 were generated by ColabFold. The content of α-helix and β-sheet in the AlphaFold2 model were determined using PyMOL (version 2.5 Schrödinger, LLC).

## MD simulations and analysis

The simulations were performed using GROMACS 2021.3/2022.3[63]. Truncated versions of the AlphaFold2 models of CpxA$^{QTY}$, CpxA$^{QTY}$ -v.2 and CpxA (residues 1 to 245, homodimer, without the catalytic domain) were used as the input structures.

For CpxA$^{QTY}$ and CpxA$^{QTY}$ -v.2, the topology file was created by GROMACS engine. The protein was centered in a cubic box, with a distance of 1.0 nm from the box edge. The classical AMBER99SB-ILDN force field[64] and TIP3P water model were selected. The system energy was minimized using the steepest descent method and the maximum forces were converged below 1000 kJ mol$^{-1}$ nm$^{-1}$. Electrostatics were treated with Particle Mesh Ewald, and the cutoff for both Coulomb and van der Waals interactions was 1.0 nm. 2 fs time step were used. The modified Berendsen thermostat was used, with system coupled to a 300 K bath. The Parrinello-Rahman barostat was used with isotropic coupling to 1 bar. All bonds were constrained using LINCS algorithm. The NVT and NPT equilibriums were run for 100 ps. Following a pre-production run of 10 ns, the production run of 100 ns was performed, with the same parameters in the equilibration stage.

For CpxA, the topology file was created by CHARMM-GUI membrane builder[65]. The protein in POPC bilayer was centered in a cubic box, with a distance of 1.0 nm from the box edge. The ff14SB[66] plus LIPID21 Amber force field and TIP3P water model were selected. The system energy was minimized using the steepest descent method and the maximum forces were converged below 1000 kJ mol$^{-1}$ nm$^{-1}$. Electrostatics were treated with Particle Mesh Ewald, and the cutoff for both Coulomb and van der Waals interactions was 1.0 nm. Then, 125 ps equilibration simulations were performed using the standard six-step CHARMM-GUI protocol[67]. The Berendsen thermostat was used, with system coupled to a 300 K bath. All bonds were constrained using LINCS algorithm. The Berendsen barostat was used with semi-isotropic coupling to 1 bar. For the production run of 100 ns, the Nose-Hoover thermostat was used, with system coupled to a 300 K bath. The Parrinello-Rahman barostat was used with semi-isotropic coupling to 1 bar. Other parameters are in line with those in the equilibration stage.

For analysis, we mainly concentrated on the transmembrane four-helix bundle (residues 8–29 and 164–184, both chains) and selected this region for following analyses. Simulation frames every 100 ps were extracted. Simulation frames were clustered (GROMACS built-in command "gmx cluster") with backbone RMSD of the transmembrane domain using the linkage method (cutoff = 0.08 for CpxA$^{QTY}$ and CpxA$^{QTY}$ -v.2; cutoff = 0.055 for CpxA) and the medoid structures were exported. The medoid of the largest cluster was selected as the medoid of the simulation. RMSD and RMSF analyses were conducted by

superimposing the domain of interest. SASA analysis was conducted by command "gmx sasa".

We analyze the H-bonds by GROMACS built-in command "gmx hbond". The side chains of all QTY residues were grouped and as the analysis subject. For the calculation of the formed H-bond numbers of individual residues, the side chain of each QTY residue was grouped as "group 1", of which the complementary set was grouped as "group 2". To determine if a hydrogen bond exists, the default geometrical criterion of GROMACS was used: the distance between the acceptor and donor heavy atoms was less than 3.5 Å and the angle between the donor proton, the donor heavy atom, and the acceptor oxygen was less than 30°.

Structure figures in this article were prepared using PyMOL and Chimera[68].

### PBMetaD simulations and analysis

The PBMetaD simulations were performed in GROMACS 2022.3 with the PLUMED 2.8.1 plugin[69]. The input structures were the same as those in classical MD simulations. For CpxA$^{QTY}$ and CpxA$^{QTY}$-v.2, the topology file was created by AmberTools Leap program[70]. The protein was centered in a cubic box, with a distance of 1.2 nm from the box edge. The ff14SB Amber force field[66] and TIP3P water model were selected. The system energy was minimized using the steepest descent method and the maximum forces were converged below 1000 kJ mol$^{-1}$ nm$^{-1}$. Electrostatics were treated with Particle Mesh Ewald, and the cutoff for both Coulomb and van der Waals interactions was 1.2 nm. 2 fs time step were used. The modified Berendsen thermostat was used, with system coupled to a 300 K bath. For pressure coupling, C-rescale barostat was used with isotropic coupling to 1 bar. H-bonds were constrained using LINCS algorithm. The NVT and NPT equilibriums were run for 100 ps. For CpxA, the topology file and the equilibration process were the same as that in classical MD simulations. After the NPT equilibrium, the outputs were used for PBMetaD simulations.

We used a well-tempered Parallel Bias MetaDynamics method[43] to efficiently sample the free energy landscapes. We defined two collective variables (CVs) representing the distance between the two TM1 helices (D1), and between the two TM2 helices (D2). "D1" was defined using the center of mass of residue 6−30 and "D2" was defined using the center of mass of residue 164−189; for CpxA, "D1" includes that of residue 17−27 and "D2" includes that 164−174. Gaussian potentials were added every 500 steps with the height of 0.3 kJ mol$^{-1}$, and the bias factor was set to 8. The width of the Gaussian hills was 0.03 nm. The lower/upper bound for the grid was 0.1 nm/2.3 nm for D1 and 0.5 nm/2.8 nm for D2. The number of bins for the grid was 150. We defined a pair of walls for the CVs, which limit the region of the phase space accessible during the simulation. The lower/upper bound for the wall was 0.2 nm/2.0 nm for D1 and 0.6 nm/2.5 nm for D2. The force constant for the wall was 150 for CpxA$^{QTY}$, 300 for CpxA$^{QTY}$-v.2, and 500 for CpxA. The simulations were run until the convergence was reached. The total simulation time is ~1500 ns for CpxA$^{QTY}$, ~800 ns for CpxA$^{QTY}$-v.2, and 2000 ns for CpxA. Convergence was assessed by aligning the free energy profile and calculating its RMSD every 50 ns for CpxA and CpxA$^{QTY}$, and every 25 ns for CpxA$^{QTY}$-v.2. The trajectories were analyzed using PyMOL.

For conformational analysis, we only analyzed the transmembrane four-helix bundle. Simulation frames were extracted to ~5000 frames. Simulation frames were clustered with backbone RMSD of the transmembrane domain using linkage method (cutoff = 0.1 for CpxA$^{QTY}$ and 0.062 for CpxA) and the medoid frames were exported. The interhelical H-bond network were analyzed using PyMOL.

The stable states were local free energy minimum defined by the location where the first derivative of the free energy profile equals zero with the second derivative greater than zero, for both D1 and D2.

### Reporting summary

Further information on research design is available in the Nature Portfolio Reporting Summary linked to this article.

## Data availability

The simulation data generated in this study have been deposited in the Zenodo database under accession code 10516203. All other data generated in this study are provided in the Supplementary Information/Source Data file. The CpxA sequence used in this study are available in the Uniprot database under accession P0AE82. Source data are provided with this paper.

## Code availability

Input files for simulations, and example scripts for running or analyzing simulations were are publicly available at https://doi.org/10.5281/zenodo.10516203.

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

## Acknowledgements

This study is supported by the grant from National Key R&D Program of China (2018YFA0903600) and National Natural Science Foundation of China (32170105). The computations in this paper were run on the π 2.0 supercomputer supported by the Center for High Performance Computing at Shanghai Jiao Tong University (SJTU) and on the MIT SuperCloud (Lincoln Laboratory Supercomputing Center, Massachusetts Institute of Technology). We thank Prof. Shaoqing Zhang (Shanghai Institute of Biochemistry and Cell Biology), Prof. Yilei Zhao (SJTU), Xingyu Ouyang (SJTU), Huahua Song (Core Facility of Basic Medical Sciences, SJTU), Hehua Liu (Fudan University) and Jingli Hou (SJTU) for assistance or advice on this work.

## Author contributions

F.T., S.Z., M.L., and Y.W. designed the experiments. M.L., Y.W., J.L., R.W., S.L., and L.M. performed the experiments. M.L., F.T., S.Z., H.T., R.Q., and Y.W. analyzed the data. M.L. and F.T. wrote the manuscript. M.L., F.T., S.Z., P.X., H.T., R.Q., and Y.W. revised the manuscript. F.T., S.Z., M.L., H.T., and P.X. conceived the project. F.T., P.X., and H.T. provided the funding.

## Competing interests

S.Z. is a member of board director of 511 Therapeutics that generates therapeutic monoclonal antibodies against solute carrier transporters. He is also a scientific advisor for OH2 Laboratories that works on generating therapeutic monoclonal antibodies against GPCRs. However, this study does not involve in GPCRs and solute carrier transporters. OH2 Laboratories licensed the QTY code technology from MIT. However, this article does not study GPCRs. S.Z. is the inventor of the QTY code and has a minor equity of OH2 Laboratories and majority equity in 511 Therapeutics shares that works on solute carrier transporters. S.Z. is also a scientific advisor for 3DMatrix Co Ltd, that commercializes self-assembling peptide hydrogels for surgical and accelerated wound-healing applications. MIT filed several patent applications for the QTY code for GPCRs and glucose transporters. The current study does not involve in GPCRs and glucose transporters. S.Z. is a co-founder and board director of Molecular Frontiers Foundation that encourages young people to ask good questions about science and nature and also organizes Molecular Frontiers Symposia around the world. There is no compensation for the activities. The remaining authors declare no competing interests.
