## [Peer Review File · Nature Communications]

Design of a water-soluble transmembrane receptor kinase
with intact molecular function by QTY codeReviewer #1 (Remarks to the Author):

The manuscript

"Design of a water-soluble transmembrane receptor kinase with intact molecular function by QTY code"

- The work is comprehensive and attempts to answer the following:

Hypothesis: using the QTY method to create a water-soluble membrane protein could fully retain their intrinsic functions.

AGREEMENT:

- Histidine kinase CpxAQTY is completely water-soluble.
- The designed CpxAQTY exhibits expected biophysical properties and fully preserved native molecular function, including: i) autokinase activity, ii) phosphotransferase activity, iii) phosphatase activity.

DISAGREEMENT:

- The designed CpxAQTY exhibits expected biophysical properties and fully preserved native molecular function, iv) signaling receptor activity, involving a water-solubilized transmembrane domain.

This point is not well addressed beyond the *in silico* work. The authors analyze H-bond networks in CpxAQTY but that is a water-soluble protein, very different than a hydrophobic environment like the cell membrane (lipids, cholesterol, etc.). Hydrophobic interactions occur because of hydrogen bonding between water molecules around the hydrophobe. When a hydrophobe is dropped in an aqueous medium, hydrogen bonds between water molecules will be broken to make room for the hydrophobe. However, water molecules do not react with hydrophobe. The QTY method creates new H-bonding and stability to the once transmembrane domain. As an QTY-effect, the *in silico* work is a nice addition to the discussion and serves as evidence of the properties of the method in transforming hydrophobic proteins/domains into water soluble.

AGREEMENT:

Additionally, we probed the principles underlying successful balance of structural stability and activity in the water-solubilized transmembrane domain.

DISAGREEMENT:

Through molecular dynamics simulations, we highlight important roles of an extensive and dynamic hydrogen-bond network and its flexibility introduced by QTY code.

The authors equate this network as principles of stability but also of signal transduction. The work is *in silico*, missing structural data for comparison. Unfortunately, membrane proteins structural data is scarce. Similarly, activity of these enzymes *in vitro*. A simple search in PubMed for CpxA/CpxR shows only 175 articles. This finding indicates the system might not be well characterized *in vivo* or *in vitro*.

DISAGREEMENT:

Our successful functional preservation of the transmembrane domain by QTY code takes water-solubilization design to the next level.

The authors have already published other articles using QTY, covering among other membrane proteins GPCRs.

1 Structural bioinformatics studies of bacterial outer membrane beta-barrel transporters and their AlphaFold2 predicted water-soluble QTY variants.

Sajeev-Sheeja A, Smorodina E, Zhang S.

PLoS One. 2023 Aug 22;18(8):e0290360. doi: 10.1371/journal.pone.0290360. eCollection 2023.

PMID: 37607179 Free PMC article.

2 Scalable biomimetic sensing system with membrane receptor dual-monolayer probe and graphene transistor arrays.

Qing R, Xue M, Zhao J, Wu L, Breitwieser A, Smorodina E, Schubert T, Azzellino G, Jin D, Kong J, Palacios T, Sleytr UB, Zhang S.

Sci Adv. 2023 Jul 21;9(29):eadf1402. doi: 10.1126/sciadv.adf1402. Epub 2023 Jul 21.

3 Reverse-QTY code design of active human serum albumin self-assembled amphiphilic nanoparticles for effective anti-tumor drug doxorubicin release in mice.

Meng R, Hao S, Sun C, Hou Z, Hou Y, Wang L, Deng P, Deng J, Yang Y, Xia H, Wang B, Qing R, Zhang S.

Proc Natl Acad Sci U S A. 2023 May 23;120(21):e2220173120. doi: 10.1073/pnas.2220173120. Epub 2023 May 15.

PMID: 37186820 Free PMC article.

4 Bioinformatics-aided Protein Sequence Analysis and Engineering.

Zhang W, Wang T.

Curr Protein Pept Sci. 2023;24(6):477-487. doi: 10.2174/1389203724666230509124300.

PMID: 37287293 Review.

5 Structural informatic study of determined and AlphaFold2 predicted molecular structures of 13 human solute carrier transporters and their water-soluble QTY variants. Smorodina E, Diankin I, Tao F, Qing R, Yang S, Zhang S. Sci Rep. 2022 Nov 22;12(1):20103. doi: 10.1038/s41598-022-23764-y.

PMID: 36418372 Free PMC article.

6 Protein Design: From the Aspect of Water Solubility and Stability.

Qing R, Hao S, Smorodina E, Jin D, Zalevsky A, Zhang S.

Chem Rev. 2022 Sep 28;122(18):14085-14179. doi: 10.1021/acs.chemrev.1c00757. Epub 2022 Aug 3.

PMID: 35921495 Free PMC article. Review.

7 Hiding in plain sight: three chemically distinct α -helix types.

Zhang S, Egli M.

Q Rev Biophys. 2022 Jun 20;55:e7. doi: 10.1017/S0033583522000063.

PMID: 35722863 Review.

8 Comparing 2 crystal structures and 12 AlphaFold2-predicted human membrane glucose transporters and their water-soluble glutamine, threonine and tyrosine variants. Smorodina E, Tao F, Qing R, Jin D, Yang S, Zhang S. QRB Discov. 2022 Jun 13;3:e5. doi: 10.1017/qrd.2022.6. eCollection 2022.

PMID: 37529287 Free PMC article.

9 Enabling QTY Server for Designing Water-Soluble α -Helical Transmembrane Proteins.

Tao F, Tang H, Zhang S, Li M, Xu P. mBio. 2022 Feb 22;13(1):e0360421. doi: 10.1128/mbio.03604-21. Epub 2022 Jan 18.

PMID: 35038913 Free PMC article.

10 Comparing Native Crystal Structures and AlphaFold2 Predicted Water-Soluble G Protein-Coupled Receptor QTY Variants.

Skuhersky MA, Tao F, Qing R, Smorodina E, Jin D, Zhang S.

Life (Basel). 2021 Nov 24;11(12):1285. doi: 10.3390/life11121285.

PMID: 34947816 Free PMC article.

Other items and suggestions:

1. The work is heavy in computational analysis and models of the structure of CpxA by AlphaFold2. This goes back to a previous comment about Our successful functional preservation of the transmembrane domain by QTY code takes water-solubilization design to the next level. CpxA activity examined has no point of comparison either in the manuscript or from the literature.
2. Early in the introduction it will be interesting to hear the limitations of detergents, then move to the benefits to water-soluble methods.
3. Line 63 and 64 seem to conflict with previous work from the authors using other membrane proteins, in active conformations. If this is true, then the statement negates the need for this article, and its impact.
4. The article does not describe the “trial and error” of the QTY method. Is it necessary to replace ALL respective residues in the method or does the algorithm suggest different alternatives/permutations of the mutagenesis?
5. Line 81, the term “biomimetic sensing” is used. Would this imply that the protein be prepared on a membrane or substrate? For that native lipids or detergents, etc. can be used on the native protein. It is not clear to me how QTY will aid in this application.
6. Line 131, use of “without detergent” I suggest “detergent-free.” It was used by the authors once before.
7. Line 131 and 134, introduce the result of DIMER formation of the CpxAQTY. Examining the crystal structure of *E. coli* CpxA amino acids 187-457, for example PDB ID 4BIU, we do observe a dimer. If this is a crystallography dimer and not a true dimer that could expand on your inference that CpxAQTY is a dimer. Also, a reference for the native CpxA being a dimer will be good to have.
8. Before I forget, please provide an Accession Number for the protein synthesized, and also a full amino acid sequence to observe where the TAGs are placed.
9. Line 152, what’s the source of the antibodies used? pHIS and any others.
10. Line 177, I would like to see an introduction via references, of the expected catalytic activities of CpxA. Reference 30 seems a good starting point, hopefully not the only one. This could be a good point to compare the use of detergents and proteoliposomes against the water-soluble CpxA.
11. Line 262-279, the discussion of the water molecules do not belong in this manuscript, especially the discussion of “a water molecule,” line 270. Too much of a stretch to add to the desired conclusion. As mentioned above a hydrophobic and hydrophilic environment are very different. You could have created your CpxAQTY model, revert the mutations, add lipids and run simulations anew. Then you could compare which environments have water or need water to support stability. You could also feed those new coordinates to the Collective Variables, etc. A lot of work, perhaps for a separate article.
12. Figure 4, and its discussion; I would like part of the discussion to center around the native CpxA, from reference 30 or others.
13. Line 485, the gradient was run in the AKTA?
14. Figure 2a. Currently websites like PubPeer and others enable the scrutiny of images like Western Blot. A feature of these analysis is that the picture could have been purposely

changed. If you change Brightness/Contrast Figure 2a looks different, including the new band on Lane 4

15.

16. Figure 3b, changing Brightness/Contrast shows a sharp line on top of the bands.

REVIEWER COMMENTS

Reviewer #2 (Remarks to the Author):

In their manuscript, Li and coauthors describe reengineering of a bacterial transmembrane receptor into a soluble protein by replacing amino acids in its transmembrane region. They then characterize the protein itself and its signaling ability.

The approach itself is interesting and worth probing. If transmembrane proteins could be transformed into soluble while maintaining their ligand binding and functional properties by following a simple amino acid replacement rule, this would strongly advance the field.

However, it is unlikely that a simple replacement rule would generally work for all transmembrane proteins, especially complex multistate signaling proteins.

I believe that the data reported in this manuscript does not unambiguously demonstrate that the function of their engineered protein (CpxA-QTY) is indeed conserved. If the authors accurately performed all required experiments, the likely outcome would be that they would prove that the engineered protein is not comparable to wild type protein, invalidating the basic premise of the work. Consequently, this would be a technical report more suitable for a specialized journal.

1) Regarding CpxA-QTY:

- SEC chromatogram presented in Suppl. Fig. 1b shows some aggregation. Is this aggregation ongoing? You cannot claim that the protein is soluble if it eventually aggregates.
- are the sensor and the TM domain of CpxA-QTY folded in solution? Please provide some experimental characterization (circular dichroism, small-angle scattering, maybe electron microscopy)
- out of three functional tests, only one (pH dependence) shows that CpxA-QTY is able to recognize signals, which are not recognized by a truncated CpxAc protein. However, pH is a very generic environmental condition that affects many molecular processes, so the data does not demonstrate that pH is recognized by the sensor domain via the same mechanism as in WT CpxA

- in the KCl dependence test (Fig 4c), CpxA-QTY is responsive to KCl levels only above 10 mM, whereas WT CpxA in liposomes appears to be activated by as little as 0.5 mM KCl (Fig 3A, doi 10.1074/jbc.M605785200). This likely indicates that the signaling properties of CpxA-QTY are altered.

2) Regarding replacement of the buried amino acids in the TM part: in my view, this replacement completely changes the energetic landscape of the transmembrane domain (available conformations, energy differences between them, and energy barriers) and the resulting protein's dynamics is not in any way indicative of the dynamics of the original protein (CpxA). The manuscript currently lacks the appropriate controls (both in experimental and modeling parts) for comparing the dynamics to WT CpxA, as CpxA-QTY is only compared to CpxAc or CpxA-QTY-v.2.

3) Regarding the modeling part of the manuscript:

- the authors present an AlphaFold model of dimeric CpxA, which shows unexpected arrangement of sensor domains, because most sensor domains dimerize via alpha-helices. Consequently, authors need to validate their model, or construct (and validate) a model that conforms to the usual expectations of domain structure in sensor histidine kinases
- the authors need to compare dynamics of CpxA-QTY with that of WT CpxA in a bilayer to claim that the signaling mechanism is conserved
- RMSD data (ED Fig. 1a) shows that the structure is likely very dynamic, whereas no data on the HAMP domain and sensor domain is presented at all.

Less important issues:

- page 3 line 57: the authors might want to discuss a recent preprint about generating a soluble GPCR (doi 10.1101/2023.05.09.540044)

- page 4 lines 94-96: the mechanism remained elusive, but data generated in recent ~10 years provides more and more details. On the opposite, the data presented by the authors do not clarify anything about the mechanism of transmembrane signaling

- page 5 lines 104-107: this is a strong and unproven assumption that is very likely to be incorrect. Polar amino acids are able to form hydrogen bonds, and interactions between them would be stickier and likely to attract water molecules

- page 10 lines 223-224 while pLDDT score for the transmembrane part is okay, the pLDDT scores for the linker between the TM part and sensor is low, and the linker structure is unreliable. The authors either need to validate the linker structure in the starting model or generate a reliable model

- please include as a supplementary information the atomistic models used as starting structures, reference structures for RMSD/RMSF calculations and used for preparing the figures of the manuscript so that they can be analyzed by reviewers and readers

Fig 2c: please indicate whether these are technical or biological replicates

Fig 3a: makes an impression that all signals are recognized in the periplasm, whereas below you show that only pH is possibly recognized by the periplasmic domain

Fig 3a: please clarify in the legend that only a part of CpxR is shown

Fig 4a vs b: do they match? (at 0 CpxP and pH 7.5)

Fig 4c: please provide quantification similar to that in Fig. 4a,b

ED Fig. 1a: please also show RMSD relative to the starting model

Fig S1b: please indicate the column used for chromatography

Reviewer #3 (Remarks to the Author):

This manuscript uses a previously developed strategy for solubilizing membrane proteins (QTY code) to render signaling protein, CpxA water-soluble. A series of biophysical

experiments are used to test functional preservation of the signaling protein. Using computer-generated models and simulation, functional preservation is attributed to a dynamic hydrogen bonding network introduced to the transmembrane region of the protein by the QTY code.

This work is further demonstration of the power of QTY code for generating and studying soluble versions of membrane protein receptors.

Comments for further discussion and analysis:

1. The conclusion about signal transmission preservation in CpxAQTY is stronger than the data suggest. A more nuanced conclusion of the results in Figure 4 and signal preservation throughout the text should be provided.

Example conclusion given in the text:

“Nevertheless, all these results indicated that CpxAQTY responded to the signals in a manner highly similar to CpxA.”

While this statement is supported by the pH-dependence of autokinase activity in CpxA (Figure 4A) the results of the two subsequent experiments (4B and 4C) only allow for the conclusion that there is a difference between studying the water-soluble protein and the membrane-bound protein.

Because the control experiment demonstrates that CpxP and potassium ion are influencing the isolated cytoplasmic domain, these experiments do not rule out the possibility that the CpxAQTY data is purely a result of the cytoplasmic interaction and not the periplasmic interaction and subsequent signaling studied in reference 30. As discussed briefly in the text, the result of these experiments could point to new biology, but it is just as likely that the results describe artifacts introduced by extracting a membrane protein from its native lipid environment (i.e. the absence of a hydrophobic barrier between sections of the protein).

The conclusion should highlight the triumph of functional preservation using the QTY

method (demonstrated in Figure 4A) while also highlighting the difference between studying the water-soluble protein and the membrane-bound protein given the confounding results in Figures 4B and 4C.

2. Simulation of the CpxA AlphaFold model in a lipid membrane would enable general conclusions about QTY code and membrane protein solubilization to be made, rather than the conclusions that are currently enabled by the data, which are strictly about how a dynamic hydrogen bonding network in the CpxAQTY protein enables CpxAQTY to mimic some of the signaling function of CpxA.

If simulations of the CpxA model in a lipid membrane are included and more comparisons like those in Supplementary Figure 3 are made, this work could make more general, evidence-supported conclusions about the type of interactions necessary to move a membrane protein into water while still retaining its function. These comparison would be especially useful in the absence of structural data, which is one of the goals of protein solubilization.

Below are examples (in no particular order) from the manuscript that either limit the scope of the manuscript to a single designed protein (CpxAQTY) or make assumptions about the ways in which CpxAQTY mimics the function of CpxA, and would benefit from the simulation of CpxA in the lipid environment.

A. In the introduction there is mention of activity preservation and “the specific interaction network essential for the intrinsic biological [activity]’ in the unmodified protein.

The hydrogen bonding study suggests that these are the essential interactions for activity for CpxAQTY, but what are the essential interactions in CpxA?

B. “Another advantage of this strategy, i.e., mimicking themselves and substituting both interior and exterior residues, is its compatibility to conformational changes essential to function.”

This sentence states an assumption that the CpxAQTY protein behaves the same in water as the CpxA protein behaves in lipid. Simulation of CpxA in lipid would provide insight into conformational states explored in lipid similar to those explored in water and corroborate the assumption.

C. "Theoretically, designed interhelical polar interaction networks by QTY code may replace the interaction networks in transmembrane hydrophobic helices responsible for SHK signaling and realize signal transduction through the conformational changes that mimic the original ones."

What is the dynamic hydrogen bond network introduced by QTY code mimicking? Why is function therefore retained?

D. "In summary, the structural flexibility introduced by QTY design could play an important role in transmembrane signaling of CpxAQTY."

It would be useful to know if the dynamic nature of the water-solubilized structure is comparable to that of the membrane structure in order to draw parallels between the hydrogen bonding network and the hydrophobic interactions in the membrane. Perhaps the dynamic nature of the protein makes structural characterization challenging.

- What interactions are replacing the hydrogen bonding networks found in CpxAQTY (particularly the hydrogen bonds to water) in the CpxA protein?

- What insight do we gain into the function of CpxA to enable the use of QTY as a tool for study of other membrane proteins?

E. "The simple restoration of the original hydrophobic core from the polar core of the transmembrane helical bundle resulted in the loss of the transmembrane signaling receptor activity, underscoring the crucial role of the H-bond network in transmembrane signaling of CpxAQTY."

This statement highlights the crucial role of a dynamic structure, enabled by highly dynamic hydrogen bond network. How does this pertain to the case of membrane-bound CpxA?

F. “From these results, we reasoned that extensive H-bond formation might account for the stabilization of the transmembrane domain of CpxAQTY. Moreover, the dynamics of the H-bond network provided appropriate structural flexibility, largely contributed by QTY side chains and water molecules.”

A discussion of structural flexibility in CpxA would make this work more generalizable.

G. “we reasoned that extensive H-bond formation might account for the stabilization of the transmembrane domain of CpxAQTY”

The hydrogen bonding data from Figure 5 are relevant to how CpxAQTY is soluble, maintains a folded state predicted by AlphaFold, and suggests how it might be performing signaling function, but do not give insight into how the native protein, CpxA, behaves.

Minor comments

3. Typo in Fig 3. ‘Phosphotransfase activity’ should be phosphotransferase activity

4. From the text the “signal-blind control protein (CpxAc) does not contain the transmembrane region and it’s purely the cytoplasmic portion of the protein serving as a control. These two constructs should perform similarly as controls, but please specify in the text for clarity that CpxAc is composed solely of the cytoplasmic portion and briefly discuss why the water-soluble transmembrane region is not part of the control.

5. Please specify what is meant by “with great variation (58%)” in the following sentence: “Therefore, we verified that CpxAQTY responded to pH variations in a highly similar manner to CpxA by signaling through the designed transmembrane domain, though with great variation (58%).”

I assume this is sequence variation, but the way this sentence currently reads suggests variation in pH response.

6. The following sentence is confusing. Please break it up.

“Therefore, the high content of Gln, determined by the fact that Leu contributes the most to the composition of transmembrane α -helices, in QTY-designed transmembrane domain might contribute greatly to H-bond formation and indeed we observed that Gln performed very well in H-bond formation, in both stable and mobile manners (Fig. 5e, Extended Data Fig. 1c-j).”

It reads more clearly with a period after ‘...contribute greatly to H-bond formation’.

7. The data in supplementary Figure 3 would do well in the main text as it visually highlights the power of the QTY design approach. Please consider incorporating it into the MD simulation main text figure. See comment 2.

8. The use of italicized transmembrane to describe the CpxAQTY transmembrane domain is inconsistent (see the MD results section). Ensure consistency throughout the text for clarity.

9. Please clarify throughout the text that all of structures in this work are predicted models.

Example: “mimicked the packing of the original hydrophobic residues in CpxA indicating the structural preservation by QTY design.”

could be: “mimicked the packing of the original hydrophobic residues in the predicted structure model of CpxA indicating the structural preservation by QTY design.”

10. Please provide the superposition of the transmembrane four-helix bundle of AlphaFold2 models of CpxAQTY and CpxAQTY-v.2 along with backbone RMSD in supplementary Figure 2, as this highlights the effect of only modifying surface residues on the structure of the transmembrane region.

11. CpxAQTY phosphotransferase activity seems to be significantly faster than phosphotransferase activity of CpxA in proteoliposomes (based on reference 30), an

observation that the text briefly mentions.

While this difference could be attributed to experimental setup, this change in rate is potentially a result of protein accessibility in the absence of lipids. This observation should add to the discussion from comment 1 about the difference between studying the water-soluble protein and the membrane-bound protein.

Responses to the reviewers' comments point-by-point

We sincerely appreciate reviewers' professional and constructive comments and suggestions. We have read through comments carefully and have made corrections accordingly. Based on the instructions provided in your letter, we uploaded the file of the revised manuscript, supplemental information, and figures. Revisions in the text are highlighted in red. With regard to reviewers' comments and suggestions, we reply as follows:

Reviewer #1:

The work is comprehensive and attempts to answer the following:

Hypothesis: using the QTY method to create a water-soluble membrane protein could fully retain their intrinsic functions.

Response: We appreciate all the time and effort from the reviewer, and thank the reviewer for the professional comments and suggestions and acknowledging the quality of our manuscript. The summary of our hypothesis is accurate. Nonetheless, after careful consideration, we decided to replace 'fully' in our manuscript with 'highly'. This is because we actually emphasize functional integrity **in a qualitative manner**, rather than quantitatively; we aim to show that the **general** signaling mechanism is conserved (how the signals are sensed and transduced; what are the major conformational changes like), rather than the detailed mechanism (which are the key residues and interactions). We appreciate the reviewer's AGREEMENT & DISAGREEMENT commenting style, which greatly helped us in critically reflecting on our manuscript and understanding the comments. We thank the reviewer to carefully evaluate our manuscript and please find the detailed responses to the concerns below.

1. AGREEMENT:

- Histidine kinase CpxAQTY is completely water-soluble.*
- The designed CpxAQTY exhibits expected biophysical properties and fully preserved native molecular function, including: i) autokinase activity, ii) phosphotransferase activity, iii) phosphatase activity.*

DISAGREEMENT:

• The designed CpxA^{QTY} exhibits expected biophysical properties and fully preserved native molecular function, iv) signaling receptor activity, involving a water-solubilized transmembrane domain.

This point is not well addressed beyond the in silico work. The authors analyze H-bond networks in CpxA^{QTY} but that is a water-soluble protein, very different than a hydrophobic environment like the cell membrane (lipids, cholesterol, etc.).

Hydrophobic interactions occur because of hydrogen bonding between water molecules around the hydrophobe. When a hydrophobe is dropped in an aqueous medium, hydrogen bonds between water molecules will be broken to make room for the hydrophobe. However, water molecules do not react with hydrophobe. The QTY method creates new H-bonding and stability to the once transmembrane domain. As an QTY-effect, the in silico work is a nice addition to the discussion and serves as evidence of the properties of the method in transforming hydrophobic proteins/domains into water soluble.

Response: We thank the reviewer for the comment and raising this concern. The reviewer's theoretical analysis is very insightful. We agree that the aqueous environment and lipid environment are very different, and therefore it is unlikely that the stabilization and detailed signaling mechanism of the water-soluble CpxA^{QTY} and membrane-bound CpxA would be completely identical. Nonetheless, we do not expect that the stabilization and mechanism of CpxA^{QTY} and CpxA is identical. Our *in silico* analysis about the H-bond network mainly aims to probe the principles underlying the balance of solubility, stability and activity in CpxA^{QTY}, rather than provide insights into the stabilization or detailed signaling mechanism of CpxA, or demonstrate that the detailed mechanism between CpxA^{QTY} and CpxA is identical.

First, we acknowledge that our previous statement should be more precise. The word 'fully' may have given the impression that we were claiming the signaling properties and mechanism of CpxA^{QTY} are fully identical to those of wild-type CpxA. In fact, we intended to emphasize functional integrity **in a qualitative manner**, rather than quantitatively; we aim to show that the **general** signaling mechanism is conserved (how the signals are sensed and transduced; what are the major conformational changes like), rather than the detailed mechanism (which are the key

residues and interactions). Therefore, we have replaced ‘fully’ in our manuscript with ‘highly’. For example,

ABSTRACT: “We show that the designed CpxA^{QTY} exhibits expected biophysical properties and highly preserved native molecular function, ...”

We also agree that “signaling receptor activity involving a water-solubilized transmembrane domain was preserved” should be further addressed beyond the *in silico* work. In this revision, we experimentally validated the consistency of the general signaling mechanism between CpxA^{QTY} and CpxA, i.e. sensing pH by the sensor domain and transducing the signal through the transmembrane domain. We attempted to “turn off” the signaling activity by mutagenesis in the transmembrane domain and sensor domain based on modeling and simulation results. We selected four residues in transmembrane domain that played critical roles in the H-bond network of CpxA^{QTY} in the simulations, and two residues in sensor domain that might be involved in the perception of alkaline pH. All the mutants showed abolished or reduced sensitivity to pH variation. Below are the added contents about this part of work:

Page 18:

Point mutations diminish the signaling activity

We attempted to “turn off” the signaling activity by mutagenesis, based on modeling and simulation results, to further experimentally validate the consistency of the general signaling mechanism between CpxA^{QTY} and CpxA, i.e. sensing pH by the sensor domain and transducing the signal through the transmembrane domain. Four residues in the transmembrane domain were individually mutated back to their original hydrophobic counterparts, namely Q17L, T20V, Q21L and Q182L. These residues played critical roles in the H-bond network of CpxA^{QTY} in the simulations (Fig. 5e). We found that all these mutants exhibited diminished (Q17L, T20V and

Q182L) or abolished sensitivity (Q21L) to pH variation (Supplementary Fig. 13). Furthermore, two basic residues in the sensor domain were mutated to alanine individually. We inferred that these two residues might be involved in the perception of alkaline pH, as they are both located at the domain center and exposed to the solvent according to the AlphaFold2 model (Supplementary Fig. 14a). It was observed that R99A no longer responds to pH variation, while R106A exhibits reduced pH sensitivity (Supplementary Fig. 14b, c).

The point mutations in the transmembrane domain and sensor domain diminished the signaling receptor activity of CpxA^{QTY}, further supporting that the general transmembrane signaling mechanism of CpxA is well-conserved by QTY design.

Supplementary Figure 13 | pH-sensing activities of transmembrane domain mutants of CpxA^{QTY}. **a-d**, Results of CpxA^{QTY}-Q17L, T20V, Q21L, and Q182L, respectively. The mutants were phosphorylated with 1 mM ATP at RT for 2 min, with indicated pH. The phosphorylation level at pH8.5 was set as 100% for normalization. For all the three experiments, representative pHs western blot results were shown (left). The quantified phosphorylation level of the three independent experiments was shown (right). Data are shown as mean \pm SEM.

Supplementary Figure 14 | pH-sensing activities of sensor domain mutants of CpxA^{QTY}. a, Charged residues in the sensor domain. All charged residues are shown as sticks. Chain A is shown in green and Chain B in yellow. **b-c,** Results of CpxA^{QTY}-R99A and R106A, respectively. The mutants were phosphorylated with 1 mM ATP at RT for 2 min, with indicated pH. The phosphorylation level at pH8.5 was set as 100% for normalization. For all the three experiments, representative pHs western blot results were shown (left). The quantified phosphorylation level of the three independent experiments was shown (right). Data are shown as mean \pm SEM.

As for the detailed mechanism, we have included additional computational data in this revision, considering that it is challenging for us to provide experimental comparison in the current manuscript (as discussed in the next response). We conducted simulations of CpxA in the lipid bilayer. The consistent performance of CpxA^{QTY} and CpxA in both MD simulations and PBmetaD simulations further supports that CpxA^{QTY} shares the similar structural characteristics and conformational changes with wild-type CpxA. Please see the details in the new section, *CpxA shows similar structural dynamics and conformational changes* (Page 15).

2. AGREEMENT:

Additionally, we probed the principles underlying successful balance of structural stability and activity in the water-solubilized transmembrane domain.

DISAGREEMENT:

Through molecular dynamics simulations, we highlight important roles of an extensive and dynamic hydrogen-bond network and its flexibility introduced by QTY code.

The authors equate this network as principles of stability but also of signal transduction. The work is in silico, missing structural data for comparison.

Unfortunately, membrane proteins structural data is scarce. Similarly, activity of these enzymes in vitro. A simple search in PubMed for CpxA/CpxR shows only 175 articles. This finding indicates the system might not be well characterized in vivo or in vitro.

Response: We thank the reviewer for the professional and insightful comment.

First, we acknowledge that our previous statement may not be very appropriate. The statement that the reviewer disagreed tends to be kind of assertive. In fact, we did not intend to **equate** this network as principles; rather, according to the *in silico* results, we infer that this network might play an important role. Since this part of work is *in silico*, indeed we should use the appropriate strength for the statement. Accordingly, we have revised it as:

ABSTRACT: “Computational approaches **suggest** that an extensive and dynamic hydrogen-bond network and its flexibility introduced by QTY code **may** play an important role.”

We agree that the structural data are important for shedding light on how QTY code preserved the signaling activity. To address this issue, the direct approach would be resolving the multi-state structures of both CpxA^{QTY} and CpxA. However, it is a very challenging and complicated task, warranting numerous attempts and long-term

effort. We plan to work on this issue in a separate paper dedicated for structure determination. In the current manuscript, we have utilized structures generated by the state-of-the-art structure prediction tool AlphaFold2, which has proven highly-accurate¹⁻⁵. Our work combines QTY code and AlphaFold2, presenting an example for mechanism investigation with the reduced reliance on experimental structures. Moreover, molecular dynamics (MD) simulations serve an irreplaceable role in visualizing and analyzing molecular dynamics. Herein, we conducted a comprehensive and in-depth analysis of the dynamic H-bond network, aiming to identify some principles underlying the balance of stability and activity by QTY code. We chose to conduct simulations to compare the structural characteristics and conformational changes between different states. Through the simulations, we found similar structural characteristics and conformational changes between CpxA^{QTY} and CpxA (the new section, *CpxA shows similar structural dynamics and conformational changes*). Furthermore, our reverse-substitution results in reduced or abolished signaling activity, including entire (CpxA^{QTY}-v.2) and individual (point mutants) reverse-substitution of QTY residues that actively contribute to the H-bond network. This is the experimental evidence that supports the important roles of the H-bond network and the conserved general signaling mechanism.

Regarding CpxAR two-component system (TCS), it is one of the classical TCS models and one of the most studied TCSs. CpxAR is often discussed in the reviews about TCSs⁶⁻⁹. There are a substantial body of research involving *in vivo*¹⁰⁻¹⁴ and *in vitro*¹⁵⁻²¹ characterization of CpxAR TCS. We also searched other classical TCSs in PubMed, and only found two TCSs that showed more articles, PhoPQ (475) and EnvZ/OmpR (350), supporting that CpxAR is one of the most studied TCSs. Nonetheless, we acknowledge that overall, histidine kinases still have not been characterized very well, especially *in vitro*. As a membrane protein, their characterization is still challenging and their detailed mechanism remains elusive, which is also the issue that QTY code and other water solubilization design methods aim to tackle.

3. DISAGREEMENT:

Our successful functional preservation of the transmembrane domain by QTY code takes water solubilization design to the next level.

The authors have already published other articles using QTY, covering among other membrane proteins GPCRs.

Response: We thank the reviewer for the kind reminder. We intended to convey that, through further characterization in the current manuscript, we have demonstrated that the QTY code has shown the excellent performance that is not previously realized by previous methods, from the perspective of the development of the field of water solubilization design. Our work provides the first evidence that water-solubilization design is capable of retaining the integrity of molecular function. However, this statement seems inappropriate and potentially misleading, so we have revised it in Abstract:

ABSTRACT: “Our successful functional preservation further substantiates the robustness and comprehensiveness of QTY code.”

Other items and suggestions:

1). The work is heavy in computational analysis and models of the structure of CpxA by AlphaFold2. This goes back to a previous comment about Our successful functional preservation of the transmembrane domain by QTY code takes water-solubilization design to the next level. CpxA activity examined has no point of comparison either in the manuscript or from the literature.

Response: We thank the reviewer for the comment.

We agree that more activity comparison would enable more comprehensive understanding of the similarity and discrepancy between CpxA^{QTY} and native CpxA. In this revision, we made a more thorough activity comparison in the text, as shown below:

Page 7: “We noticed that, after 10 minutes of autophosphorylation, CpxA in lipid environment (proteoliposomes, detergent or nanodiscs) approaches

saturation^{29,30,33,37}, whereas CpxAQTY remains unsaturated, albeit with a significantly reduced rate. This may suggest a slight decrease in the autokinase activity of CpxAQTY compared to CpxA.”

Page 8: “Phosphorylated CpxA^{QTY} became almost undetectable within 0.5 minutes, with saturation of CpxR phosphorylation reached within 1 minute; in contrast, after 15 minutes of phosphotransfer, phosphorylated CpxA became barely detectable, with CpxR still acquiring phosphate groups^{30,33}. This tends to suggest that the phosphotransferase activity of CpxA^{QTY} is higher than that of CpxA.”

Page 8: “In the case of CpxA, phosphorylated CpxR became barely detectable within 30 minutes after the addition of CpxA^{33,37}; whereas after 120 minutes of dephosphorylation by CpxA^{QTY}, a small amount of phosphorylated CpxR could still be detected. We speculate that the phosphatase activity of CpxA^{QTY} is lower than that of CpxA.”

Page 9: “In the case of CpxA³³, compared to pH 7.0, the phosphorylation rate increased by approximately 2.5-fold at pH 7.5 and by about 4-fold at pH 8.0; as for CpxA^{QTY}, compared to pH 7.0, the phosphorylation rate increased by approximately 2.7-fold at pH 7.5 and by about 7.5-fold at pH 8.0.”

Page 9: “The autophosphorylation rate of CpxA³³ was reported to be inhibited by approximately half when equimolar concentration of CpxP was added, while CpxA^{QTY} requires approximately 10-fold CpxP concentrations. This indicates that the sensitivity of CpxA^{QTY} to CpxP tends to be lower than that of CpxA.”

Page 10: “The approximate response range for CpxA^{QTY} was 10 to 100 mM, and for CpxAc, it was 5 to 50 mM. CpxA in liposomes showed a minor response to 0.5 mM KCl and a major response to 500 mM³³. It appears that CpxA^{QTY} is less sensitive to K⁺ compared to CpxA.”

Furthermore, we have conducted further analysis regarding the activity comparison and have included additional discussion. Please see details in the response to 10th *Other items and suggestions*.

Regarding the computational analysis, we acknowledge that we have incorporated a substantial amount of computational analysis into our current manuscript. As mentioned earlier, the direct determination of structures requires substantial work, which would be presented a separate manuscript in a follow-up study. We harnessed the advantages of AlphaFold2 and MD simulations to probe the principles underlying the balance of solubility, stability and activity. Furthermore, experimental validations were provided to support our hypotheses.

2). *Early in the introduction it will be interesting to hear the limitations of detergents, then move to the benefits to water-soluble methods.*

Response: We thank the reviewer for the advice and totally agree with the reviewer. We have added the specific challenges of membrane proteins and the limitations of detergents to the early part of the introduction:

Page 3: “Challenges associated with membrane proteins include low expression levels, high cost and laboriousness for detergent screening, low stability in detergent-solubilized forms, and greater difficulties in obtaining high-quality crystals².”

3). *Line 63 and 64 seem to conflict with previous work from the authors using other membrane proteins, in active conformations. If this is true, then the statement negates the need for this article, and its impact.*

Response: We thank the reviewer for the comment. The statement “*It is unclear whether we can simultaneously render membrane proteins completely water-soluble and retain their **function***” in the manuscript indeed failed to convey what we intend to convey here, namely, the previously-unachieved preservation of the **functional integrity**, not limited to the single dimension of function, such as ligand-binding. So we have revised it as:

Page 3: “It is unclear whether we can simultaneously render membrane proteins completely water-soluble and retain their **functional integrity**.”

4). The article does not describe the “trial and error” of the QTY method. Is it necessary to replace ALL respective residues in the method or does the algorithm suggest different alternatives/permutations of the mutagenesis?

Response: We thank the reviewer for raising this question and being interested in QTY code. In our first article of QTY code²², we introduced the “trial and error” of QTY code (see the 'Initial Development of the QTY Variant of CXCR4^{QTY}' section in the Results). Initially, we applied QTY code to only 28 lipid-facing residues and later expanded it to 56 residues. Although we established a library of ~ 2 million variants and conducted yeast two-hybrid screening, we were unable to obtain well-expressed variants. A radical shift occurred when we replaced all relevant hydrophobic residues: the designed variant performed well in both expression and detergent-free purification, even with retained ligand-binding activity. Since then, we usually apply QTY code to all candidate residues and typically obtain well-behaved designs for expression and detergent-free purification. On our QTY design server, we offer options to meet customized needs, such as specifying target positions. We agree with the review’s concern that it might not be necessary to replace all respective residues, and the substitutions could be excessive. However, in practice, we found it extremely challenging to determine which substitutions are essential and which are not, as mentioned in our earlier trial-and-error stage. Particularly, most membrane proteins lack structural information as a reference, making it even more challenging to identify necessary substitutions. Therefore, replacing all candidates is the current optimal approach, which also reflects a core advantage of QTY code, i.e., independence on structural information.

5). Line 81, the term “biomimetic sensing” is used. Would this imply that the protein be prepared on a membrane or substrate? For that native lipids or detergents, etc. can be used on the native protein. It is not clear to me how QTY will aid in this application.

Response: We thank the reviewer for raising this question and being interested in this application of QTY code. In the recently published paper²³, the authors devised a biomimetic sensing system.

This system is referred to as "biomimetic" because it mimics the way biological systems detect and recognize analytes. i) It employs critical receptors found in human physiological processes and integrates them with electronic devices, simulating functionalities found in biological systems. ii) The monolayer component, S-layer protein, is originally a part of the cell envelope of some microbes, providing the microarray pattern mimicking cell envelope. iii) The graphene-based field-effect transistor (GFET) array in the system imitates the redundancy of receptors found in mammalian sensory systems, providing a multitude of sensing units that function similarly to a natural sensory system. The QTY-membrane receptor functionalized biosensor detects biological signal in biofluids in a manner that highly resembling the performance from a living organism, but gives statistically significant electrical signals that can be easily obtained for data processing.

Its architecture encompasses a QTY code designed membrane protein -based "dual-monolayer" construct as the bio-specific functional probe.

Within this system, compared to the use of native membrane-bound receptors, the roles of QTY code include: 1) making the production of the receptors more efficiently with lower cost; 2) rendering the receptors highly-thermostable with native ligand selectivity; 3) circumventing the potential adverse effects of detergents, such as decreasing the signal-to-noise ratio during electrical sensing and incapability of keeping membrane proteins stable for a long term.

We will be delighted if you would like to learn more about this application.

6). Line 131, use of "without detergent" I suggest "detergent-free." It was used by the authors once before.

Response: We thank the reviewer for the comment. We fully agree with the reviewer that the depiction should be consistent. We have revised it as:

Page 6: “It is worth noting that as with the typical water-soluble proteins, we purified CpxA^{QTY} under the native **and detergent-free** condition in the entire process.”

7). Line 131 and 134, introduce the result of DIMER formation of the CpxAQTY. Examining the crystal structure of E. coli CpxA amino acids 187-457, for example PDB ID 4BIU, we do observe a dimer. If this is a crystallography dimer and not a true dimer that could expand on your inference that CpxAQTY is a dimer. Also, a reference for the native CpxA being a dimer will be good to have.

Response: We thank the reviewer for raising this concern and the advice. There has been a consensus that histidine kinases function conservatively as dimers⁶⁻⁸, with only a few exceptions. Their cytoplasmic portions contain a specific domain responsible for dimerization, known as the dimerization and histidine-phosphotransfer (DHp) domain. The multiple crystal structures of CpxA cytoplasmic portion consistently reveal that the minimal asymmetric units are dimers and the dimerization interfaces are formed by the conserved DHp domain with extensive interface area (see PDB: 4BIU, 4BIV, 4BIW, 4BIX, 4BIY, 4BIZ, 4CB0, 5LFK). In fact, structural biology studies indicate that their kinase activity requires asymmetric coordinated movements of the two protomers¹⁶. Actually a study¹⁹ explicitly indicating that the native CpxA is a dimer. We have added the brief discussion about the dimerization of CpxA to the main text with the citation on this reference:

Page 6: “Size exclusion chromatography (SEC) results showed that CpxA^{QTY} predominantly folded as dimers, the functional oligomeric state of a canonical SHK (Supplementary Fig. 1b). **Previous biochemical studies also indicated that CpxA functions as dimers³⁰.**”

8). Before I forget, please provide an Accession Number for the protein synthesized, and also a full amino acid sequence to observe where the TAGs are placed.

Response: We thank the reviewer for raising this concern. We synthesized the QTY variant of CpxA, namely CpxA^{QTY}, which has no accession number as a designed protein. P0AE82 is the accession number of CpxA from *E. coli* K-12 strain, the template we used for design. Here is the full amino acid sequence of CpxA^{QTY} (we placed the 6xhis-tag to the C-ter, shown in cyan):

MIGSLTARTYATYWQTQAQTQMOTQMOPKLDNRQMTTELLDSEQRQGLMIEQH
VEAELANDPPNDLMWWRRLFRAIDKWAPPGQRLLLVTTEGRVIGAERSEMQUIIRNFIGQ
ADNADHPQKKKYGRVELVGPFSVRDGEDNYQLYLIRPASSSQSDFINLLFDRPQQQOTT
TMQTSTPQQWQAWSLAKPARKLKNAADEVAQGNLRQHPELEAGPQEFLAAGASFNQ
MVTALERMMTSQQRLLSDISHELRTPLTRLQLGTALLRRRSGESKELERIEAQRLLDSM
INDLLVMSRNQKQNALVSETIKANQLWSEVLDNAAFEAEQMGKSLTVNFPPGPWPLYG
NPNALSALENIVRNALRYSHTKIEVGFVAVDKDGITITVDDDGPVSPEDREQIFRPFYRT
DEARDRESGGTGLGLAIVETAIQQHRGWVKAEDSPLGGLRLVIWLPLYKRSHHHHHH

9). *Line 152, what's the source of the antibodies used? pHis and any others.*

Response: We thank the reviewer for raising this concern. The source of anti-N3-Phosphohistidine (3-pHis), clone SC39-6 is rabbit. The source of goat anti-rabbit IgG-HRP conjugate is goat.

10). *Line 177, I would like to see an introduction via references, of the expected catalytic activities of CpxA. Reference 30 seems a good starting point, hopefully not the only one. This could be a good point to compare the use of detergents and proteoliposomes against the water-soluble CpxA.*

Response: We thank the reviewer for raising this concern and the valuable suggestion. In the initial manuscript, we standardized our activity comparison between CpxA and CpxA^{QTY} using the proteoliposomes results from Ref. 33 (Ref. 30 previously), due to Ref. 33's comprehensive functional characterization and the high similarity of liposomes to native membrane bilayers. Based on our research, there are four references that characterized *in vitro* catalytic activities of full-length CpxA, yet only Ref. 33¹⁵ characterized its responses to various signals.

Ref. 33 characterized the activities of CpxA mainly in proteoliposomes, while it was also mentioned in the text that the autokinase activity of CpxA in proteoliposomes was 1.5 times higher compared to CpxA in the detergent decylmaltoside. Ref. R24²⁴ characterized autokinase, phosphotransferase and phosphatase activity with the detergent Brij35. Ref. R16¹⁶ characterized autokinase and phosphotransferase activities, also with Brij35 as the detergent. Ref. R19¹⁹ characterized autokinase and phosphotransferase activities using nanodiscs and proteoliposomes. Comparing these results, we found that CpxA prepared by different methods showed consistency in catalytic activities, albeit also some differences. Thus, for the three basic catalytic activities, we have incorporated other references into the activity comparison, in addition to Ref. 33. Here is the example:

Page 7: “We noticed that, after 10 minutes of autophosphorylation, CpxA in lipid environment (proteoliposomes, detergent or nanodiscs) approaches saturation^{29,30,33,37}, whereas CpxAQTY remains unsaturated, albeit with a significantly reduced rate. This may suggest a slight decrease in the autokinase activity of CpxAQTY compared to CpxA.”

Furthermore, in this revision, we conducted a more detailed activity comparison with CpxA in lipids, showing that actually there are some discrepancies in the functional performance between CpxA^{QTY} and CpxA. These discrepancies did not show a consistent trend; for example, CpxA^{QTY} exhibited lower autokinase activity but higher phosphotransferase activity, implying the complexity of this issue. We hypothesize that these discrepancies may be attributed to at least three possible reasons: (1) some alterations in activity caused by QTY design-induced structural perturbations; (2) the removal of the lipid environment; (3) potential subtle differences in experimental conditions. Therefore, this issue needs further experimental studies. We have added an exclusive paragraph in the Discussion section for this concern and its relevance to the focus of our work:

Page 21: “Despite the remarkable preservation of the molecular function by QTY design, we still noticed some discrepancies in functional performance between CpxA^{QTY} in water and CpxA in lipid environment, by comparing our results with data from previous publications. Considering the significant sequence variation, the structure of the transmembrane domain may be still slightly affected. Moreover, interactions with the lipid bilayer are one of the key factors influencing the activity of membrane proteins¹. Therefore, the discrepancies could be attributed to some alterations in activity caused by QTY design-induced structural perturbations, the removal of the lipid environment, or potential subtle differences in experimental setups, which requires further investigation.”

11). Line 262-279, the discussion of the water molecules do not belong in this manuscript, especially the discussion of “a water molecule,” line 270. Too much of a stretch to add to the desired conclusion. As mentioned above a hydrophobic and hydrophilic environment are very different. You could have created your CpxAQTY model, revert the mutations, add lipids and run simulations anew. Then you could compare which environments have water or need water to support stability. You could also feed those new coordinates to the Collective Variables, etc. A lot of work, perhaps for a separate article.

Response: We thank the reviewer for raising this concern and valuable suggestions.

We agree that the role of water molecules revealed by CpxA^{QTY} simulations may be irrelevant to the stabilization and signaling of the native CpxA. In the newly-added simulations of CpxA in a lipid bilayer, we observed only a few water molecules (roughly two) involved in stabilization. Apparently, these water molecules could only make limited contribution to stabilization. The relevance of water molecules to signaling is also hard to elucidate. Therefore, we deleted some extended discussion in the Discussion section about the roles of water molecules in native signaling proteins, and focused more on the roles in CpxA^{QTY} indicated by our results.

Page 21: “... considerably facilitated the interactions of the buried polar side chains, smoothening the contacts of residues like ‘lubricant’. **During the conformational changes, the water molecules actively engaged in the rearrangement of the H-bond network. These findings underscore the important role of water molecules in the stabilization and dynamics of CpxA^{QTY}. In the apo and holo crystal structures of NarQ, multiple water molecules in the center of transmembrane helical bundle participated in the conformational change, mediating the interaction rearrangement upon binding the signal molecule. Moreover, the role of water molecules in stabilization and signaling activation also was observed in other transmembrane signaling proteins**”

Nonetheless, our findings about water molecules actually aim to probe the basis of the dynamic stabilization in **CpxA^{QTY}**, rather than provide insights into the stabilization of CpxA. Therefore, we kept other contents about water molecules unchanged. We believe that water molecules play a significant role in the stabilization and dynamics of CpxA^{QTY}. The transmembrane domain of CpxA^{QTY} is mainly composed of polar residues. How such a four-helix bundle, whose core is predominantly comprised of polar residues, maintains stability and undergoes appropriate conformational changes is an intriguing question. The interactions among these polar residues have not been elaborately designed. However, the polar atoms of buried polar residues must participate in hydrogen bonding; otherwise, a considerable energetic penalty would be incurred²⁵. Furthermore, even if these polar residues closely mimic the shape of the original hydrophobic residues, their substitution is likely to introduce minor clashes or voids within the structure. As our results demonstrate, water molecules play a crucial role. We observed that the mobile water molecules, flexibly forming H-bonds (Supplementary Fig. 4, 5), considerably facilitated the interactions of the buried polar side chains, smoothening the contacts of residues like “lubricant”. During the conformational changes, the water molecules actively engaged in the rearrangement of the H-bond network (Supplementary Fig. 7b-e).

12). Figure 4, and its discussion; I would like part of the discussion to center around the native CpxA, from reference 30 or others.

Response: We thank the reviewer for raising this concern. We believe that our responses to 1st and 10th issue of your *Other items and suggestions* are adequate to address this concern. We compared the activities of CpxA^{QTY} with CpxA and discussed the catalytical activities of CpxA prepared by different approaches.

13). Line 485, the gradient was run in the AKTA?

Response: We thank the reviewer for the question. For immobilized metal affinity chromatography (IMAC) stage, we used the traditional gravity columns, instead of fast protein liquid chromatography by AKTA systems. We found the statement in Line 485 indeed not appropriate, so we have revised it as:

Page 23: “After centrifugation, the supernatant was loaded onto an Ni-NTA **gravity** column (Qiagen) (2 mL) equilibrated with the binding buffer. After washes with the same buffer, the protein was then eluted with a 20-50-80-300 mM imidazole gradient.”

14). Figure 2a. Currently websites like PubPeer and others enable the scrutiny of images like Western Blot. A feature of these analysis is that the picture could have been purposely changed. If you change Brightness/Contrast Figure 2a looks different, including the new band on Lane 4

Response: We thank the reviewer for raising this concern. We did change Brightness/Contrast to lower down the background noises. In the revised manuscript, we have canceled the Brightness/Contrast changes and use the original image:

16). *Figure 3b, changing Brightness/Contrast shows a sharp line on top of the bands.*

Response: We thank the reviewer for raising this concern. In fact, for this image, we did not make any processing. We looked into this issue and found the sharp line showed up after the manuscript was converted from word file into pdf file. We suppose this issue could be solved when published. Here is the original image:

Reviewer #2:

In their manuscript, Li and coauthors describe reengineering of a bacterial transmembrane receptor into a soluble protein by replacing amino acids in its transmembrane region. They then characterize the protein itself and signaling ability. The approach itself is interesting and worth probing. If transmembrane proteins could be transformed into soluble while maintaining their ligand binding and functional properties by following a simple amino acid replacement rule, this would strongly advance the field. However, it is unlikely that a simple replacement rule would generally work for all transmembrane proteins, especially complex multistate signaling proteins.

I believe that the data reported in this manuscript does not unambiguously demonstrate that the function of their engineered protein (CpxA-QTY) is indeed conserved. If the authors accurately performed all required experiments, the likely outcome would be that they would prove that the engineered protein is not comparable to wild type protein, invalidating the basic premise of the work. Consequently, this would be a technical report more suitable for a specialized journal.

Response: We appreciate all the time and effort from the reviewer, and thank the reviewer for acknowledging the significance of our manuscript, carefully reviewing and objectively commenting our results, and offering numerous constructive suggestions. These comments and suggestions have greatly prompted us to reexamine our work, enabling a substantial improvement in the overall quality of our manuscript. According to your requirements, we carefully designed and conducted additional experiments and simulations to provide more robust and comprehensive supports for our study. Below are the main added contents in this revision:

- 1) Circular dichroism characterization of CpxA^{QTY};
- 2) Finer characterization of K⁺ response of CpxA^{QTY};
- 3) Classical MD simulations and PBmetaD simulations of the CpxA AlphaFold2 model in a lipid membrane;
- 4) Functional characterization of point mutants in the transmembrane domain and sensor domain of CpxA^{QTY}.

We agree it is unlikely that a simple replacement rule would generally work for all transmembrane proteins. After all, any method has its optimal scope. In fact, our current work is also the exploration of QTY code's optimal scope. The successful preservation of functional integrity of a histidine kinase in this current manuscript represents a milestone of QTY code, offering novel insights into the field of water-soluble design for membrane proteins. Furthermore, it is likely that some of the substitutions are unnecessary or even unfavorable, if we apply the simple mode (substituting all L/I/V/F) of QTY design to various membrane proteins. We indeed observed the differences in performance on different types of targets and the presence of failure. Nevertheless, any method undergoes continuous refinements until proven robust. Currently, while we consider the general replacement rules underlying QTY code to be effective, one of our ongoing efforts is to enhance QTY design by incorporating more rational elements.

We agree that a more comprehensive and in-depth comparison with native CpxA would greatly benefit the manuscript. In this revision, we conducted simulations of CpxA in a lipid bilayer and functional characterization of some point mutants, further validating the consistency of the structural characteristics and general signaling mechanism between native CpxA and CpxA^{QTY}. Additionally, we made a more thorough activity comparison in the text. We suppose that we should not only highlight the consistency, but also objectively discuss the discrepancies in the performance between CpxA^{QTY} and CpxA. Accordingly, we have added an exclusive paragraph in the Discussion section for this issue. Below are the updates:

Page 7: “We noticed that, after 10 minutes of autophosphorylation, CpxA in lipid environment (proteoliposomes, detergent or nanodiscs) approaches saturation^{29,30,33,37}, whereas CpxAQTY remains unsaturated, albeit with a significantly reduced rate. This may suggest a slight decrease in the autokinase activity of CpxAQTY compared to CpxA.”

Page 8: “Phosphorylated CpxA^{QTY} became almost undetectable within 0.5 minutes, with saturation of CpxR phosphorylation reached within 1 minute; in contrast, after 15 minutes of phosphotransfer, phosphorylated CpxA became barely detectable, with

CpxR still acquiring phosphate groups^{30,33}. This tends to suggest that the phosphotransferase activity of CpxA^{QTY} is higher than that of CpxA.”

Page 8: “In the case of CpxA, phosphorylated CpxR became barely detectable within 30 minutes after the addition of CpxA^{33,37}; whereas after 120 minutes of dephosphorylation by CpxA^{QTY}, a small amount of phosphorylated CpxR could still be detected. We speculate that the phosphatase activity of CpxA^{QTY} is lower than that of CpxA.”

Page 9: “In the case of CpxA³³, compared to pH 7.0, the phosphorylation rate increased by approximately 2.5-fold at pH 7.5 and by about 4-fold at pH 8.0; as for CpxA^{QTY}, compared to pH 7.0, the phosphorylation rate increased by approximately 2.7-fold at pH 7.5 and by about 7.5-fold at pH 8.0.”

Page 9: “The autophosphorylation rate of CpxA³³ was reported to be inhibited by approximately half when equimolar concentration of CpxP was added, while CpxA^{QTY} requires approximately 10-fold CpxP concentrations. This indicates that the sensitivity of CpxA^{QTY} to CpxP tends to be lower than that of CpxA.”

Page 10: “The approximate response range for CpxA^{QTY} was 10 to 100 mM, and for CpxAc, it was 5 to 50 mM. CpxA in liposomes showed a minor response to 0.5 mM KCl and a major response to 500 mM³³. It appears that CpxA^{QTY} is less sensitive to K⁺ compared to CpxA.”

Page 21 (Discussion): “Despite the remarkable preservation of the molecular function by QTY design, we still noticed some discrepancies in functional performance between CpxA^{QTY} in water and CpxA in lipid environment, by comparing our results with data from previous publications. Considering the significant sequence variation, the structure of the transmembrane domain may be still slightly affected. Moreover, interactions with the lipid bilayer are one of the key factors influencing the activity of membrane proteins¹. Therefore, the discrepancies could be attributed to some alterations in activity caused by QTY design-induced

structural perturbations, the removal of the lipid environment, or potential subtle differences in experimental setups, which requires further investigation.”

Additionally, we agree that our interpretation for data and conclusions should be clear and more precise. The notable example is

ABSTRACT: “We show that the designed CpxA^{QTY} exhibits expected biophysical properties and *fully* preserved native molecular function, ...”.

The word ‘fully’ may have given the impression that we were claiming the signaling properties and mechanism of CpxA^{QTY} are fully identical to those of wild-type CpxA. In fact, we intended to emphasize functional integrity **in a qualitative manner**, rather than quantitatively; we aim to show that the **general** signaling mechanism is conserved (how the signals are sensed and transduced; what are the major conformational changes like), rather than the detailed mechanism (which are the key residues and interactions). Therefore, we have replaced ‘**fully**’ in our manuscript with ‘**highly**’.

Below are the detailed responses to the concerns.

1. Regarding CpxA-QTY:

1.1 SEC chromatogram presented in Suppl. Fig. 1b shows some aggregation. Is this aggregation ongoing? You cannot claim that the protein is soluble if it eventually aggregates.

Response: We thank the reviewer for raising this concern. It is true that some aggregates were observed in the SEC result of the purified CpxA^{QTY}. In fact, protein aggregation in solution is very normal in the purification process²⁶; different proteins show different kinetics of aggregation²⁷. From our purification experience of various soluble proteins, after the whole process of purification, it is normal that there are

some aggregates shown by SEC. We suppose it is not surprising that CpxA^{QTY}, as a water-solubilized variant of a transmembrane protein, showed some aggregation propensity. If CpxA^{QTY} were insoluble, it would completely precipitate during the purification process and would not be obtained under detergent-free conditions, with the expected dimeric form. Our results show that the aggregates are only a minor fraction of the total protein, with the majority being dimers. Actually, just as working with conventional soluble proteins, we subjected the protein to a complete purification process, froze it, thawed it for multiple experiments, and did not observe visible precipitation. Furthermore, if CpxA^{QTY} were insoluble, we would not have obtained the major dimer peak in SEC and the dominant dimeric shape in AUC (continuous high-speed centrifugation at approximately 20°C for about 5 hours).

1.2 are the sensor and the TM domain of CpxA-QTY folded in solution? Please provide some experimental characterization (circular dichroism, small-angle scattering, maybe electron microscopy).

Response: We thank the reviewer for raising this concern and the valuable suggestion. We agree that we should provide experimental characterization to validate if the sensor and TM domain are folded. We suppose that circular dichroism is a good choice, as it can show the folding state, but also enable the comparison of the secondary structure content in the experimental data and in the AlphaFold2 structure model, providing more evidence for the expected folding of CpxA^{QTY}. Therefore, we have conducted the circular dichroism characterization of CpxA^{QTY} and analyzed the secondary structure content (see details in Methods). The results show that the secondary structure content (α -helix: 60.9%, β -sheet: 15.0%) were close to that in the AlphaFold2 model (α -helix: 61.9%, β -sheet: 17.5%). The high content of ordered secondary structures and the high consistency in the experimental and predicted secondary structure content support that CpxA^{QTY} is well-folded. We have added the results of this part to the main text:

Fig. 2d. Circular dichroism (CD) results of CpxA^{QTY}. CD spectra (Total, purple solid line) and its α -helix (red) and β -sheet (blue) component spectra. The secondary structure analysis was performed using JASCO multivariate secondary structure (SS) estimation program. The content of α -helix and β -sheet in the AlphaFold2 (AF2) model were determined using PyMOL.

Page 7: “To further characterize the folding state of CpxA^{QTY}, we conducted circular dichroism measurements (Fig. 2d). The analysis of the secondary structure composition revealed α -helix and β -sheet content of 60.9% and 15.0%, respectively, which closely matched the content in the predicted structure by AlphaFold2³⁴, 61.9% for α -helix and 17.5% for β -sheet. The high content of ordered secondary structures and the high consistency in the experimental and predicted secondary structure content support that CpxA^{QTY} is well-folded.”

1.3 out of three functional tests, only one (pH dependence) shows that CpxA-QTY is able to recognize signals, which are not recognized by a truncated CpxAc protein. However, pH is a very generic environmental condition that affects many molecular processes, so the data does not demonstrate that pH is recognized by the sensor domain via the same mechanism as in WT CpxA.

Response: We thank the reviewer for raising this concern. We agree that the additional data should be provided to sufficiently demonstrate that pH is recognized by the sensor domain via the same mechanism as in WT CpxA.

To experimentally validate the consistency of the general signaling mechanism between CpxA^{QTY} and CpxA, i.e. sensing pH by the sensor domain and transducing the signal through the transmembrane domain, we attempted to “turn off” the signaling activity by mutagenesis in the transmembrane domain and sensor domain, based on modeling and simulation results. We selected four residues in transmembrane domain that played critical roles in the H-bond network of CpxA^{QTY} in the simulations, and two residues in sensor domain that might be involved in the perception of alkaline pH. All the mutants showed abolished or reduced sensitivity to pH variation. Below are the added contents about this part of work:

Page 18:

Point mutations diminish the signaling activity

We attempted to “turn off” the signaling activity by mutagenesis, based on modeling and simulation results, to further experimentally validate the consistency of the general signaling mechanism between CpxA^{QTY} and CpxA, i.e. sensing pH by the sensor domain and transducing the signal through the transmembrane domain. Four residues in the transmembrane domain were individually mutated back to their original hydrophobic counterparts, namely Q17L, T20V, Q21L and Q182L. These residues played critical roles in the H-bond network of CpxA^{QTY} in the simulations (Fig. 5e). We found that all these mutants exhibited diminished (Q17L, T20V and Q182L) or abolished sensitivity (Q21L) to pH variation (Supplementary Fig. 13). Furthermore, two basic residues in the sensor domain were mutated to alanine individually. We inferred that these two residues might be involved in the perception of alkaline pH, as they are both located at the domain center and exposed to the solvent according to the AlphaFold2 model (Supplementary Fig. 14a). It was observed that R99A no longer responds to pH variation, while R106A exhibits reduced pH sensitivity (Supplementary Fig. 14b, c).

The point mutations in the transmembrane domain and sensor domain diminished the signaling receptor activity of CpxA^{QTY}, further supporting that the general transmembrane signaling mechanism of CpxA is well-conserved by QTY design.

Supplementary Figure 13 | pH-sensing activities of transmembrane domain mutants of CpxA^{QTY}. **a-d**, Results of CpxA^{QTY}-Q17L, T20V, Q21L, and Q182L, respectively. The mutants were phosphorylated with 1 mM ATP at RT for 2 min, with indicated pH. The phosphorylation level at pH8.5 was set as 100% for normalization. For all the three experiments, representative pHs western blot results were shown (left). The quantified phosphorylation level of the three independent experiments was shown (right). Data are shown as mean \pm SEM.

Supplementary Figure 14 | pH-sensing activities of sensor domain mutants of CpxA^{QTY}. **a**, Charged residues in the sensor domain. All charged residues are shown as sticks. Chain A is shown in green and Chain B in yellow. **b-c**, Results of CpxA^{QTY}-R99A and R106A, respectively. The mutants were phosphorylated with 1 mM ATP at RT for 2 min, with indicated pH. The phosphorylation level at pH8.5 was set as 100% for normalization. For all the three experiments, representative pHis western blot results were shown (left). The quantified phosphorylation level of the three independent experiments was shown (right). Data are shown as mean ± SEM.

1.4 in the KCl dependence test (Fig 4c), CpxA-QTY is responsive to KCl levels only above 10 mM, whereas WT CpxA in liposomes appears to be activated by as little as 0.5 mM KCl (Fig 3A, doi 10.1074/jbc.M605785200). This likely indicates that the signaling properties of CpxA-QTY are altered.

Response: We thank the reviewer for raising this concern. We also noticed that the K⁺ signaling properties of CpxA^{QTY} seem altered to some extent. As mentioned above, we actually emphasize functional integrity in a qualitative manner, rather than quantitatively. Therefore, we do not expect that the signaling properties of CpxA^{QTY} are fully identical to those of WT CpxA. We consider it is acceptable that the activity alteration is not profound.

In the reference¹⁵, CpxA exhibited a minor response to 0.5 mM KCl, whereas CpxA^{QTY} is responsive to KCl levels above 10 mM. As discussed above, the discrepancies in functional performance between CpxA^{QTY} in water and CpxA in lipid environment may be attributed to at least three possible reasons: 1) some alterations in activity caused by QTY design-induced structural perturbations; 2) the removal of the lipid environment; 3) potential subtle differences in experimental setup. Therefore, currently we could not draw a conclusion about the cause of the discrepancies, warranting another dedicated investigation.

Regarding this part about K⁺ response, we also acknowledge that we should provide a detailed description of the differences in signaling properties between CpxA^{QTY} and CpxA, and our conclusion appeared to be imprecise. In this revision, we revised this part significantly and compared the KCl responses of CpxA^{QTY} and CpxAc over a finer concentration gradient.

Fig. 4c. CpxA^{QTY} can sense K⁺. CpxA^{QTY} (1 μM) or CpxAc (1 μM) were phosphorylated with 1 mM ATP at RT for 2 min, with different concentration of K⁺. The phosphorylation level with 200 mM KCl was set as 100% for normalization. For all the three experiments, representative pHis western blot results were shown (left). The quantified phosphorylation level of the three independent experiments was shown (right). Data are shown as mean ± SEM.

Page 10: “The other signal, K⁺, reported as the stimulus on the autokinase activity of CpxA³¹, strongly activated the autokinase activity of CpxA^{QTY}, but also that of CpxAc (Fig. 4c). The approximate response range for CpxA^{QTY} was 10 to 100 mM, and for CpxAc, it was 5 to 50 mM. CpxA in liposomes showed a minor response to 0.5 mM KCl and a major response to 500 mM³¹. It appears that CpxA^{QTY} is less sensitive to K⁺ compared to CpxA.

Therefore, we speculated that these two signals might regulate the autokinase activity of CpxA by interacting with both the extracytoplasmic sensor domain and the cytoplasmic portion, or only by interacting with cytoplasmic portion. Nevertheless, all these results indicated that CpxA^{QTY} remained sensitive to the signals that CpxA responds to, although the sensitivity to the signals seems to be altered to varying degrees.”

2. Regarding replacement of the buried amino acids in the TM part: in my view, this replacement completely changes the energetic landscape of the transmembrane domain (available conformations, energy differences between them, and energy barriers) and the resulting protein's dynamics is not in any way indicative of the dynamics of the original protein (CpxA). The manuscript currently lacks the appropriate controls (both in experimental and modeling parts) for comparing the dynamics to WT CpxA, as CpxA-QTY is only compared to CpxAc or CpxA-QTY-v.2.

Response: We thank the reviewer for the comment. We agree that dynamics data of WT CpxA for comparison with CpxA^{QTY} would provide stronger support to our conclusions. In this revision, we have conducted classical MD simulations and PBmetaD simulations of WT CpxA.

We ran 100-ns MD simulations of CpxA in the POPC bilayer. Similar to CpxA^{QTY} simulations, we input the AlphaFold2 model without the catalytic domain. We observed similar overall structural stability (Supplementary Fig. 8a) and local flexibility (Supplementary Fig. 8b) to those of CpxA^{QTY}. The periplasmic-facing region of CpxA TM domain exhibited high RMSF, consistent with the trend seen in CpxA^{QTY}, suggesting similar dynamics. We compared interactions in CpxA and CpxA^{QTY}, and have provided some examples (Fig. 7), demonstrating that hydrogen bonding between QTY residues in CpxA mimics the packing between hydrophobic residues in CpxA^{QTY} (replacing the previous Supplementary Fig. 3, which showed comparison with CpxA AlphaFold2 model).

Supplementary Figure 8 | RMSD and RMSF curves in MD simulations of CpxA. a, Backbone RMSD of the transmembrane domain, versus medoid frame. **b,** RMSF of the transmembrane domain, versus medoid frame.

Fig. 7 | Interactions of polar residues in CpxA^{QTY} mimicked the packing of the original hydrophobic residues in CpxA, revealed by MD simulations. a, (Left) Q17 in Chain A and Q17 in Chain B of CpxA^{QTY}. (Right) L17 in Chain A and L17 in Chain B of CpxA. **b,** (Left) Q21 in Chain A and T24 in Chain B of CpxA^{QTY}. (Right) L21 in Chain A and V24 in Chain B of CpxA. **c,** (Left) T12 in Chain B and Q182 in Chain B of CpxA^{QTY}. (Right) I12 in Chain B and L182 in Chain B of CpxA. Transmembrane regions are shown as cyan cartoon in CpxA^{QTY} simulation (left) and yellow cartoon in CpxA (right). Highlighted residues are shown as spheres. The structural snapshots were from the simulation medoid frames.

Moreover, we ran 2000-ns PBmetaD simulations of CpxA. Similar to CpxA^{QTY}, we observed two stable states and one intermediate state (Supplementary Fig. 9f). The conformational transition between the two stable states also corresponds to diagonal scissoring motion (Supplementary Fig. 9g). By comparing the interhelical interactions with those in CpxA^{QTY}, we found that many hydrophobic residues went through similar transition of roles in the conformation compared to those corresponding QTY residues, and provided the example of A: L/Q21 and A: L/Q168 in the top layer of the TM core (Supplementary Fig. 9h). The consistent performance of CpxA^{QTY} and CpxA in both MD simulations and PBmetaD simulations further supports that CpxA^{QTY} shares the similar structural characteristics and conformational changes with wild-type CpxA.

Supplementary Figure 9 (f-h) | PBMetaD simulation of CpxA. **f**, The curves of the CVs with simulation time. The patterns of the CV variation were classified into three states, separated by grey dashed lines. **g**, The conformational transition of the different states of the transmembrane domain. Top view of the conformations of State-1 (cyan), State-M (transparent grey) and State-2 (red), as viewed from the periplasm looking into the cytoplasm. “TM1” indicates TM1 helix in Chain A and “TM1’ ” indicates TM1 helix in Chain B. **h**, Snapshots of the top layer of the interaction network of State-1 (left) and State-2 (right). The display style is the same as Fig. 6d. The structural snapshots of **g** and **h** were from the medoid frames of the clusters, corresponding to **e**.

Please see the detailed contents in the newly-added section *CpxA shows similar structural dynamics and conformational changes*.

As for the significance of CpxA^{QTY}-v.2 (removing QTY design in the TM core), actually we did not intend to use it to reflect the dynamics of WT CpxA but rather to illustrate the contribution of the H-bond network formed by the buried QTY residues to dynamics and signaling activity. Furthermore, one of the key features of QTY code compared to other water-solubilization design approaches is the replacement of buried residues, not only exterior ones, with polar residues. The results of CpxA^{QTY}-v.2 suggest that the substitution of both interior and exterior residues may be a crucial factor in the successful maintenance of transmembrane functionality by QTY code. Therefore, we have retained this portion, albeit moving the previous Fig. 7 to the extended data. For a detailed discussion on this part, please refer to the third paragraph in the Discussion section.

3. Regarding the modeling part of the manuscript:

3.1 the authors present an AlphaFold model of dimeric CpxA, which shows unexpected arrangement of sensor domains, because most sensor domains dimerize via alpha-helices. Consequently, authors need to validate their model, or construct (and validate) a model that conforms to the usual expectations of domain structure in sensor histidine kinases.

Response: We thank the reviewer for raising this concern. We agree that the dimerization of the sensor domain (SD) in the AlphaFold2 model seems not typical. However, here we reason that this dimerization mode is likely to be reliable.

First, in Supplementary Fig. 2, we showed the PAE plots, which can indicate complex structure confidence. The overall confidence for the interactions between the two SD protomers is good, revealed by the high content of blue area (see the figure below).

Supplementary Figure 2a (right) | PAE plots for assessing complex structure confidence. “A” and “B” denotes chain A and B. TM1 (red) and TM2 (cyan) has been labeled. **The dashed rectangle indicates the region for sensor domain.**

Furthermore, we analyzed the dimerization profiles of CpxA SD in the simulations. RMSD curves showed a good stability of CpxA SD (Supplementary Fig. 3c, d). We then conducted binding free energy (BFE) calculation using the Molecular Mechanics Poisson–Boltzmann Surface Area (MMPBSA) algorithm. The results showed a decent binding strength ($BFE = -36.25 \pm 0.09$ kcal/mol). We observed eight interface residues that significantly contribute to the binding (per-residue contribution < -1 kcal/mol), and no interface residues that show negative contributions (Supplementary Fig. 15c). These results suggest that the dimerization mode in the AlphaFold2 model is reliable.

Supplementary Figure 15 | CpxA SD in the AlphaFold2 model shows a distinct dimerization mode. **a**, Comparison of CpxA SD dimer in the AlphaFold2 model with typical SD dimers. **b**, The crystal structures of CpxA SD. **c**, Dimer interface of CpxA SD revealed by MMPBSA binding free energy calculation. The residues that contribute (per-residue contribution > 1 or < -1 kcal/mol) to interface formation are shown as sticks, and in blue, with the color intensity determined by their contribution to binding free energy (as the bar shown below).

In fact, the SD in the CpxA AlphaFold2 model indeed dimerizes via α -helices. Nonetheless, typically the SD dimerizes via the first helix (starting from the N-terminus), whereas in this model it dimerizes via the third helix (Supplementary Fig. 15a). There are two released crystal structures of CpxA SD: one is PDB: 3V67, from *Vibrio parahaemolyticus*¹⁸; the other is recently-released PDB: 8UK7, from *E. coli* (Supplementary Fig. 15b). We noticed that two significant differences between CpxA SD and that of other typical SHKs: 1) its N-terminus forms a 10-residue loop

before the first helix rather than a continuous helix connected with the transmembrane helix; 2) the crystal structures of CpxA SD only revealed the monomeric structure (in fact they are dimeric, but they are crystallographic dimers), rather than dimeric structure like most resolved SD crystal structures. The failure of both the two CpxA SD crystals in forming a bona fide dimer suggests that the dimerization might be not as strong as that of others. We inferred that the dimerization of CpxA SD might be distinct from the typical SD dimerization.

We included this part of discussion in the Supplementary Information and updated the main text:

Page 11: “Interestingly, the sensor domain in this model showed an untypical dimerization (Supplementary Discussion).”

3.2 the authors need to compare dynamics of CpxA-QTY with that of WT CpxA in a bilayer to claim that the signaling mechanism is conserved.

Response: We thank the reviewer for raising this concern. We show that the consistent performance of CpxA^{QTY} and CpxA in both MD simulations and PBmetaD simulations further supports that CpxA^{QTY} shares the similar structural characteristics and conformational changes with wild-type CpxA. We believe our response to Comment 2 above could address this concern.

3.3 RMSD data (ED Fig. 1a) shows that the structure is likely very dynamic, whereas no data on the HAMP domain and sensor domain is presented at all.

Response: We thank the reviewer for raising this concern. We agree that we should also present RMSD data of the sensor domain and HAMP domain, for the data integrity and comparison with the transmembrane domain. Both the two domains showed RMSD around 2 Å for the most of time, although HAMP domain fluctuated somewhat more, indicating the similar overall structural stability of the three domains. As the reviewer suggested, we have added the RMSD data of the HAMP domain and sensor domain to Supplementary Information and updated the main text:

Supplementary Figure 3 (a-f) | RMSD and RMSF curves in MD simulations of CpxA^{QTY}. Backbone RMSD of the transmembrane domain, versus medoid frame (a) or versus starting frame (b); Backbone RMSD of the sensor domain, versus medoid frame (c) or versus starting frame (d); Backbone RMSD of the HAMP domain, versus medoid frame (e) or versus starting frame (f).

Page 11: “During the 100-ns simulation, the transmembrane domain showed favorable structural stability, with backbone RMSD versus medoid below 2 Å for the majority of the simulation time (1.88 Å on average) (Supplementary Fig. 3a); the sensor domain and HAMP domain showed similar stability (Supplementary Fig. 3c, e). In the transmembrane domain, the root-mean-square fluctuation (RMSF) of most residues was around 1 Å, but that of the periplasm-facing region reached 2 Å, even 3 Å (Supplementary Fig. 3g), indicating the high local flexibility.”

Here we still would like to stress that in fact, the focus of MD simulations is the transmembrane domain, where we imposed design. In addition, we intended to show the local dynamics within the transmembrane domain by per-residue RMSF data, instead of RMSD data, which mirrored the overall dynamics, as the above text stated.

Less important issues:

1) page 3 line 57: the authors might want to discuss a recent preprint about generating a soluble GPCR (doi 10.1101/2023.05.09.540044).

Response: We thank the reviewer for informing us this manuscript and we agree that it is necessary to discuss this recent paper²⁸ as it is highly related to our work. It integrated advanced computational techniques to design water-soluble analogs of membrane proteins and validated the high structural precision of the designs. We suppose that this work introduces a novel computational pipeline for designing water-soluble membrane proteins and further demonstrates the structural reversibility of membrane proteins with water-soluble counterparts, providing positive support for our manuscript. Unfortunately, this article lacks functional characterization, and the proposed methods rely on available structures. We have incorporated a discussion of this reference in Introduction of our current manuscript.

Page 3: “Since the early 2000s, successive efforts have proved its feasibility, with designed water-soluble variants showing various degree of solubility and preservation of structural characteristics^{2,5-11}. Particularly, the NMR structural study of the water-soluble analog of the potassium channel KcsA provided concrete evidence that water-solubilization design was capable of preserving the three-dimensional structure well⁸. **A recent study accurately designed the water-soluble analogs of integral membrane protein folds, further demonstrating the feasibility of converting membrane protein folds into water-soluble protein folds¹¹.**”

2) page 4 lines 94-96: the mechanism remained elusive, but data generated in recent ~10 years provides more and more details. On the opposite, the data presented by the authors do not clarify anything about the mechanism of transmembrane signaling.

Response: We thank the reviewer for the comment. We agree that in recent years the researches on transmembrane signaling mechanism provide more and more details. Here we intended to emphasize the scarcity of the structural biology study on

transmembrane signaling mechanism by SHKs, yet we should also mention the progress in recent years. We have revised the statement as:

Page 5: “Due to the intrinsic difficulty in studying the structural biology, shared by most membrane proteins, the mechanism of transmembrane signaling by SHKs remained elusive, **although the researches in recent years on transmembrane signaling mechanism provide more and more details²⁰⁻²²”.**

In our initial manuscript, we actually provided our insights into the mechanism of transmembrane signaling. Regarding the general principles underlying the signaling mechanism, we highlighted the important role of the structural flexibility of the transmembrane domain in signaling through the MD simulations of CpxA^{QTY}, CpxA and CpxA^{QTY-v.2}. We discussed this issue in fourth paragraph of the Discussion section. As for the signaling mechanism in the case of CpxA^{QTY}, in this revision we conducted simulations of native CpxA to validate the consistency of the structural dynamics and conformational changes between CpxA^{QTY} and CpxA. Please see the detailed contents in the newly-added section *CpxA shows similar structural dynamics and conformational changes*. The comprehensive clarification for the signaling mechanism by CpxA^{QTY} warrants the determination of the multi-state structures, which would be a complex and long-term task. In fact, we have set about this task and it may be published in another dedicated manuscript.

3) page 5 lines 104-107: this is a strong and unproven assumption that is very likely to be incorrect. Polar amino acids are able to form hydrogen bonds, and interactions between them would be stickier and likely to attract water molecules.

Response: We thank the reviewer for raising this concern. We agree that this assumption seems strong. We have weakened the strength and revised it as:

Page 5: “Theoretically, **it is possible that** designed interhelical polar interaction networks by QTY code may replace the interaction networks in transmembrane hydrophobic helices responsible for SHK signaling and realize signal transduction through the conformational changes that mimic the original ones.”

Here we proposed a hypothesis. Based on this hypothesis, we conducted the subsequent experiments. Our experimental data showed that the function was indeed mimicked (Fig. 4a); our *in silico* data showed that the conformational changes were indeed mimicked (Fig. 6c vs Supplementary Fig. 9g) and the interactions were indeed mimicked (Fig. 7; Fig. 6d vs Supplementary Fig. 9h).

We agree that the packing of hydrophobic residues is very different from the interactions between polar residues. However, we suppose that it is still possible that polar interactions could mediate the same conformation changes that hydrophobic interactions mediate. The structure of a protein determines its function. Since the structures of polar residues could mimic those of hydrophobic residues (the essence of QTY code), we believe that it is possible the function could also be mimicked. Although our current data could not fully demonstrate the correctness of our hypothesis, overall, all our data contribute to supporting the hypothesis.

4) page 10 lines 223-224 while pLDDT score for the transmembrane part is okay, the pLDDT scores for the linker between the TM part and sensor is low, and the linker structure is unreliable. The authors either need to validate the linker structure in the starting model or generate a reliable model.

Response: We thank the reviewer for raising this concern. We agree that the pLDDT score for the linker between the TM part and sensor is low.

In fact, this linker is very likely to be mainly loops. Generally, the secondary structure prediction by AlphaFold2 is accurate. As shown in the CpxA SD crystal structures in Fig. R1b, the linker is indeed mainly loop. Typically, loops are intrinsically disordered, thus they are highly flexible. The pLDDT score reflects

local confidence in the predicted structure. Therefore, the disordered loops naturally show low pLDDT scores^{1,29-31}. The high flexibility of loops also affects the local flexibility of the surrounding region, causing the relatively lower pLDDT scores of the surrounding region. In summary, we suppose that the linker is loop and the low pLDDT score is expected.

5) please include as a supplementary information the atomistic models used as starting structures, reference structures for RMSD/RMSF calculations and used for preparing the figures of the manuscript so that they can be analyzed by reviewers and readers.

Response: We thank the reviewer for the suggestion. In this revision, we have uploaded our code and relevant files that were used or output in the simulations to the repository Zenodo: <https://doi.org/10.5281/zenodo.10516203>.

6) Fig 2c: please indicate whether these are technical or biological replicates.

Response: We thank the reviewer for raising this concern. As the sample for nanoDSF was the same batch of purified CpxA^{QTY} and nanoDSF is an *in vitro* assay, the replicates are technical replicates. We have revised the caption of Fig. 2c:

Fig. 2c: “**Technical triplicates** of CpxA^{QTY} (red lines) were assayed...”

7) Fig 3a: makes an impression that all signals are recognized in the periplasm, whereas below you show that only pH is possibly recognized by the periplasmic domain.

Response: We thank the reviewer for raising this concern. We agree that the previous Fig. 3a may have not conveyed the message precisely. We have also placed some “signals” to the cytoplasm side and enlarged the size of the question mark to better convey the main idea of this schematic diagram.

Fig. 3a | Schematic diagram showing the activities of CpxA^{QTY}.

8) Fig 3a: please clarify in the legend that only a part of CpxR is shown.

Response: We thank the reviewer for the kind reminder. It also reminds us to clarify that the CpxA^{QTY} structure shown in this figure was predicted by AlphaFold2. We have revised the legend as:

Fig. 3a: “Schematic diagram showing the activities of CpxA^{QTY} (left). Phosphotransferase activity and phosphatase activity involve CpxR (right). **The predicted structure of CpxA^{QTY} by AlphaFold2 was shown.** The transmembrane region was shown in cyan. **Only the receiver domain of CpxR was shown.**”

9) Fig 4a vs b: do they match? (at 0 CpxP and pH 7.5).

Response: We thank the reviewer for the question. Yes, they match. While theoretically representing the same reaction conditions, these two bands originate from different experimental batches, involve distinct experimental groups for comparison, and appear on different blots. Typically, the exposure time for each blot should be adjusted to clearly visualize all bands and differences between them. Hence, the intensity of these two bands may appear significantly different, but direct comparison is not valid.

10) Fig 4c: please provide quantification similar to that in Fig. 4a,b.

Response: We thank the reviewer for raising this concern. We agree that we should provide quantification to consistently present the profiles of the responses to different signals. In this revision, we compared the KCl responses of CpxA^{QTY} and CpxAc over a finer concentration gradient. The updated figure and the text have been presented in the response to Comment 1.4.

11) ED Fig. 1a: please also show RMSD relative to the starting model.

Response: We thank the reviewer for raising this concern. We agree that along with RMSD versus the medoid frame, simultaneously providing RMSD relative to the starting model would help the readers better understand the dynamics of molecular trajectory. We have added the RMSD relative to the starting model for TM, SD and HAMP domain, respectively. We placed them into a separate figure in the supplementary information. Please see Comment 3.3.

12) Fig S1b: please indicate the column used for chromatography.

Response: We thank the reviewer for the suggestion. We have added this information:

Fig. S1b: “...Superdex 200 (Cytiva Life Sciences) was used for SEC”

Reviewer #3:

This manuscript uses a previously developed strategy for solubilizing membrane proteins (QTY code) to render signaling protein, CpxA water-soluble. A series of biophysical experiments are used to test functional preservation of the signaling protein. Using computer-generated models and simulation, functional preservation is attributed to a dynamic hydrogen bonding network introduced to the transmembrane region of the protein by the QTY code.

This work is further demonstration of the power of QTY code for generating and studying soluble versions of membrane protein receptors.

Comments for further discussion and analysis:

Response: We appreciate all the time and effort from the reviewer, and thank the reviewer for acknowledging the significance of our manuscript. The reviewer's comments and suggestions are very detailed and professional, guiding us to further scrutinize the weaknesses and limitations of this work. We have provided additional data as suggested by the reviewer and have diligently revised the manuscript. Please find the detailed responses to the concerns below.

1. The conclusion about signal transmission preservation in CpxAQTY is stronger than the data suggest. A more nuanced conclusion of the results in Figure 4 and signal preservation throughout the text should be provided.

Example conclusion given in the text:

“Nevertheless, all these results indicated that CpxAQTY responded to the signals in a manner highly similar to CpxA.”

While this statement is supported by the pH-dependence of autokinase activity in CpxA (Figure 4A) the results of the two subsequent experiments (4B and 4C) only allow for the conclusion that there is a difference between studying the water-soluble protein and the membrane-bound protein.

Because the control experiment demonstrates that CpxP and potassium ion are influencing the isolated cytoplasmic domain, these experiments do not rule out the possibility that the CpxAQTY data is purely a result of the cytoplasmic interaction and not the periplasmic interaction and subsequent signaling studied in reference 30.

As discussed briefly in the text, the result of these experiments could point to new biology, but it is just as likely that the results describe artifacts introduced by extracting a membrane protein from its native lipid environment (i.e. the absence of a hydrophobic barrier between sections of the protein).

The conclusion should highlight the triumph of functional preservation using the QTY method (demonstrated in Figure 4A) while also highlighting the difference between studying the water-soluble protein and the membrane-bound protein given the confounding results in Figures 4B and 4C.

Response: We thank the reviewer for the professional comment and the constructive suggestions, which greatly provoked our reflection on the statements in the initial manuscript.

We agree that our data interpretation and conclusions should be clearer and more precise. The example given by the reviewer (top) and another example (bottom) are the most representative ones. We have revised them as:

Page 10: “Nevertheless, all these results indicated that CpxA^{QTY} remained sensitive to the signals that CpxA responds to, although the sensitivity to the signals seems to be altered to varying degrees.”

ABSTRACT: “We show that the designed CpxA^{QTY} exhibits expected biophysical properties and highly preserved native molecular function, ...”

Another notable example is the word ‘fully’ in the abstract. The word ‘fully’ may have given the impression that we were claiming the signaling properties and mechanism of CpxA^{QTY} are fully identical to those of wild-type CpxA. In fact, we intended to emphasize functional integrity **in a qualitative manner**, rather than quantitatively; we aim to show that the **general** signaling mechanism is conserved (how the signals are sensed and transduced; what are the major conformational

changes like), rather than the detailed mechanism (which are the key residues and interactions). Therefore, we have replaced ‘fully’ with ‘highly’.

Furthermore, we agree that we should not only highlight the consistency, but also objectively discuss the discrepancies in the performance between CpxA^{QTY} and CpxA. In this revision, we made a more thorough activity comparison in the text, showing that actually there are some discrepancies in the functional performance between CpxA^{QTY} in water and CpxA in lipid environment.

Page 7: “We noticed that, after 10 minutes of autophosphorylation, CpxA in lipid environment (proteoliposomes, detergent or nanodiscs) approaches saturation^{29,30,33,37}, whereas CpxA^{QTY} remains unsaturated, albeit with a significantly reduced rate. This may suggest a slight decrease in the autokinase activity of CpxA^{QTY} compared to CpxA.”

Page 8: “Phosphorylated CpxA^{QTY} became almost undetectable within 0.5 minutes, with saturation of CpxR phosphorylation reached within 1 minute; in contrast, after 15 minutes of phosphotransfer, phosphorylated CpxA became barely detectable, with CpxR still acquiring phosphate groups^{30,33}. This tends to suggest that the phosphotransferase activity of CpxA^{QTY} is higher than that of CpxA.”

Page 8: “In the case of CpxA, phosphorylated CpxR became barely detectable within 30 minutes after the addition of CpxA^{33,37}; whereas after 120 minutes of dephosphorylation by CpxA^{QTY}, a small amount of phosphorylated CpxR could still be detected. We speculate that the phosphatase activity of CpxA^{QTY} is lower than that of CpxA.”

Page 9: “In the case of CpxA³³, compared to pH 7.0, the phosphorylation rate increased by approximately 2.5-fold at pH 7.5 and by about 4-fold at pH 8.0; as for CpxA^{QTY}, compared to pH 7.0, the phosphorylation rate increased by approximately 2.7-fold at pH 7.5 and by about 7.5-fold at pH 8.0.”

Page 9: “The autophosphorylation rate of CpxA³³ was reported to be inhibited by approximately half when equimolar concentration of CpxP was added, while CpxA^{QTY} requires approximately 10-fold CpxP concentrations. This indicates that the sensitivity of CpxA^{QTY} to CpxP tends to be lower than that of CpxA.”

Page 10: “The approximate response range for CpxA^{QTY} was 10 to 100 mM, and for CpxAc, it was 5 to 50 mM. CpxA in liposomes showed a minor response to 0.5 mM KCl and a major response to 500 mM³³. It appears that CpxA^{QTY} is less sensitive to K⁺ compared to CpxA.”

We found that these discrepancies did not show a consistent trend; for example, CpxA^{QTY} exhibited lower autokinase activity but higher phosphotransferase activity, implying the complexity of this issue. We suppose that these discrepancies may be attributed to at least three possible reasons: 1) some alterations in activity caused by QTY design-induced structural perturbations; 2) the removal of the lipid environment; 3) potential subtle differences in experimental conditions. Therefore, this issue may need further future experiments to resolve. We have added an exclusive paragraph in the Discussion section for this concern and its relevance to the focus of our work:

Page 21: “Despite the remarkable preservation of the molecular function by QTY design, we still noticed some discrepancies in functional performance between CpxA^{QTY} in water and CpxA in lipid environment, by comparing our results with data from previous publications. Considering the significant sequence variation, the structure of the transmembrane domain may be still slightly affected. Moreover, interactions with the lipid bilayer are one of the key factors influencing the activity of membrane proteins¹. Therefore, the discrepancies could be attributed to some alterations in activity caused by QTY design-induced structural perturbations, the removal of the lipid environment, or potential subtle differences in experimental setups, which requires further investigation.”

2. Simulation of the CpxA AlphaFold model in a lipid membrane would enable general conclusions about QTY code and membrane protein solubilization to be made, rather than the conclusions that are currently enabled by the data, which are strictly about how a dynamic hydrogen bonding network in the CpxAQTY protein enables CpxAQTY to mimic some of the signaling function of CpxA.

If simulations of the CpxA model in a lipid membrane are included and more comparisons like those in Supplementary Figure 3 are made, this work could make more general, evidence-supported conclusions about the type of interactions necessary to move a membrane protein into water while still retaining its function. These comparison would be especially useful in the absence of structural data, which is one of the goals of protein solubilization.

Below are examples (in no particular order) from the manuscript that either limit the scope of the manuscript to a single designed protein (CpxAQTY) or make assumptions about the ways in which CpxAQTY mimics the function of CpxA, and would benefit from the simulation of CpxA in the lipid environment.

Response: We thank the reviewer for this insightful suggestion. We agree that simulations of the CpxA AlphaFold2 model in a lipid membrane would largely enhance the strength of our validation for the hypotheses we made. Therefore, we have conducted classical MD simulations and PBmetaD simulations of WT CpxA.

We ran 100-ns MD simulations of CpxA in the POPC bilayer. Similar to CpxA^{QTY} simulations, we input the AlphaFold2 model without the catalytic domain. We observed similar overall structural stability (Supplementary Fig. 8a) and local flexibility (Supplementary Fig. 8b) to those of CpxA^{QTY}. The periplasmic-facing region of CpxA TM domain exhibited high RMSF, consistent with the trend seen in CpxA^{QTY}, suggesting their similar dynamics. We compared interactions in CpxA and CpxA^{QTY}, and have provided some examples (Fig. 7), demonstrating that hydrogen bonding between QTY residues in CpxA mimics the packing between hydrophobic residues in CpxA^{QTY} (replacing the previous Supplementary Fig. 3, which showed comparison with CpxA AlphaFold2 model).

Supplementary Figure 8 | RMSD and RMSF curves in MD simulations of CpxA. a, Backbone RMSD of the transmembrane domain, versus medoid frame. **b,** RMSF of the transmembrane domain, versus medoid frame.

Fig. 7 | Interactions of polar residues in CpxA^{QTY} mimicked the packing of the original hydrophobic residues in CpxA, revealed by MD simulations. a, (Left) Q17 in Chain A and Q17 in Chain B of CpxA^{QTY}. (Right) L17 in Chain A and L17 in Chain B of CpxA. **b,** (Left) Q21 in Chain A and T24 in Chain B of CpxA^{QTY}. (Right) L21 in Chain A and V24 in Chain B of CpxA. **c,** (Left) T12 in Chain B and Q182 in Chain B of CpxA^{QTY}. (Right) I12 in Chain B and L182 in Chain B of CpxA. Transmembrane regions are shown as cyan cartoon in CpxA^{QTY} simulation (left) and yellow cartoon in CpxA (right). Highlighted residues are shown as spheres. The structural snapshots were from the simulation medoid frames.

Moreover, we ran 2000-ns PBmetaD simulations of CpxA. Similar to CpxA^{QTY}, we observed two stable states and one intermediate state (Supplementary Fig. 9f). The conformational transition between the two stable states also corresponds to diagonal scissoring motion (Supplementary Fig. 9g). By comparing the interhelical interactions with those in CpxA^{QTY}, we found that many hydrophobic residues went through similar transition of roles in the conformation compared to those corresponding QTY residues, and provided the example of A: L/Q21 and A: L/Q168 in the top layer of the TM core (Supplementary Fig. 9h). The consistent performance of CpxA^{QTY} and CpxA in both MD simulations and PBmetaD simulations further supports that CpxA^{QTY} shares the similar structural characteristics and conformational changes with wild-type CpxA.

Supplementary Figure 9 (f-h) | PBMetaD simulation of CpxA. **f**, The curves of the CVs with simulation time. The patterns of the CV variation were classified into three states, separated by grey dashed lines. **g**, The conformational transition of the different states of the transmembrane domain. Top view of the conformations of State-1 (cyan), State-M (transparent grey) and State-2 (red), as viewed from the periplasm looking into the cytoplasm. “TM1” indicates TM1 helix in Chain A and “TM1’ ” indicates TM1 helix in Chain B. **h**, Snapshots of the top layer of the interaction network of State-1 (left) and State-2 (right). The display style is the same as Fig. 6d. The structural snapshots of **g** and **h** were from the medoid frames of the clusters, corresponding to **e**.

Please see the detailed contents in the newly-added section *CpxA shows similar structural dynamics and conformational changes*.

Additionally, we also made responses to your questions/suggestions below.

A. In the introduction there is mention of activity preservation and “the specific interaction network essential for the intrinsic biological [activity]’ in the unmodified protein.

The hydrogen bonding study suggests that these are the essential interactions for activity for CpxAQTY, but what are the essential interactions in CpxA?

Response: For CpxA, the essential interactions should be some essential contact of some hydrophobic residues, possibly also some hydrogen bonds formed by the several polar residues in TM domain. For CpxA^{QTY}, the essential interactions should be mainly the hydrogen bonds formed by QTY residues, original polar residues and water molecules, possibly also some hydrophobic contact.

As for the specific essential interactions for activity, it warrants solid experimental evidence. In this revision, we indeed found four QTY residues in the TM domain that were very likely to be essential to signaling activity. Please see the detailed contents in the newly-added section *Point mutations diminish the signaling activity*.

B. “Another advantage of this strategy, i.e., mimicking themselves and substituting both interior and exterior residues, is its compatibility to conformational changes essential to function.”

This sentence states an assumption that the CpxAQTY protein behaves the same in water as the CpxA protein behaves in lipid. Simulation of CpxA in lipid would provide insight into conformational states explored in lipid similar to those explored in water and corroborate the assumption.

Response: Our simulations of CpxA indeed revealed the consistency of the behavior between CpxA^{QTY} with CpxA. The conformational changes explored in lipids are indeed similar to those explored in water.

C. “Theoretically, designed interhelical polar interaction networks by QTY code may replace the interaction networks in transmembrane hydrophobic helices responsible for SHK signaling and realize signal transduction through the conformational changes that mimic the original ones.”

*What is the dynamic hydrogen bond network introduced by QTY code mimicking?
Why is function therefore retained?*

Response: The dynamic hydrogen bond network introduced by QTY code mimics the packing of the original hydrophobic residues, as shown in Fig. 7. We currently attributed the functional preservation mainly to two aspects.

- 1) Remarkable preservation of the original structure. The high similarity between the QTY residues and the original hydrophobic residues determines the basis of the structural preservation. Furthermore, QTY design replaces both the exposed and buried hydrophobic residues in transmembrane domain with polar residues, mimicking the circumstances of the original membrane protein in the lipids.
- 2) Remarkable preservation of the structural flexibility. The significance of the structural flexibility in transmembrane signaling has been discussed in the fourth paragraph of Discussion section. The preserved flexibility enables the essential conformational changes for activity.

D. “In summary, the structural flexibility introduced by QTY design could play an important role in transmembrane signaling of CpxAQTY.”

It would be useful to know if the dynamic nature of the water-solubilized structure is comparable to that of the membrane structure in order to draw parallels between the hydrogen bonding network and the hydrophobic interactions in the membrane. Perhaps the dynamic nature of the protein makes structural characterization challenging.

Response: The dynamic nature of CpxA^{QTY} is indeed comparable to that of CpxA, comparing the RMSF and RMSD curves of the two, and the conformational changes.

- What interactions are replacing the hydrogen bonding networks found in CpxAQTY (particularly the hydrogen bonds to water) in the CpxA protein?

Response: As discussed in the Comment 2-C, the hydrogen bonding networks found in CpxA^{QTY} are replacing the packing of the hydrophobic residues. We raise the example of L17. L17 of the both chains in CpxA simply interact by packing (Fig. 7a), whereas Q17 of the both chains in CpxA^{QTY} interact not only by a direct H-bond (Fig. 5c), but also by water-mediated H-bonds (Extended Data Fig. 1c-j). It suggests that water molecules help the buried polar residues maximally hydrogen bonded.

- What insight do we gain into the function of CpxA to enable the use of QTY as a tool for study of other membrane proteins?

Response: We suppose that this question can also be answer by the response to Comment 2-C.

E. “The simple restoration of the original hydrophobic core from the polar core of the transmembrane helical bundle resulted in the loss of the transmembrane signaling receptor activity, underscoring the crucial role of the H-bond network in transmembrane signaling of CpxAQTY.”

This statement highlights the crucial role of a dynamic structure, enabled by highly dynamic hydrogen bond network. How does this pertain to the case of membrane-bound CpxA?

Response: In CpxA, the dynamic nature seems close to that of CpxA^{QTY}. Since the predominant content of hydrophobic residues in TM domain of CpxA, especially the flexible periplasm-facing region, we infer that the flexibility might be from the loose

packing. In fact, the role of the flexibility caused by loose packing in transmembrane signaling has been discussed in multiple studies³²⁻³⁵.

F. “From these results, we reasoned that extensive H-bond formation might account for the stabilization of the transmembrane domain of CpxAQTY. Moreover, the dynamics of the H-bond network provided appropriate structural flexibility, largely contributed by QTY side chains and water molecules.”

A discussion of structural flexibility in CpxA would make this work more generalizable.

Response: We have discussed the structural flexibility in CpxA in the last comment and in the main text.

G. “we reasoned that extensive H-bond formation might account for the stabilization of the transmembrane domain of CpxAQTY”

The hydrogen bonding data from Figure 5 are relevant to how CpxAQTY is soluble, maintains a folded state predicted by AlphaFold, and suggests how it might be performing signaling function, but do not give insight into how the native protein, CpxA, behaves.

Response: In the newly-added text and figures, we have shown the interactions inside the CpxA transmembrane domain and conformational changes.

Minor comments

3. Typo in Fig 3. ‘Phosphotransfase activity’ should be phosphotransferase activity

Response: We thank the reviewer for the correction. We have corrected it.

4. From the text the “signal-blind control protein (CpxAc) does not contain the transmembrane region and it’s purely the cytoplasmic portion of the protein serving as a control. These two constructs should perform similarly as controls, but please

specify in the text for clarity that CpxAc is composed solely of the cytoplasmic portion and briefly discuss why the water-soluble transmembrane region is not part of the control.

Response: We thank the reviewer for raising this concern. We agree that we should clarify why the water-soluble transmembrane region is not part of the control.

We did not include the water-soluble transmembrane domain because it is rare that SHKs sense signals through the transmembrane domain, especially when a sensor domain exists; additionally, considering that the transmembrane domain is separated by the sensor domain in the sequence, removing only the sensor domain and artificially adding a linker would be challenging to maintain its original conformational constraints on the transmembrane domain.

We have added the discussion about this concern in the text:

Page 9: “Here we did not include the water-soluble transmembrane domain considering the rarity of signal-sensing by SHK transmembrane domains and the difficulty of restoring the original conformational constraints of the sensor domain on the transmembrane domain using a designed linker.”

5. Please specify what is meant by “with great variation (58%)” in the following sentence:

“Therefore, we verified that CpxA^{QTY} responded to pH variations in a highly similar manner to CpxA by signaling through the designed transmembrane domain, though with great variation (58%).”

I assume this is sequence variation, but the way this sentence currently reads suggests variation in pH response.

Response: We thank the reviewer for the kind reminder. We agree this statement may not precisely convey the message. We have revised it as:

Page 9: “...though with great variation **in sequence** (58%).”

6. The following sentence is confusing. Please break it up.

“Therefore, the high content of Gln, determined by the fact that Leu contributes the most to the composition of transmembrane α -helices, in QTY-designed transmembrane domain might contribute greatly to H-bond formation and indeed we observed that Gln performed very well in H-bond formation, in both stable and mobile manners (Fig. 5e, Extended Data Fig. 1c-j).”

It reads more clearly with a period after ‘...contribute greatly to H-bond formation’.

Response: We thank the reviewer for the suggestion. We have broken this sentence up into two sentences.

Page 21: “...in QTY-designed transmembrane domain might contribute greatly to H-bond formation. **In fact, we indeed** observed that Gln performed very well in H-bond formation”

7. The data in supplementary Figure 3 would do well in the main text as it visually highlights the power of the QTY design approach. Please consider incorporating it into the MD simulation main text figure. See comment 2.

Response: We thank the reviewer for the valuable suggestion. We agree that this figure could clearly show the power of QTY design. In this revision, we used this figure as Fig. 7 in the main text. Nonetheless, this time the CpxA structures are from MD simulations, instead of the AlphaFold2 model, since we have conducted the MD simulations.

8. The use of italicized transmembrane to describe the CpxAQTY transmembrane domain is inconsistent (see the MD results section). Ensure consistency throughout the text for clarity.

Response: We thank the reviewer for the kind reminder. In fact, here the main idea of using the italicized *transmembrane* is to “differentiate the signaling through the

water-soluble transmembrane domain from the bona fide transmembrane signaling”. We aimed to clarify that CpxA^{QTY} cannot transduce the signals **across the membrane**, yet we supposed that we could still called the water-soluble transmembrane domain as the “transmembrane” domain. Nonetheless, reminded by the reviewer, we found several places that should use the italicized form, for example:

Page 17: “...underscoring the crucial role of the H-bond network in *transmembrane* signaling of CpxA^{QTY}”

9. Please clarify throughout the text that all of structures in this work are predicted models.

Example: “mimicked the packing of the original hydrophobic residues in CpxA indicating the structural preservation by QTY design.”

could be: “mimicked the packing of the original hydrophobic residues in the predicted structure model of CpxA indicating the structural preservation by QTY design.”

Response: We thank the reviewer for the kind reminder. We agree that it is necessary to clarify this important information. We have made additional statement about the use of AlphaFold2 in several places. (The sentence in the example given by the reviewer has been replaced by new results, thus not being shown below)

Fig. 3a: “The predicted structure of CpxA^{QTY} by AlphaFold2 was shown.”

Page 13: “From these results of MD simulations of CpxA^{QTY} AlphaFold2 structure...”

10. Please provide the superposition of the transmembrane four-helix bundle of AlphaFold2 models of CpxA^{QTY} and CpxA^{QTY-v.2} along with backbone RMSD in supplementary Figure 2, as this highlights the effect of only modifying surface

residues on the structure of the transmembrane region.

Response: We thank the reviewer for the suggestion. We agree that we should also provide the superposition of CpxA^{QTY} and CpxA^{QTY-v.2}. We have added the figures below to Supplementary Fig. 2 and updated the text.

Supplementary Figure 2 (f-g) | AlphaFold2 models. f-g, Superposition of the transmembrane four-helix bundle of AlphaFold2 models of CpxA^{QTY} (cyan) and CpxA^{QTY-v.2} (green). The display style is the same as in **d-e**.

Page 17: “We generated the structure model of CpxA^{QTY-v.2} by AlphaFold2, which showed high prediction quality in the transmembrane domain **and a similar structure to CpxA^{QTY}** (Supplementary Fig. 2c, f, g).”

11. CpxA^{QTY} phosphotransferase activity seems to be significantly faster than phosphotransferase activity of CpxA in proteoliposomes (based on reference 30), an observation that the text briefly mentions.

While this difference could be attributed to experimental setup, this change in rate is potentially a result of protein accessibility in the absence of lipids. This observation should add to the discussion from comment 1 about the difference between studying the water-soluble protein and the membrane-bound protein.

Response: We thank the reviewer for raising this concern. We believe our response to Comment 1 above could address this concern.

References

1. Akdel, M. et al. A structural biology community assessment of AlphaFold2 applications. *Nat Struct Mol Biol* **29**, 1056-1067 (2022).
2. Mirdita, M. et al. ColabFold: making protein folding accessible to all. *Nat Methods* **19**, 679-682 (2022).
3. Ibrahim, T. et al. AlphaFold2-multimer guided high-accuracy prediction of typical and atypical ATG8-binding motifs. *PLoS Biol* **21**, e3001962 (2023).
4. Zhu, W., Shenoy, A., Kundrotas, P. & Elofsson, A. Evaluation of AlphaFold-Multimer prediction on multi-chain protein complexes. *Bioinformatics* **39**(2023).
5. Yin, R., Feng, B.Y., Varshney, A. & Pierce, B.G. Benchmarking AlphaFold for protein complex modeling reveals accuracy determinants. *Protein Sci* **31**, e4379 (2022).
6. Buschiazzo, A. & Trajtenberg, F. Two-Component Sensing and Regulation: How Do Histidine Kinases Talk with Response Regulators at the Molecular Level? *Annu Rev Microbiol* **73**, 507-528 (2019).
7. Jacob-Dubuisson, F., Mechaly, A., Betton, J.M. & Antoine, R. Structural insights into the signalling mechanisms of two-component systems. *Nat Rev Microbiol* **16**, 585-593 (2018).
8. Zschiedrich, C.P., Keidel, V. & Szurmant, H. Molecular Mechanisms of Two-Component Signal Transduction. *J Mol Biol* **428**, 3752-75 (2016).
9. Gao, R. & Stock, A.M. Biological insights from structures of two-component proteins. *Annu Rev Microbiol* **63**, 133-54 (2009).
10. Raivio, T.L. & Silhavy, T.J. Transduction of envelope stress in Escherichia coli by the Cpx two-component system. *J Bacteriol* **179**, 7724-33 (1997).
11. Danese, P.N. & Silhavy, T.J. CpxP, a Stress-Combative Member of the Cpx Regulon. *J Bacteriol* **180**, 831-839 (1998).
12. Clarke, E.J. & Voigt, C.A. Characterization of combinatorial patterns generated by multiple two-component sensors in E. coli that respond to many stimuli. *Biotechnol Bioeng* **108**, 666-75 (2011).
13. Weatherspoon-Griffin, N., Yang, D., Kong, W., Hua, Z. & Shi, Y. The CpxR/CpxA two-component regulatory system up-regulates the multidrug resistance cascade to facilitate Escherichia coli resistance to a model antimicrobial peptide. *J Biol Chem* **289**, 32571-82 (2014).
14. Xu, Y. et al. An acid-tolerance response system protecting exponentially growing Escherichia coli. *Nat Commun* **11**, 1496 (2020).
15. Fleischer, R., Heermann, R., Jung, K. & Hunke, S. Purification, reconstitution, and characterization of the CpxRAP envelope stress system of Escherichia coli. *J Biol Chem* **282**, 8583-93 (2007).
16. Mechaly, A.E., Sassoon, N., Betton, J.M. & Alzari, P.M. Segmental helical motions and dynamical asymmetry modulate histidine kinase autophosphorylation. *PLoS Biol* **12**, e1001776 (2014).
17. Mechaly, A.E. et al. Structural Coupling between Autokinase and Phosphotransferase Reactions in a Bacterial Histidine Kinase. *Structure* **25**, 939-944.e3 (2017).
18. Kwon, E. et al. The crystal structure of the periplasmic domain of Vibrio parahaemolyticus

-
- CpxA. *Protein Sci* **21**, 1334-43 (2012).
19. Hörnschemeyer, P., Liss, V., Heermann, R., Jung, K. & Hunke, S. Interaction Analysis of a Two-Component System Using Nanodiscs. *PLoS One* **11**, e0149187 (2016).
 20. Bouillet, S., Wu, T., Chen, S., Stock, A.M. & Gao, R. Structural asymmetry does not indicate hemiphosphorylation in the bacterial histidine kinase CpxA. *J Biol Chem* **295**, 8106-8117 (2020).
 21. Clark, I.C. et al. Protein design-scapes generated by microfluidic DNA assembly elucidate domain coupling in the bacterial histidine kinase CpxA. *Proc Natl Acad Sci U S A* **118**, e2017719118 (2021).
 22. Zhang, S. et al. QTY code enables design of detergent-free chemokine receptors that retain ligand-binding activities. *Proc Natl Acad Sci U S A* **115**, E8652-E8659 (2018).
 23. Qing, R. et al. Scalable biomimetic sensing system with membrane receptor dual-monolayer probe and graphene transistor arrays. *Sci Adv* **9**, eadf1402 (2023).
 24. Miot, M. & Betton, J.M. Reconstitution of the Cpx signaling system from cell-free synthesized proteins. *N Biotechnol* **28**, 277-81 (2011).
 25. Boyken, S.E. et al. De novo design of protein homo-oligomers with modular hydrogen-bond network-mediated specificity. *Science* **352**, 680-687 (2016).
 26. Wang, W. & Roberts, C.J. Protein aggregation - Mechanisms, detection, and control. *Int J Pharm* **550**, 251-268 (2018).
 27. Trevino, S.R., Scholtz, J.M. & Pace, C.N. Measuring and increasing protein solubility. *J Pharm Sci* **97**, 4155-66 (2008).
 28. Goverde, C.A. et al. Computational design of soluble analogues of integral membrane protein structures. *bioRxiv*, 2023.05.09.540044 (2023).
 29. Guo, H.B. et al. AlphaFold2 models indicate that protein sequence determines both structure and dynamics. *Sci Rep* **12**, 10696 (2022).
 30. Ruff, K.M. & Pappu, R.V. AlphaFold and Implications for Intrinsically Disordered Proteins. *J Mol Biol* **433**, 167208 (2021).
 31. Varadi, M. et al. AlphaFold Protein Structure Database: massively expanding the structural coverage of protein-sequence space with high-accuracy models. *Nucleic Acids Res* **50**, D439-d444 (2022).
 32. Gushchin, I. et al. Mechanism of transmembrane signaling by sensor histidine kinases. *Science* **356**, eaah6345 (2017).
 33. Gordeliy, V.I. et al. Molecular basis of transmembrane signalling by sensory rhodopsin II-transducer complex. *Nature* **419**, 484-7 (2002).
 34. Manson, M.D. Transmembrane signaling is anything but rigid. *J Bacteriol* **193**, 5059-61 (2011).
 35. Barnakov, A., Altenbach, C., Barnakova, L., Hubbell, W.L. & Hazelbauer, G.L. Site-directed spin labeling of a bacterial chemoreceptor reveals a dynamic, loosely packed transmembrane domain. *Protein Sci* **11**, 1472-81 (2002).

REVIEWER COMMENTS

Reviewer #1 (Remarks to the Author):

My concerns have been addressed.

Reviewer #2 (Remarks to the Author):

In the updated version of the manuscript, the authors resolved some of the issues raised during the previous review round and corrected some of the exaggerated claims made previously.

However, the main issue remains unaddressed. The premise of the work is that membrane proteins are difficult to study and that the QTY code somehow helps to study them by rendering them soluble. However, presented data do not reflect this at all. There is no structural information on signaling mechanism of WT CpxA, there is no structural information on signaling mechanism of designed CpxA-QTY, consequently no claim on whether the mechanism is conserved after QTY redesign, and there is no new information about WT CpxA obtained by studying CpxA-QTY. The authors redesigned one of several domains of multidomain protein, and the domains that were not redesigned conserved their function. The data does not support (and does not contradict) the assumption that the signal is transduced through the redesigned domain, especially given that the nature of the signal is not known. Therefore, I believe that the revised manuscript is not suitable for publication in Nature Communications and unlikely to be improved by further revision.

List of issues:

1) Abstract mentions "flexibility introduced by QTY code". If QTY code explicitly changes any physico-chemical properties of the protein, it inevitably changes the mechanism of signal transduction, which makes the approach unfit for gaining information on WT proteins. Replacing oily hydrophobic interfaces with sticky polar interfaces very probably affects not only flexibility but other properties of the protein as well, and thus precludes gaining information on WT protein.

2) The authors present modeling data that transmembrane helices in original and redesigned CpxA display scissoring motion. However, no data is presented that would support the assumption that this degree of freedom is linked to signal transduction in CpxA and that it is the only degree of freedom of the protein that is needed/utilized for signal transduction.

In their responses, the authors write "we aim to show that the general signaling mechanism is conserved (how the signals are sensed and transduced; what are the major conformational changes like)", however they do not answer any of these questions. They do not show that pH is sensed by sensor domain of CpxA-QTY and do not show how the corresponding signal is transduced towards the catalytic domain of the kinase. Given that pH is a very general condition affecting all titratable residues, it very well might be that pH is sensed by CpxA-QTY via a different mechanism compared to CpxA.

3) The authors omit the direct comparison of conformations of WT CpxA and CpxA-QTY from the main text, although in my view it is very informative: it shows that conformations are different. TM1 distances in the two states are 9 and 19 Å in WT CpxA and 10 and 17 Å in CpxA-QTY; TM2 distances in the two states are 20 and 15 Å in WT CpxA and 17 and 12 Å in CpxA-QTY (Supplementary Figs. 6de, 9cd). These differences are far from negligible and show that QTY approach changes the structural properties of the domain, which makes the method unfit for its purpose.

4) In their responses, the authors write "any method undergoes continuous refinements until proven robust". What refinement of the method is presented in the manuscript under consideration, given that the method consists of replacing TM amino acids with Q/T/Y?

5) Fig 2d: is the plot correct? The total is below the value for α -helices around 200 nm, where β -sheet signal is positive, and also around 220 nm, where β -sheet signal is negative.

6) Fig 4c: why do you normalize phosphorylation level to 1? Current plot is misleading. Given the loading control signals, it appears that the signal from CpxA-QTY is much lower than that of CpxAc.

7) Regarding the model of the relative arrangement of sensor domains: the authors reference PDB ID 8UK7, where two molecules in the asymmetric unit are oriented with their α -helices towards each other, as would normally be expected for cytoplasmic sensor domains of receptor histidine kinases, although at a large angle. The authors cannot claim that data from PDB ID 8UK7 supports their model.

Regarding PDB ID 3V67, where the domain is presumably monomeric, for NarQ and NarX pair, which are relatively well-studied, it was shown that the sensor domain of NarX in the apo form is also monomeric, while the full-length protein clearly functions as a dimer with a traditional dimerization interface.

Disregarding PDB ID 8UK7 and NarQ/NarX, the authors still do not provide any support or validation for the AlphaFold model used in simulations, which has low confidence pLDDT values for the inter-domain linkers; the structure of the linkers is key for signal transduction.

Reviewer #3 (Remarks to the Author):

The revised version of this manuscript does a thorough job of addressing reviewer comments and suggestions. The new simulation and point mutation data provide interesting structure-function insight.

Comments for further discussion and analysis

1. The authors do a good job of pointing out differences and similarities between CpxAQTY and CpxA along the manuscript text. It would be useful to also have a summary table in the supplementary information for quick reference.

2. For the PBMetaD simulations: In state 1, distances between helices appear shorter in CpxAQTY than in CpxA (Figure 6B and Supplementary Fig 9F). Perhaps this is indicative of how the collective variable is defined, or potentially demonstrates a structural difference between the native and solubilized proteins that results in the observed functional differences.

In a supplementary figure, please include comparisons between CpxAQTY and CpxA of structural cluster centroids for state 1, 2 and M (for example: structural centroid of state 1 CpxA overlapped with state 1 of CpxAQTY with RMSD or distance analysis, the same for state 2 and state M). This type of comparison could be useful to begin to understand how structural differences introduced by the QTY design change the functional landscape of the protein.

A mention of this comparison would fit nicely within the following paragraph from the Discussion section.

“Despite the remarkable preservation of the molecular function by QTY design, we still noticed some discrepancies in functional performance between CpxAQTY in water and CpxA in lipid environment...”

Responses to the reviewers' comments point-by-point

We sincerely appreciate reviewers' professional and constructive comments and suggestions. We have read through comments carefully and have made corrections accordingly. Revisions in the text are highlighted in red. With regard to reviewers' comments and suggestions, we reply as follows:

Reviewer #1:

My concerns have been addressed.

Response: Many thanks for the reviewer's encouragement and positive comments.

Reviewer #2:

In the updated version of the manuscript, the authors resolved some of the issues raised during the previous review round and corrected some of the exaggerated claims made previously.

However, the main issue remains unaddressed. The premise of the work is that membrane proteins are difficult to study and that the QTY code somehow helps to study them by rendering them soluble. However, presented data do not reflect this at all. There is no structural information on signaling mechanism of WT CpxA, there is no structural information on signaling mechanism of designed CpxA-QTY, consequently no claim on whether the mechanism is conserved after QTY redesign, and there is no new information about WT CpxA obtained by studying CpxA-QTY. The authors redesigned one of several domains of multidomain protein, and the domains that were not redesigned conserved their function. The data does not support (and does not contradict) the assumption that the signal is transduced through the redesigned domain, especially given that the nature of the signal is not known. Therefore, I believe that the revised manuscript is not suitable for publication in Nature Communications and unlikely to be improved by further revision.

Response: We appreciate all the time and effort from the reviewer, and thank the reviewer for reviewing our manuscript. The most of reviewer's comments and suggestions have prompted us to improve quality of our manuscript.

Regarding the "structural information", we provided structural information through widely accepted AlphaFold2 protein structure prediction, as well as validation data and molecular dynamic simulations to evaluate their performance. In the first-round revision, based on our AlphaFold2 structural information and subsequent simulations, the reviewer requested additional simulations of WT CpxA in a lipid bilayer and other additional work related to AlphaFold2 structural information. And we diligently completed the required work and tried our best to address the concerns. It is also likely that the reviewer is referring to experimental structure. While we indeed have not experimentally resolved the structure of either WT CpxA or CpxA^{QTY}, this was out of the scope of the current manuscript. Structural determination of either WT CpxA or CpxA^{QTY} is a multi-year effort with significant additional funding required. Thus, it is for a completely new manuscript. The main focus of the current manuscript is the functional preservation and the insights into the possible principles, instead of the detailed mechanism or gaining new information about the WT protein.

As for the assumption that the signal is transduced through the redesigned domain, in our initial manuscript, we have demonstrated that the pH-response capability, preserved in full-length CpxA^{QTY}, is absent in CpxAc (the sole cytoplasmic portion); in the first-round revised manuscript, the pH sensitivity was reduced/abolished in CpxA^{QTY} mutants with mutations in the sensor or transmembrane domain. We believe these data unambiguously support that the signal is transduced through the redesigned domain.

We sincerely hope that the reviewer could reexamine our efforts and understand the main idea of our current manuscript.

Below are the detailed responses to the concerns in this round.

1) Abstract mentions "flexibility introduced by QTY code". If QTY code explicitly changes any physico-chemical properties of the protein, it inevitably changes the

mechanism of signal transduction, which makes the approach unfit for gaining information on WT proteins. Replacing oily hydrophobic interfaces with sticky polar interfaces very probably affects not only flexibility but other properties of the protein as well, and thus precludes gaining information on WT protein.

Response: We thank the reviewer for the comment. First, regarding the statement “*Computational approaches suggest that an extensive and dynamic hydrogen-bond network and its flexibility introduced by QTY code may play an important role*”, we agree that it seems somewhat misleading, likely suggesting that QTY code significantly influences flexibility. In fact, what we actually want to emphasize is that the extensive and dynamic hydrogen-bond network introduced by QTY code plays an important role, while its flexibility is also important. We have revised it as:

ABSTRACT: “Computational approaches suggest that an extensive and dynamic hydrogen-bond network **introduced by QTY code** and its flexibility may play an important role.”

As mentioned in our first-round response, we aim to show that the **general** signaling mechanism is conserved (how the signals are sensed and transduced; what are the major conformational changes like), rather than the detailed mechanism (which are the key residues and interactions). We do not deny that water solubilization design may influence certain properties and structural details of the protein to some extent, but this does not mean that it cannot aid in the study of WT proteins. As our functional characterization demonstrated, the designed proteins retained multiple molecular functions of the WT protein, albeit with some quantitative differences. The successful preservation of functional integrity in our current work implies that characterizing the function of water-solubilized variants designed with QTY code can provide supports and insights into the WT membrane proteins whose molecular function have been rarely characterized, especially for those, of which the *in vitro* system stabilized by lipids with expected activities, are difficult to construct.

2) The authors present modeling data that transmembrane helices in original and redesigned CpxA display scissoring motion. However, no data is presented that would support the assumption that this degree of freedom is linked to signal transduction in CpxA and that it is the only degree of freedom of the protein that is needed/utilized for signal transduction.

In their responses, the authors write "we aim to show that the general signaling mechanism is conserved (how the signals are sensed and transduced; what are the major conformational changes like)", however they do not answer any of these questions. They do not show that pH is sensed by sensor domain of CpxA-QTY and do not show how the corresponding signal is transduced towards the catalytic domain of the kinase. Given that pH is a very general condition affecting all titratable residues, it very well might be that pH is sensed by CpxA-QTY via a different mechanism compared to CpxA.

Response: We thank the reviewer for the comment.

Regarding the scissoring motion, we actually have provided the references in the original manuscript to support that, it is very likely that scissoring motion is involved in signal transduction in CpxA. We have stated “*Diagonal scissoring, ..., is one of the most conserved conformational changes in SHK signaling²² observed in multiple structural studies of SHK signaling²⁰⁻²²*”. Ref.22¹ cited here conducted a comprehensive quantitative structural analysis of SHK domains and revealed that scissoring motion is the most conserved and largest conformational change pattern in SHK signaling, compared to the other two conformational change patterns (gearbox and piston shift motion). In the scarce structural studies of SHK transmembrane signaling, both the NarQ crystal structure (Ref.20²) and the cross-linking (Ref.22¹) and simulation data (Ref.21³) of PhoQ confirm the important role of scissoring motion. Moreover, we did not claim that “it is the only degree of freedom of the protein that is needed/utilized for signal transduction”. Other degrees of freedom may also play a role, but it is very hard to include all degrees of freedom as collective variables. We use the most conserved conformational change in SHK transmembrane signaling as the representative to demonstrate whether and how WT CpxA and CpxA^{QTY} can undergo similar conformational changes. Furthermore, we did not claim that this part of work demonstrates that the signaling mechanism of

WT CpxA and CpxA^{QTY} is identical; instead, we stated “CpxA^{QTY} shares the similar structural characteristics and conformational changes with wild-type CpxA”.

As for the answer to “how the signals are sensed and transduced; what are the major conformational changes like”, we actually made clear investigation and conclusions. The section “Point mutations diminish the signaling activity” we added last time demonstrates that the signal, namely pH, is sensed through the sensor domain and transduced through the transmembrane domain, consistent with WT CpxA. Our sections “Identification of scissoring motion in the transmembrane domain” and “CpxA shows similar structural dynamics and conformational changes” demonstrate that CpxA and CpxA^{QTY} show similar major conformational changes, namely scissoring motion.

Regarding pH, our initial data show that the cytoplasmic domain does not respond to pH, but full-length CpxA^{QTY} does respond to pH. Our additional data demonstrate that point mutations in both the sensor domain and the transmembrane domain can reduce or abolish the pH sensitivity of CpxA^{QTY}. These data unambiguously demonstrate that pH is sensed through the sensor domain and transduced through the transmembrane domain, which is consistent with WT CpxA.

3) The authors omit the direct comparison of conformations of WT CpxA and CpxA-QTY from the main text, although in my view it is very informative: it shows that conformations are different. TM1 distances in the two states are 9 and 19 Å in WT CpxA and 10 and 17 Å in CpxA-QTY; TM2 distances in the two states are 20 and 15 Å in WT CpxA and 17 and 12 Å in CpxA-QTY (Supplementary Figs. 6de, 9cd). These differences are far from negligible and show that QTY approach changes the structural properties of the domain, which makes the method unfit for its purpose.

Response: We thank the reviewer for raising this concern. We indeed observed some discrepancies between the conformations of WT CpxA and CpxA^{QTY}, comparing the results of PBmetaD simulations. In this revision, we have provided a supplementary figure and additional discussion about these differences. Below are the supplementary figure and additional discussion:

Page 16 (Results): “Additionally, by comparing the overall conformations, we have also identified some differences. For example, the scissoring motion of CpxA^{QTY} is more pronounced than that of CpxA; in State-1/M, the crossing angle between the TM1 helices of CpxA^{QTY} is larger than that of CpxA (Supplementary Fig. 10). These differences might be relevant to the functional performance discrepancies observed previously.”

Supplementary Figure 10 | Conformational comparisons of the transmembrane domains of CpxA and CpxA^{QTY} in the corresponding stable states revealed by PBmetaD simulations. a, State-1; b, State-M; c, State-2. CpxA is in yellow and CpxA^{QTY} in cyan. Side view is shown on the left and top view right. Superpositions of the α -C atoms were conducted in PyMOL. The structural snapshots were from the simulation medoid frames.

Page 22 (Discussion): “Considering the significant sequence variation, the structure of the transmembrane domain may be still slightly affected, **which is indeed observed in the simulations (Supplementary Fig. 10).**”

Nevertheless, we do not think these discrepancies render QTY code “unfit for its purpose”. As mentioned earlier, our focus is on the similarity between WT CpxA and CpxA^{QTY} in a qualitative manner, rather than quantitatively. We do not consider a difference of 1–3 Å to be profound. For instance, for the majority of determined structures, cryo-EM is routinely resolving macromolecular assemblies at resolutions between 2.5 and 4.0 Å⁴. In our previous revision, we discussed the differences in activity performance between WT CpxA and CpxA^{QTY}, which is similar to this question raised by the reviewer regarding conformation. We expect CpxA^{QTY} to retain the conformational change pattern of WT CpxA (with scissoring motion as the representative), which is highly relevant to functional preservation, but we accept that there may be some discrepancies in conformation, given the significant sequence variation.

4) In their responses, the authors write "any method undergoes continuous refinements until proven robust". What refinement of the method is presented in the manuscript under consideration, given that the method consists of replacing TM amino acids with Q/T/Y?

Response: We thank the reviewer for raising this question. This statement was our response to the reviewer’s first-round comment “*However, it is unlikely that a simple*

replacement rule would generally work for all transmembrane proteins, especially complex multistate signaling proteins”. We wrote this statement to mainly segue into the next sentence “Currently, while we consider the general replacement rules underlying QTY code to be effective, one of our ongoing efforts is to enhance QTY design by incorporating more rational elements”, where we pointed out the planned future improvement of QTY code. In the current manuscript, we did not intend to improve the approach itself; instead, as we stated in the abstract, “Our successful functional preservation further substantiates the robustness and comprehensiveness of QTY code”.

5) Fig 2d: is the plot correct? The total is below the value for α -helices around 200 nm, where β -sheet signal is positive, and also around 220 nm, where β -sheet signal is negative.

Response: We thank the reviewer for raising this question. This plot is correct, albeit with some unnecessary information not shown. Considering the simplicity, this plot only presents the data of α -helices and β -sheets, as we aimed to show the curve and content of the two most common ordered secondary structures. In fact, the secondary structure analysis software also presented the data of turns and loops. Below is the plot with complete information.

6) Fig 4c: why do you normalize phosphorylation level to 1? Current plot is misleading. Given the loading control signals, it appears that the signal from CpxA-QTY is much lower than that of CpxAc.

Response: We thank the reviewer for raising this question. It is very common to normalize western-blot bands to generate plots with error bars for analyzing the variation trend of the relative band intensity⁵⁻⁷. In Fig. 4c, our focus is on the trend of phosphorylation levels with varying signal concentrations, with triplicate data. We need to integrate data from triplicates, but obviously, a single blot cannot accommodate so many bands. Integrating data from different blots requires using the intensity of the band corresponding to a fixed condition as a reference and then normalizing to reflect the relative fold-change trend of the bands corresponding to other conditions. In Fig. 4c, the overall intensity difference of loading control bands in the two sets of blots (CpxA^{QTY} vs CpxAc) merely indicates differences in protein amounts on the different blots, which is very common and could be due to various experimental factors in western blotting. However, this does not affect our experimental purpose; we just need to ensure that the amount of the same protein on the same blot is close. Moreover, directly comparing band intensities on different blots is inappropriate, since the absolute band intensity is affected by multiple factors such as exposure time and background noise intensity. Furthermore, we did not intend to compare the basic autokinase activity of CpxA^{QTY} with that of CpxAc, as this is irrelevant to our experimental objectives.

7) Regarding the model of the relative arrangement of sensor domains: the authors reference PDB ID 8UK7, where two molecules in the asymmetric unit are oriented with their α -helices towards each other, as would normally be expected for cytoplasmic sensor domains of receptor histidine kinases, although at a large angle. The authors cannot claim that data from PDB ID 8UK7 supports their model. Regarding PDB ID 3V67, where the domain is presumably monomeric, for NarQ and NarX pair, which are relatively well-studied, it was shown that the sensor domain of NarX in the apo form is also monomeric, while the full-length protein clearly functions as a dimer with a traditional dimerization interface. Disregarding PDB ID 8UK7 and NarQ/NarX, the authors still do not provide any

support or validation for the AlphaFold model used in simulations, which has low confidence pLDDT values for the inter-domain linkers; the structure of the linkers is key for signal transduction.

Response: We thank the reviewer for raising this concern.

Regarding PDB ID 8UK7, we still suppose that the crystal structure displays crystallographic dimers. In this structure, the interface α -helices of the two monomers cross at approximately 90 degrees and only contact via two pairs of residues (Fig. R1). We do not consider this to be a typical dimerization pattern for sensor domains.

Fig. R1 | The crystal structure of CpxA sensor domain (PDB ID 8UK7). The protomers from different chains are in green and cyan, respectively. The residues that contact on the interface are indicated by sticks.

Regarding PDB ID 3V67, we did not claim that this structure suggests CpxA sensor domain is monomeric, but rather speculated “*the dimerization might not be as strong as that of others*” (typical SHK sensor domains). As for the example of NarX mentioned by the reviewer, firstly, the NarX/NarQ sensor domain and CpxA sensor domain are quite different in their domain classification: NarX/NarQ sensor domain is “all-helical sensing domain”⁸, while CpxA sensor domain is PAS domain⁹, which

is the most frequent sensor domain found in SHKs¹⁰ and typically consisting of a five-stranded β -sheet and five α -helices, so this example may not be appropriate. Furthermore, the example of NarX does not contradict our viewpoint. The sensor domain of NarX is monomeric in its apo form, indicating weak dimerization in the apo form, and undergoes conformational changes induced by ligand-binding to form a highly-stable dimer¹¹. Similarly, our AlphaFold2 model is very likely to represent the apo form conformation, with modest dimerization strength; whether binding of signal molecules induces major conformational changes in CpxA sensor domain and leads to a more stable conformation, requires further dedicated work investigation and falls outside the scope of our current work. Therefore, we still suppose that PDB ID 3V67 supports that the dimerization of CpxA sensor domain might be not as strong as that of typical SHK sensor domains, at least in the apo form.

For the AlphaFold2 model used in simulations, in the initial manuscript, we provided plots of pLDDT and PAE, demonstrating that the prediction confidence of these AlphaFold models is generally satisfactory. For the reviewer's concern regarding the sensor domain, we provided a dedicated supplementary discussion and figure in the last revision to demonstrate the validity of this AlphaFold model, even though the sensor domain is not the focus of our simulation section and is unlikely to have a decisive impact on the simulation results. Regarding the inter-domain linkers of interest to the reviewer, we have already explained to the reviewer in the previous response that these linkers are very likely to be loops, hence naturally showing low pLDDT values.

Reviewer #3:

The revised version of this manuscript does a thorough job of addressing reviewer comments and suggestions. The new simulation and point mutation data provide interesting structure-function insight.

Response: Many thanks for the reviewer's encouragement and positive comments. The reviewer's new suggestions are also very constructive and professional. We have carefully revised the manuscript. Please find the detailed responses to the concerns below.

1. The authors do a good job of pointing out differences and similarities between CpxA^{QTY} and CpxA along the manuscript text. It would be useful to also have a summary table in the supplementary information for quick reference.

Response: We thank the reviewer for this valuable suggestion. According to this suggestion, we added a summary table in the supplementary information:

Supplementary Table 1. Comparison of functional performance between CpxA^{QTY} in water and CpxA in lipid environment

Activity	CpxA (Ref.)	CpxA ^{QTY}
Autokinase	Autophosphorylation was almost saturated when t=10 min (in proteoliposomes ¹² , detergent ^{13,14} and nanodiscs ¹⁵)	(Lower) Autophosphorylation was unsaturated when t=10 min, with a significantly reduced rate
Phosphotransferase	Phosphorylated CpxA became barely detectable, with CpxR still acquiring phosphate groups when t=15 min (in proteoliposomes ¹² , and nanodiscs ¹⁵)	(Higher) Phosphorylated CpxA ^{QTY} became almost undetectable when t=0.5 min, with saturation of CpxR phosphorylation reached when t=1 min
Phosphatase	Phosphorylated CpxR became barely detectable when t=30 min (in proteoliposomes ¹² and detergent ¹⁴)	(Lower) Phosphorylated CpxR could still be detected when t=120 min
pH sensing	Compared to pH 7.0, the phosphorylation rate increased by ~ 2.5-fold at pH 7.5 and by ~ 4-fold at pH 8.0 (in proteoliposomes ¹²)	(Slightly higher) Compared to pH 7.0, the phosphorylation rate increased by ~ 2.7-fold at pH 7.5 and by ~ 7.5-fold at pH 8.0
CpxP sensing	Phosphorylation rate was inhibited by ~50% when with equimolar	(Lower) Phosphorylation rate was inhibited by ~50% when with 10-fold

	CpxP (in proteoliposomes ¹²)	CpxP
K ⁺ sensing	Minor response to 0.5 mM and major response to 500 mM (in proteoliposomes ¹²)	(Lower) Approximate response range was 10 to 100 mM

Note: For CpxA, the types of the lipid environment used in the references were indicated at the end of description; for CpxA^{QTY}, the apparent relative level of its activities compared to CpxA was indicated at the beginning.

2. For the PBMetaD simulations: In state 1, distances between helices appear shorter in CpxAQTY than in CpxA (Figure 6B and Supplementary Fig 9F). Perhaps this is indicative of how the collective variable is defined, or potentially demonstrates a structural difference between the native and solubilized proteins that results in the observed functional differences.

In a supplementary figure, please include comparisons between CpxAQTY and CpxA of structural cluster centroids for state 1, 2 and M (for example: structural centroid of state 1 CpxA overlapped with state 1 of CpxAQTY with RMSD or distance analysis, the same for state 2 and state M). This type of comparison could be useful to begin to understand how structural differences introduced by the QTY design change the functional landscape of the protein.

A mention of this comparison would fit nicely within the following paragraph from the Discussion section.

“Despite the remarkable preservation of the molecular function by QTY design, we still noticed some discrepancies in functional performance between CpxAQTY in water and CpxA in lipid environment...”

Response: We thank the reviewer for this insightful suggestion. We indeed observed some conformational differences between CpxA^{QTY} and CpxA in the PBmetaD simulations and we agree that a supplementary figure and additional discussion about these differences would benefit our manuscript. Below are the supplementary figure and additional discussion:

Page 16 (Results): “Additionally, by comparing the overall conformations, we have also identified some differences. For example, the scissoring motion of CpxA^{QTY} is

more pronounced than that of CpxA; in State-1/M, the crossing angle between the TM1 helices of CpxA^{QTY} is larger than that of CpxA (Supplementary Fig. 10). These differences might be relevant to the functional performance discrepancies observed previously.”

Supplementary Figure 10 | Conformational comparisons of the transmembrane domains of CpxA and CpxA^{QTY} in the corresponding stable states revealed by PBmetaD simulations. a, State-1; b, State-M; c, State-2. CpxA is in yellow and CpxA^{QTY} in cyan. Side view is shown on the left and top view right. Superpositions of the α -C atoms were conducted in PyMOL. The structural snapshots were from the simulation medoid frames.

Page 22 (Discussion): “Considering the significant sequence variation, the structure of the transmembrane domain may be still slightly affected, **which is indeed observed in the simulations (Supplementary Fig. 10).**”

References

1. Molnar, K.S. et al. Cys-scanning disulfide crosslinking and bayesian modeling probe the transmembrane signaling mechanism of the histidine kinase, PhoQ. *Structure* **22**, 1239-1251 (2014).
2. Gushchin, I. et al. Mechanism of transmembrane signaling by sensor histidine kinases. *Science* **356**, eaah6345 (2017).
3. Lemmin, T., Soto, C.S., Clinthorne, G., DeGrado, W.F. & Dal Peraro, M. Assembly of the transmembrane domain of E. coli PhoQ histidine kinase: implications for signal transduction from molecular simulations. *PLoS Comput Biol* **9**, e1002878 (2013).
4. Beckers, M., Mann, D. & Sachse, C. Structural interpretation of cryo-EM image reconstructions. *Prog Biophys Mol Biol* **160**, 26-36 (2021).
5. Wang, B., Zhao, A., Novick, R.P. & Muir, T.W. Activation and inhibition of the receptor histidine kinase AgrC occurs through opposite helical transduction motions. *Mol Cell* **53**, 929-40 (2014).
6. Ellermann, M. et al. Endocannabinoids Inhibit the Induction of Virulence in Enteric Pathogens. *Cell* **183**, 650-665.e15 (2020).
7. Zhang, P., Catterson, J.H., Grönke, S. & Partridge, L. Inhibition of S6K lowers age-related inflammation and increases lifespan through the endolysosomal system. *Nat Aging* (2024).
8. Zschiedrich, C.P., Keidel, V. & Szurmant, H. Molecular Mechanisms of Two-Component Signal Transduction. *J Mol Biol* **428**, 3752-75 (2016).
9. Kwon, E. et al. The crystal structure of the periplasmic domain of Vibrio parahaemolyticus CpxA. *Protein Sci* **21**, 1334-43 (2012).
10. Krell, T. et al. Bacterial sensor kinases: diversity in the recognition of environmental signals. *Annu Rev Microbiol* **64**, 539-59 (2010).
11. Cheung, J. & Hendrickson, W.A. Structural analysis of ligand stimulation of the histidine kinase NarX. *Structure* **17**, 190-201 (2009).
12. Fleischer, R., Heermann, R., Jung, K. & Hunke, S. Purification, reconstitution, and

-
- characterization of the CpxRAP envelope stress system of *Escherichia coli*. *J Biol Chem* **282**, 8583-93 (2007).
13. Mechaly, A.E., Sassoon, N., Betton, J.M. & Alzari, P.M. Segmental helical motions and dynamical asymmetry modulate histidine kinase autophosphorylation. *PLoS Biol* **12**, e1001776 (2014).
 14. Miot, M. & Betton, J.M. Reconstitution of the Cpx signaling system from cell-free synthesized proteins. *N Biotechnol* **28**, 277-81 (2011).
 15. Hörnschemeyer, P., Liss, V., Heermann, R., Jung, K. & Hunke, S. Interaction Analysis of a Two-Component System Using Nanodiscs. *PLoS One* **11**, e0149187 (2016).

REVIEWERS' COMMENTS

Reviewer #3 (Remarks to the Author):

My comments have been addressed.

Regarding the authors' responses to Reviewer 2:

Reviewer 2 takes issue with the claim that QTY code renders membrane proteins water-soluble while retaining intact function. I agree that the claim was too strong in the original manuscript, but the authors have made significant efforts to address this concern through revision. Additionally, the work demonstrates the ways in which applying the QTY code changes the structural and functional landscape of the protein and could be used for future studies into the mechanism of signaling that Reviewer 2 has requested but is outside of the scope of this manuscript.

The way that I read this manuscript is that the authors have a method to modify the original protein that does not drastically deform the initial protein's structure or function, while also rendering it water-soluble and therefore easier to work with in both experiment and simulation. Because QTY is still in its initial stages of refinement, this manuscript serves to validate the method more than to generate a water-soluble protein that is "identical" to the wild type. In my experience, no modification made to a protein in order to make studying it easier is ever going to give you the exact results you would get from a protein in its native environment. As long as the writing is reflective of this fact, I find the work compelling.